# Incentivize without Bonus: Provably Efficient Model-based Online Multi-agent RL for Markov Games

**Tong Yang** [1]   **Bo Dai** [2]   **Lin Xiao** [3]   **Yuejie Chi** [1 3]

## Abstract

Multi-agent reinforcement learning (MARL) lies at the heart of a plethora of applications involving the interaction of a group of agents in a shared unknown environment. A prominent framework for studying MARL is Markov games, with the goal of finding various notions of equilibria in a sample-efficient manner, such as the Nash equilibrium (NE) and the coarse correlated equilibrium (CCE). However, existing sample-efficient approaches either require tailored uncertainty estimation under function approximation, or careful coordination of the players. In this paper, we propose a novel model-based algorithm, called VMG, that incentivizes exploration via biasing the empirical estimate of the model parameters towards those with a higher collective best-response values of all the players when fixing the other players' policies, thus encouraging the policy to deviate from its current equilibrium for more exploration. VMG is oblivious to different forms of function approximation, and permits simultaneous and uncoupled policy updates of all players. Theoretically, we also establish that VMG achieves a near-optimal regret for finding both the NEs of two-player zero-sum Markov games and CCEs of multi-player general-sum Markov games under linear function approximation in an online environment, which nearly match their counterparts with sophisticated uncertainty quantification.

## 1. Introduction

Multi-agent reinforcement learning (MARL) is emerging as a crucial paradigm for solving complex decision-making problems in various domains, including robotics, game theory, and machine learning (Busoniu et al., 2008). While single-agent reinforcement learning (RL) has been extensively studied and theoretically analyzed, MARL is still in its infancy, and many fundamental questions remain unanswered. Due to the interplay of multiple agents in an unknown environment, one of the key challenges is the design of efficient strategies for exploration that can be seamlessly implemented in the presence of a large number of agents[1] without the need of complicated coordination among the agents. In addition, due to the large dimensionality of the state and action spaces, which grows exponentially with respect to the number of agents in MARL, it necessitate the adoption of function approximation to enable tractable planning in modern RL regimes.

A de facto approach in exploration in RL is the principle of optimism in the face of uncertainty (Lai, 1987), which argues the importance of quantifying the uncertainty, known as the *bonus* term, in the pertinent objects, e.g., the value functions, and using their upper confidence bound (UCB) to guide action selection. This principle has been embraced in the MARL literature, leading a flurry of algorithmic developments (Liu et al., 2021; Bai et al., 2021; Song et al., 2021; Jin et al., 2021; Li et al., 2022; Ni et al., 2022; Cui et al., 2023; Wang et al., 2023; Dai et al., 2024) that claim provable efficiency in solving Markov games (Littman, 1994), a standard model for MARL. However, a major downside of this approach is that constructing the uncertainty sets quickly becomes intractable as the complexity of function approximation increases, which often requiring a tailored approach. For example, near-optimal techniques for constructing the bonus function in the tabular setting cannot be applied for general function approximation using neural networks.

Therefore, it is of great interest to explore alternative exploration strategies without resorting to explicit uncertainty

---

[1]Department of Electrical and Computer Engineering, Carnegie Mellon University, Pittsburgh, PA, USA [2]School of Computational Science and Engineering, Georgia Institute of Technology, Atlanta, GA, USA [3]Meta AI. Correspondence to: Yuejie Chi <ychi@meta.com>.

*Proceedings of the 42$^{nd}$ International Conference on Machine Learning*, Vancouver, Canada. PMLR 267, 2025. Copyright 2025 by the author(s).

---

[1]In this paper, we use the term agent and player interchangeably.

quantification, and can be adopted even for general function approximation. Our work is inspired by the pioneering work of Kumar & Becker (1982), which identified the need to regularize the maximum-likelihood estimator of the model parameters using its optimal value function to incentivize exploration, and has been successfully applied to bandits and single-agent RL problems (Liu et al., 2020; Hung et al., 2021; Mete et al., 2021; Liu et al., 2024) with matching performance of their UCB counterparts. However, this strategy of *value-incentivized* exploration has not yet been fully realized in the Markov game setting; a few recent attempts (Liu et al., 2024; Xiong et al., 2024) made progress in establishing its statistical efficiency but fell short in designing computationally tractable algorithms. These limitations motivate the development of computationally efficient algorithms for the general MARL setting while enabling symmetric and independent updates of the players. We address the following question:

*Can we develop provably efficient algorithms for online multi-player general-sum Markov games with function approximation using value-incentivized exploration?*

### 1.1. Contribution

In this paper, we propose a provably-efficient model-based framework, named VMG (*Value-incentivized Markov Game solver*), for solving online multi-player general-sum Markov games with function approximation. VMG incentivizes exploration via biasing the empirical estimate of the model parameters towards those with a higher collective *best-response* values of all the players when fixing the other players' policies, thus encouraging the policy to deviate from the equilibrium of the current model estimate for more exploration. This approach is oblivious to different forms of function approximation, bypassing the need of designing tailored bonus functions to quantity the uncertainty in standard approaches. VMG also permits simultaneous and uncoupled policy updates of all players, making it more suitable when the number of players scales. Theoretically, we also establish that VMG achieves a near-optimal regret for a number of game-theoretic settings under linear function approximation, which are on par to their counterparts requiring explicit uncertainty quantification. Specifically, our main results are as follows.

- For two-player zero-sum matrix games, VMG achieves a near-optimal regret on the order of $\widetilde{O}(d\sqrt{T})$,[2] where $d$ is the dimension of the feature space and $T$ is the number of iterations for model updates. This translates to a sample complexity of $\widetilde{O}(d^2/\varepsilon^2)$ for finding an $\varepsilon$-optimal NE in terms of the duality gap.

---

[2]The notation $\widetilde{O}(\cdot)$ hides logarithmic factors in the standard order-wise notation.

- For finite-horizon multi-player general-sum Markov games, under the linear mixture model of the transition kernel, VMG achieves a near-optimal regret on the order of $\widetilde{\mathcal{O}}(d\sqrt{H^3T})$, where $H$ is the horizon length, and $T$ is the number of iterations for model updates. This translates to a near-optimal — up to a factor of $H$ — complexity of $\widetilde{O}(Nd^2H^4/\varepsilon^2)$ samples or $\widetilde{O}(Nd^2H^3/\varepsilon^2)$ trajectories for finding an $\varepsilon$-optimal CCE in terms of the optimality gap, which is also applicable to finding $\varepsilon$-optimal NE for two-player zero-sum Markov games. We also extend VMG to the infinite-horizon setting, which achieves a sample complexity of $\widetilde{\mathcal{O}}(Nd^2/((1-\gamma)^4\varepsilon^2))$ to achieve $\varepsilon$-optimality.

- The unified framework of VMG allows its reduction to important special cases such as symmetric matrix games, linear bandits and single-agent RL, which not only recovers the existing reward-biased MLE framework but also discovers new formulation that might be of independent interest.

### 1.2. Related work

We discuss a few threads of related work, focusing on those with theoretical guarantees.

**Two-player matrix games.** Finding the equilibrium of two-player zero-sum matrix games has been studied extensively in the literature, e.g., Mertikopoulos et al. (2018); Shapley (1953); Daskalakis & Panageas (2018); Wei et al. (2020), where faster last-iterate linear convergence is achieved in the presence of KL regularization (Cen et al., 2021; Zhan et al., 2023). Many of the proposed algorithms focus on the tabular setting with full information, where the expected returns in each iteration can be computed exactly when the payoff matrix is given. More pertinent to our work, O'Donoghue et al. (2021) considered matrix games with bandit feedback under the tabular setting, where only a noisy payoff from the players' actions is observed at each round, and proposed to estimate the payoff matrix using the upper confidence bounds (UCB) in an entry-wise manner (Lai, 1987; Boun-effouf, 2016), as well as K-learning (O'Donoghue, 2021) that is akin to Thompson sampling (Russo et al., 2018). Our work goes beyond the tabular setting, and proposes an alternative to UCB-based exploration that work seamlessly with different forms of function approximation.

**Multi-player general-sum Markov games.** General-sum Markov games are an important class of multi-agent RL (MARL) problems (Littman, 1994), and a line of recent works (Liu et al., 2021; Bai et al., 2021; Mao & Başar, 2023; Song et al., 2021; Jin et al., 2021; Li et al., 2022; Sessa et al., 2022) studied the non-asymptotic sample complexity for learning various equilibria in general-sum Markov games for the tabular setting under different data generation mechanisms. These works again rely heavily on carefully

constructing confidence bounds of the value estimates to guide data collection and obtain tight sample complexity bounds. In addition, policy optimization algorithms have also been developed assuming full information of the underlying Markov games, e.g., Erez et al. (2023); Zhang et al. (2022); Cen et al. (2023).

**MARL with linear function approximation.** Modern MARL problems often involve large state and action spaces, and thus require function approximation to generalize from limited data. Most theoretical results focus on linear function approximation, where the transition kernel, reward or value functions are assumed to be linear functions of some known feature maps. The linear mixture model of the transition kernel considered herein follows a line of existing works in both single-agent and multi-agent settings, e.g., Ayoub et al. (2020); Chen et al. (2022); Modi et al. (2020); Jia et al. (2020); Chen et al. (2022); Liu et al. (2024), which is subtly different from another popular linear model (Jin et al., 2020; Wang et al., 2019; Yang & Wang, 2019; Xie et al., 2020), and these two models are not mutually exclusive in general (Chen et al., 2022). Moreover, Ni et al. (2022); Huang et al. (2022) considered general function approximation and Cui et al. (2023); Wang et al. (2023); Dai et al. (2024) considered independent function approximation to allow more expressive function classes that lead to stronger statistical guarantees, which usually require solving complicated constrained optimization problems to construct the bonus functions.

**Exploration in online RL.** Uncertainty estimation is crucial for efficient exploration in online RL. Common approaches are constructing the confidence set of the model parameters based on the observed data, which have been demonstrated to be provably near-optimal in the tabular and linear function approximation settings (Jin et al., 2018; Agarwal et al., 2023) but have limited success in the presence of function approximation in practice (Gawlikowski et al., 2023). Thompson sampling provides an alternative approach to exploration by maintaining a posterior distribution over model parameters and sampling from this distribution to make decisions, which however becomes generally intractable under complex function approximation schemes (Russo et al., 2018). Zhang (2022) proposed feel-good Thompson sampling that biases towards models with higher optimal values, which is developed further for solving general RL problems in Zhong et al. (2022); Agarwal & Zhang (2022), to name a few.

**Exploration via optimization.** Our approach draws inspiration from the reward-biased maximum likelihood estimation framework, originally proposed by Kumar & Becker (1982), which has been recently adopted in the context of bandits (Liu et al., 2020; Hung et al., 2021; Cen et al., 2024) and single-agent RL (Mete et al., 2021; Liu et al., 2024; Yang et al., 2025). Liu et al. (2024) proposed an algorithm for two-player zero-sum Markov games, however, it requires asymmetric updates and solving bilevel optimization problems with the lower level problem being a Markov game itself. Xiong et al. (2024) tackled the general-sum multiplayer Markov game setting, however, the proposed algorithm therein is generally computationally intractable. Our work, in contrast, highlights a computationally-efficient algorithm for the general multi-player game-theoretic setting, which not only recovers but leads to new formulations for the single-agent setting. Last but not least, Foster et al. (2023); Chen et al. (2025) extended the seminal framework of decision-estimation coefficient (DEC) (Foster et al., 2021) to MARL, however the algorithms proposed therein are computationally expensive, due to the presence of minimax and constrained optimization subroutines.

### 1.3. Paper organization and notation

The rest of this paper is organized as follows. Section 2 studies two-player zero-sum matrix games, Section 3 focuses on episodic multi-player general-sum Markov games, and we conclude in Section 4. The proofs as well as the extension to the infinite-horizon setting are deferred to the appendix.

**Notation.** We let $[n]$ denote the index set $\{1, \ldots, n\}$. Let $I_n$ denote the $n \times n$ identity matrix, and inner product in Euclidean space $\mathbb{R}^n$ by $\langle \cdot, \cdot \rangle$. We let $\Delta^n$ denote the $n$-dimensional simplex, i.e., $\Delta^n = \{x \in \mathbb{R}^n : x \geq 0, \sum_{i=1}^n x_i = 1\}$. For any $x \in \mathbb{R}^n$, we let $\|x\|_p$ denote the $\ell_p$ norm of $x, \forall p \in [1, \infty]$. We let $\mathbb{B}_2^d(R)$ denote the $d$-dimensional $\ell_2$ ball of radius $R$. The Kullback-Leibler (KL) divergence between two distributions $P$ and $Q$ is denoted as $\mathsf{KL}(P\|Q) := \sum_x P(x) \log \frac{P(x)}{Q(x)}$.

## 2. Two-Player Zero-Sum Matrix Games

In this section, we start with a simple setting of two-player zero-sum matrix games, to develop our algorithmic framework.

### 2.1. Problem setting

**Two-player zero-sum matrix game.** We consider the (possibly KL-regularized) two-player zero-sum matrix games with the following objective:

$$\max_{\mu \in \Delta^m} \min_{\nu \in \Delta^n} f^{\mu, \nu}(A) \tag{1}$$
$$:= \mu^\top A \nu - \beta \mathsf{KL}(\mu\|\mu_{\mathsf{ref}}) + \beta \mathsf{KL}(\nu\|\nu_{\mathsf{ref}}),$$

where $A \in \mathbb{R}^{m \times n}$ is the payoff matrix, $\mu \in \Delta^m$ and $\mu_{\mathsf{ref}} \in \Delta^m$ (resp. $\nu \in \Delta^n$ and $\nu_{\mathsf{ref}} \in \Delta^n$) are the policy and reference policy for the max (resp. min) player,

and $\beta \geq 0$ is the regularization parameter.[3] Here, the reference policies can be used to incorporate prior knowledge or preference of the game; when the reference policies are uniform distributions, the KL regularization becomes entropy regularization, which are studied in, e.g., Cen et al. (2021).

**Nash equilibrium.** The policy pair $(\mu^\star, \nu^\star)$ corresponding to the solution to the saddle-point problem (1) represents a desirable state of the game, where both players perform their (regularized) best-response strategies against the other player, so that no players will unitarily deviate from its current policy. Specifically, the policy pair $(\mu^\star, \nu^\star)$ satisfies

$$\forall (\mu, \nu) \in \Delta^m \times \Delta^n : \quad f^{\mu, \nu^\star}(A) \leq f^{\mu^\star, \nu^\star}(A) \leq f^{\mu^\star, \nu}(A),$$

and is called the *Nash equilibrium* (NE) of the matrix game (Nash, 1950).[4]

**Noisy bandit feedback.** We are interested in learning the NE when the payoff matrix $A$ is unknown and can only be accessed through a stochastic oracle. Specifically, for any $i \in [m]$ and $j \in [n]$, we can query the entry $A(i, j)$, and receive a noisy feedback $\widehat{A}(i, j)$ of $A(i, j)$ from an oracle, i.e.,

$$\widehat{A}(i, j) = A(i, j) + \xi, \tag{2}$$

where the noise $\xi$ is an i.i.d. zero-mean random variable across different queries. Each of the collected data tuple is thus in the form of $(i, j, \widehat{A}(i, j))$.

**Goal: regret minimization.** Our goal is to design an easy-to-implement framework that can find the approximate NE of the matrix game (1) with as few queries as possible to the stochastic oracle in a sequential manner. To begin, we define the following

$$f^{\star, \nu}(A) := \max_{\mu \in \Delta^m} f^{\mu, \nu}(A), \quad f^{\mu, \star}(A) := \min_{\nu \in \Delta^n} f^{\mu, \nu}(A),$$

$$\text{and} \quad f^\star(A) := \max_{\mu \in \Delta^m} \min_{\nu \in \Delta^n} f^{\mu, \nu}(A) \tag{3}$$

for any payoff matrix $A$. The duality gap of the matrix game (1) at a policy pair $(\mu, \nu)$ is defined as

$$\mathsf{DualGap}\,(\mu, \nu) := f^{\star, \nu}(A) - f^{\mu, \star}(A), \tag{4}$$

where it is evident that $\mathsf{DualGap}\,(\mu^\star, \nu^\star) = 0$. A policy pair $(\mu, \nu)$ is called an $\varepsilon$-approximate NE (abbreviated as $\epsilon$-NE) of the matrix game (1) if $\mathsf{DualGap}\,(\mu, \nu) \leq \varepsilon$.

In an online setting, given a sequence of policy updates $\{(\mu_t, \nu_t)\}_{t=1,\dots,T}$ over $T$ rounds, a common performance

---

[3]For simplicity, we set the same regularization parameter for both players; our analysis continues to hold with different regularization parameters $\beta_1$ and $\beta_2$ for each player.

[4]We note that under entropy regularization, the equilibrium is also known as the *quantal response equilibrium* (QRE) (McKelvey & Palfrey, 1995) when $\beta > 0$.

metric is the cumulative regret, defined as

$$\mathsf{Regret}(T) := \sum_{t=1}^{T} \mathsf{DualGap}\,(\mu_t, \nu_t)$$

$$= \underbrace{\sum_{t=1}^{T} (f^{\star, \nu_t}(A) - f^\star(A))}_{\text{regret for min-player}} + \underbrace{\sum_{t=1}^{T} (f^\star(A) - f^{\mu_t, \star}(A))}_{\text{regret for max-player}},$$

$$\tag{5}$$

which encapsulates the regret from both players. Our goal is to achieve a sublinear, and ideally near-optimal, regret with respect to the number of rounds $T$, by carefully balancing the trade-off between exploration and exploitation, even under function approximation of the model class.

## 2.2. Algorithm development

We propose a model-based approach, called VMG, that enables provably efficient exploration-exploitation trade-off via resorting to a carefully-regularized model (i.e., the payoff matrix) estimator without constructing uncertainty intervals. To enable function approximation, we parameterize the payoff matrix by $A_\omega \in \mathbb{R}^{m \times n}$, where $\omega \in \Omega \subset \mathbb{R}^d$ is some vector in the parameter space $\Omega$.

The proposed approach, on a high level, alternates between updating the payoff matrix based on all the samples collected so far, and collecting new samples using the updated policies. Let's elaborate a bit further. At each round $t$, let the current payoff matrix estimate be $A_{\omega_{t-1}}$, and its corresponding NE be $(\mu_t, \nu_t)$.

- *Value-incentivized model updates.* Given all the collected data tuples $\mathcal{D}_{t-1}$ and the policy pair $(\mu_t, \nu_t)$, VMG updates the model parameter $\omega_t$ via solving a regularized least-squares estimation problem as (7), favoring models that *minimizes* the squared loss between the model and the noisy feedback stored in $\mathcal{D}_{t-1}$, and *maximizes* the value of each player when the other player's strategy is fixed. In other words, the regularization term aims to maximize the duality gap at $(\mu_t, \nu_t)$, which tries to pull the model away from its current estimate $A_{t-1}$, whose duality gap is 0 at $(\mu_t, \nu_t)$. The regularized estimator thus strikes a balance of exploitation (via least-squares on $\mathcal{D}_{t-1}$) and exploration (via regularization against the current model $A_{\omega_{t-1}}$).

- *Data collection from best-response policy updates.* Using the updated payoff matrix $A_{\omega_t}$, VMG updates the best-response policy of each player while fixing the policy of the other player via (8), resulting in policy pairs $(\widetilde{\mu}_t, \nu_t)$ and $(\mu_t, \widetilde{\nu}_t)$. Finally, VMG collects one new sample from each of the policy pairs respectively following the oracle (2), and add them to the dataset $\mathcal{D}_{t-1}$ to form $\mathcal{D}_t$.

The complete procedure of VMG is summarized in Algorithm 1. VMG invokes the mechanism of regularization as a means for incentivizing exploration, rendering it more amenable to implement in the presence of function approximation. In contrast, prior approach (O'Donoghue et al., 2021) heavily relies on explicitly adding an exploration bonus to the estimate of the payoff matrix using confidence intervals, which is challenging to construct under general function approximation. In addition, VMG allows parallel and independent policy execution from both players.

---

**Algorithm 1** Value-incentivized Online Matrix Game (VMG)

---

1: **Input:** initial parameter $\omega_0$, regularization coefficient $\alpha > 0$, iteration number $T$.
2: **Initialization:** dataset $\mathcal{D}_0 := \emptyset$.
3: **for** $t = 1, \cdots, T$ **do**
4:    Compute the Nash equilibrium $(\mu_t, \nu_t)$ of the matrix game with the current parameter $\omega_{t-1}$:

$$\mu_t = \arg\max_{\mu \in \Delta^m} \min_{\nu \in \Delta^n} f^{\mu,\nu}(A_{\omega_{t-1}}),$$

$$\nu_t = \arg\min_{\nu \in \Delta^n} \max_{\mu \in \Delta^m} f^{\mu,\nu}(A_{\omega_{t-1}}). \quad (6)$$

5:    Model update: Update the parameter $\omega_t$ by minimizing the following objective:

$$\omega_t = \arg\min_{\omega \in \Omega} \sum_{(i,j,\widehat{A}(i,j)) \in \mathcal{D}_{t-1}} \left(A_\omega(i,j) - \widehat{A}(i,j)\right)^2$$
$$\underbrace{-\alpha f^{\star,\nu_t}(A_\omega) + \alpha f^{\mu_t,\star}(A_\omega)}_{\text{value-incentivized reg.}}. \quad (7)$$

6:    Compute $\widetilde{\mu}_t$ and $\widetilde{\nu}_t$ by solving the following optimization problems:

$$\widetilde{\mu}_t = \arg\max_{\mu \in \Delta^m} f^{\mu,\nu_t}(A_{\omega_t}),$$

$$\widetilde{\nu}_t = \arg\min_{\nu \in \Delta^n} f^{\mu_t,\nu}(A_{\omega_t}). \quad (8)$$

7:    Data collection: Sample $(i_t, j_t) \sim (\widetilde{\mu}_t, \nu_t)$ and $(i'_t, j'_t) \sim (\mu_t, \widetilde{\nu}_t)$ and get the noisy feedback $\widehat{A}(i_t, j_t)$ and $\widehat{A}(i'_t, j'_t)$ following the oracle (2). Update the dataset $\mathcal{D}_t = \mathcal{D}_{t-1} \cup \left\{(i_t, j_t, \widehat{A}(i_t, j_t)), (i'_t, j'_t, \widehat{A}(i'_t, j'_t))\right\}$.
8: **end for**

---

**The benefit of regularization.** While VMG is agnostic to the power of KL regularization in (1), the major benefit of regularization comes in terms of computational efficiency. When the KL regularization parameter $\beta > 0$, common first-order game solvers such as mirror descent ascent (Sokota et al., 2022) or policy extragradient (Cen et al., 2021; 2023) methods achieve a last-iterate linear convergence rate when

solving the matrix game (6). Turning to the model update, when $\beta > 0$, the regularization term in (7) can be computed in closed form:

$$-f^{\star,\nu_t}(A_\omega) + f^{\mu_t,\star}(A_\omega)$$
$$= -\beta \left[ \log\left(\sum_{i=1}^n \mu_{\text{ref},i} \exp\left(\frac{A_\omega(i,:)\nu_t}{\beta}\right)\right) \right.$$
$$\left. + \log\left(\sum_{j=1}^m \nu_{\text{ref},j} \exp\left(-\frac{\mu_t^\top A_\omega(:,j)}{\beta}\right)\right) \right] + C, \quad (9)$$

where $\mu_{\text{ref},i}$ (resp. $\nu_{\text{ref},j}$) is the $i$-th (resp. $j$-th) entry of $\mu_{\text{ref}}$ (resp. $\nu_{\text{ref}}$), $A_\omega(i,:)$ (resp. $A_\omega(:,j)$) is the $i$-th row (resp. $j$-th column) of $A_\omega$, and $C$ is a constant that does not depend on $A_\omega$. Leveraging the closed-form expression, one can bypass solving a bi-level optimization problem (7) on its surface, but resorts to more efficient first-order methods. Last but not least, the policies $\widetilde{\mu}_t$ and $\widetilde{\nu}_t$ in (8) can be computed in closed form as well:

$$\widetilde{\mu}_{t,i} \propto \mu_{\text{ref},i} \exp\left(\frac{A_{\omega_t}(i,:)\nu_t}{\beta}\right), \quad (10)$$

$$\widetilde{\nu}_{t,j} \propto \nu_{\text{ref},j} \exp\left(-\frac{\mu_t^\top A_{\omega_t}(:,j)}{\beta}\right), \quad \forall i \in [m], j \in [n].$$

**The case of symmetric payoff.** One important special class of matrix games is the symmetric matrix game (Cheng et al., 2004), with $A = -A^\top$, $\mu_{\text{ref}} = \nu_{\text{ref}}$, and $m = n$. Many well-known games are symmetric, from classic games like rock-paper-scissors to the recent example of LLM alignment (Munos et al., 2023; Swamy et al., 2024; Yang et al., 2024b). For a symmetric matrix game, it admits a symmetric Nash $(\mu^\star, \mu^\star)$, and Algorithm 1 reduces to a single-player algorithm by only tracking a single policy $\mu_t$, recognizing $\mu_t = \nu_t$ and $\widetilde{\mu}_t = \widetilde{\nu}_t$ due to $f^{\mu,\nu}(A) = -f^{\nu,\mu}(A)$. In addition, VMG only needs to collect one sample from the policy pair $(\widetilde{\mu}_t, \mu_t)$ in each iteration. This is particularly desirable when the policy is expensive to store and update, such as large-scale neural networks or LLMs.

**Reduction to the bandit case.** By setting the action space of the min player to $n = 1$, VMG seamlessly reduces to the bandit setting, where the payoff matrix becomes a reward vector $A \in \mathbb{R}^m$. Here, we let $f^\mu(A) = \mu^\top A - \beta \text{KL}(\mu\|\mu_{\text{ref}})$ and $f^\star(A) := \max_{\mu \in \Delta^m} f^\mu(A)$. Interestingly, to encourage exploration, the regularization term favors a reward estimate that maximizes its regret $f^\star(A_\omega) - f^{\mu_t}(A_\omega)$ on the current policy $\mu_t$, which is *different from* the reward-biasing framework that only regularizes against $f^\star(A_\omega)$ (Cen et al., 2024; Liu et al., 2020; Xie et al., 2024).

## 2.3. Theoretical guarantee

We demonstrate that VMG achieves near-optimal regret, assuming linear function approximation of the payoff matrix. Specifically, we have the following assumption.

**Assumption 2.1** (Linear function approximation). The payoff matrix is parameterized as

$$A_\omega(i,j) \coloneqq \phi(i,j)^\top \omega, \quad \forall i \in [m], j \in [n], \qquad (11)$$

where $\omega \in \Omega \subset \mathbb{R}^d$ is the parameter vector and $\phi(i,j) \in \mathbb{R}^d$ is the feature vector for the $(i,j)$-th entry. Here, the feature vectors are known and fixed, and satisfy $\|\phi(i,j)\|_2 \leq 1$ for all $i \in [m]$ and $j \in [n]$. For all $\omega \in \Omega$, we suppose $\|\omega\|_2 \leq \sqrt{d}$ and $\|A_\omega\|_\infty \leq B_l$ for some $B_l > 0$.

We also assume that the linear function class is expressive enough to describe the true payoff matrix $A$.

**Assumption 2.2** (realizability). There exists $\omega^\star \in \Omega$ such that $A_{\omega^\star} = A$.

Next, we impose the noise follows standard sub-Gaussian distribution.

**Assumption 2.3** (i.i.d. sub-Gaussian noise). The noise $\xi$ in (2) are i.i.d. mean-zero sub-Gaussian random variables with sub-Gaussian parameter $\sigma > 0$.

**Regret guarantee.** The following theorem states the regret bound of VMG under appropriate choice of the regularization parameter.

**Theorem 2.4.** *Suppose Assumptions 2.1, 2.2 and 2.3 hold. Let $\delta \in (0, 1)$, setting the regularization coefficient $\alpha$ as*

$$\alpha = \sqrt{\frac{T}{d \log\left(1 + (T/d)^{3/2}\right)} \left(\log(4T/\delta) + d\log(dT)\right)},$$
$$(12)$$

*then for any $\beta \geq 0$, with any initial parameter $\omega_0$ and reference policies $\mu_{\mathsf{ref}}$ and $\nu_{\mathsf{ref}}$, we have with probability at least $1 - \delta$,*

$$\mathsf{Regret}(T) = \mathcal{O}\left(B_l\left(B_l + \sigma\sqrt{2\log(8T/\delta)}\right)d\sqrt{T\log(dT)}\right)$$
$$(13)$$

*for all $T \in \mathbb{N}_+$.*

The proof of Theorem 2.4 is deferred to Appendix B.2. Theorem 2.4 establishes that by setting $\alpha$ on the order of $\widetilde{O}(\sqrt{T})$, with high probability, the regret of VMG is no larger than an order of

$$\widetilde{\mathcal{O}}\left(d\sqrt{T}\right),$$

assuming the payoff matrix and the noise $\sigma$ are well-bounded. In particular, when reduced to the linear bandit

setting, this matches with the lower bound $\Omega(d\sqrt{T})$ established in Dani et al. (2008), suggesting the near-optimality of our result. In addition, since $\min_{t \in [T]} \mathsf{DualGap}(\mu_t, \nu_t) \leq \mathsf{Regret}(T)/T$, VMG is guaranteed to find an $\varepsilon$-NE of the matrix game (1) for any $\varepsilon > 0$ within $\widetilde{\mathcal{O}}\left(d^2/\varepsilon^2\right)$ samples.

# 3. Multi-player General-sum Markov Games

We now turn to the more challenging setting of online multi-player general-sum Markov games, which includes the two-player zero-sum Markov game as a special case.

## 3.1. Problem setting

**Multi-player general-sum Markov game.** We consider an $N$-player general-sum episodic Markov game with a finite horizon denoted as $\mathcal{M}_\mathbb{P} \coloneqq (\mathcal{S}, \mathcal{A}, \mathbb{P}, r, H)$, where $\mathcal{S}$ is the state space, $\mathcal{A} \coloneqq \mathcal{A}_1 \times \cdots \times \mathcal{A}_N \coloneqq \prod_{n=1}^N \mathcal{A}_n$ is the joint action space for all players, with $\mathcal{A}_n$ the action space of player $n$, and $H \in \mathbb{N}_+$ is the horizon length. Let $\Delta(\mathcal{S})$ and $\Delta(\mathcal{A})$ denote the set of probability distributions over $\mathcal{S}$ and $\mathcal{A}$, respectively. $\mathbb{P} = \{\mathbb{P}_h\}_{h \in [H]}$ with $\mathbb{P}_h : \mathcal{S} \times \mathcal{A} \to \Delta(\mathcal{S})$ is the inhomogeneous transition kernel: at step $h$, the probability of transitioning from state $s$ to state $s'$ by the action $\boldsymbol{a} = (a^1, \cdots, a^n)$ is $\mathbb{P}_h(s'|s, \boldsymbol{a})$. $r = \{r_h^n\}_{h \in [H], n \in [N]}$ stands for the reward function with $r_h^n : \mathcal{S} \times \mathcal{A} \to [0, 1]$ the reward of the $n$-th player at step $h$.

**Markov policies.** In this paper, we focus on the class of Markov policies, where the policy of each player depends only on the current state, without dependence on the history. We let $\pi^n : \mathcal{S} \times [H] \to \Delta(\mathcal{A})$ denote the policy of player $n$, and $\pi_h^n(\cdot|s) \in \Delta(\mathcal{A}_n)$ denotes the probability distribution of the action of player $n$ at step $h$ given any state $s$. We let $\boldsymbol{\pi} = (\pi^1, \cdots, \pi^N) : \mathcal{S} \times [H] \to \Delta(\mathcal{A})$ denote the joint Markov policy (we assume all policies appear in this paper are Markovian, and we let *joint policy* stands for joint Markov policy), where $\boldsymbol{\pi}_h(\cdot|s) \coloneqq (\pi_h^1, \cdots, \pi_h^N)(\cdot|s) \in \Delta(\mathcal{A})$ for all $s \in \mathcal{S}$ and $h \in [H]$. For any joint policy $\boldsymbol{\pi}$, we let $\boldsymbol{\pi}^{-n}$ denote the joint policy excluding player $n$. With a slight abuse of notation, we write $\boldsymbol{\pi} = (\pi^n, \boldsymbol{\pi}^{-n})$. In addition, a joint policy $\boldsymbol{\pi}$ is called a *product policy* if $\pi^1, \cdots, \pi^N$ are executed independently, i.e., under policy $\boldsymbol{\pi}$, each player takes actions independently. We denote $\boldsymbol{\pi} = \pi^1 \times \cdots \times \pi^N$ for a product policy.

**KL-regularized value function and Q-function.** Given a joint policy $\boldsymbol{\pi}$, the KL-regularized state-value function (*value function*) $V_{h,n}^{\boldsymbol{\pi}} : \mathcal{S} \to \mathbb{R}$ and the KL-regularized state-action value function (*Q-function*) $Q_{h,n}^{\boldsymbol{\pi}} : \mathcal{S} \times \mathcal{A} \to \mathbb{R}$ of the $n$-th player under $\boldsymbol{\pi}$ — with regularization parameter

$\beta \geq 0$ — are respectively defined as

$$V_{h,n}^{\boldsymbol{\pi}}(s) := \mathbb{E}\left[\sum_{i=h}^{H} r_i^n(s_i, \boldsymbol{a}_i) - \beta \log \frac{\pi^n(a_i^n|s_i)}{\pi_{\mathsf{ref}}^n(a_i^n|s_i)}\bigg| s_h = s\right],$$
(14a)

$$Q_{h,n}^{\boldsymbol{\pi}}(s, \boldsymbol{a}) := r_h^n(s, \boldsymbol{a}) + \mathbb{E}_{s' \sim \mathbb{P}_h(\cdot|s,\boldsymbol{a})}\left[V_{h+1,n}^{\boldsymbol{\pi}}(s')\right],$$
(14b)

for all $(s, \boldsymbol{a}) \in \mathcal{S} \times \mathcal{A}, h \in [H]$ where $s_i$ and $\boldsymbol{a}_i$ are the state and action at step $i$, respectively, and $\boldsymbol{\pi}_{\mathsf{ref}} : \mathcal{S} \times [H] \to \Delta(\mathcal{A})$ is the reference policy. When the reference policy is a uniform distribution over the joint action space, the regularization becomes the entropy regularization. In (14a), $\pi^n(\cdot|s)$ (resp. $\pi_{\mathsf{ref}}^n(\cdot|s)$) should be understood as the marginal distribution of player $n$ under joint distribution $\boldsymbol{\pi}(\cdot|s)$ (resp. $\boldsymbol{\pi}_{\mathsf{ref}}(\cdot|s)$), and we define $V_{H+1,n}^{\boldsymbol{\pi}}(s) = 0$ for all $s \in \mathcal{S}$ and $\beta \geq 0$. To simplify the notation, we define $V_n^{\boldsymbol{\pi}} := V_{1,n}^{\boldsymbol{\pi}}$ and $Q_n^{\boldsymbol{\pi}} := Q_{1,n}^{\boldsymbol{\pi}}$ for all $n \in [N]$. We assume $\rho \in \Delta(\mathcal{S})$ is the initial state distribution, i.e., $s_1 \sim \rho$. Furthermore, we define $V_n^{\boldsymbol{\pi}}(\rho) := \mathbb{E}_{s \sim \rho}[V_n^{\boldsymbol{\pi}}(s)]$.

We let $\boldsymbol{\pi} = \pi^n \times \boldsymbol{\pi}^{-n}$ denote the policy profile where all players but the $n$-th player execute policy $\boldsymbol{\pi}^{-n}$, and the $n$-th player executes policy $\pi^n$ independent of the other players. For all $n \in [N]$, we define the best-response value function

$$V_{h,n}^{\star,\boldsymbol{\pi}^{-n}}(s) := \max_{\pi^n:\mathcal{S}\times[H]\to\Delta(\mathcal{A}_n)} V_{h,n}^{\pi^n \times \boldsymbol{\pi}^{-n}}(s),$$
(15)

for all $s \in \mathcal{S}, h \in [H], n \in [N]$, which is the optimal value of player $n$ when the policies of other agents are fixed by $\boldsymbol{\pi}^{-n}$. Importantly, there exists at least one policy $\pi^{n,\star}(\boldsymbol{\pi}^{-n})$ that achieves the maximum in (15) for all $s \in \mathcal{S}$, and this policy is referred to the *best-response policy* of player $n$ under joint policy $\boldsymbol{\pi}^{-n}$ (Shapley, 1953). We also define

$$V_n^{\star,\boldsymbol{\pi}^{-n}}(\rho) := \max_{\pi^n:\mathcal{S}\times[H]\to\Delta(\mathcal{A}_n)} V_n^{\pi^n \times \boldsymbol{\pi}^{-n}}(\rho).$$

One important thing to notice is that the best-response policy $\pi^{n,\star}(\boldsymbol{\pi}^{-n})$ does not depend on the initial state distribution $\rho$ (Mei et al., 2020).

**Equilibria of Markov games.** In a multi-player general-sum Markov game, each agent aims to maximize its own value function, where the Nash equilibrium (NE) (Nash, 1950) and the coarse correlated equilibrium (CCE) (Aumann, 1987) are two widely studied solution concepts, whose definitions are as follows.

- *Nash equilibrium (NE):* a product policy $\boldsymbol{\pi} = \pi^1 \times \cdots \times \pi^N$ is a Nash equilibrium of $\mathcal{M}_{\mathbb{P}}$ if

$$\forall s \in \mathcal{S}, n \in [N]: \quad V_n^{\star,\boldsymbol{\pi}^{-n}}(s) = V_n^{\pi^n,\boldsymbol{\pi}^{-n}}(s).$$
(16)

- *Coarse correlated equilibrium (CCE):* a joint policy $\boldsymbol{\pi}$ is a CCE of $\mathcal{M}_{\mathbb{P}}$ if

$$\forall s \in \mathcal{S}, n \in [N]: \quad V_n^{\star,\boldsymbol{\pi}^{-n}}(s) \leq V_n^{\pi^n,\boldsymbol{\pi}^{-n}}(s).$$
(17)

It is obvious that every NE is a CCE, but the converse is not true in general. In general, computing the NE in general-sum Markov games is intractable (Daskalakis et al., 2009), except for two-player zero-sum Markov games.

**Goal: regret minimization.** To measure the proximity of a policy $\boldsymbol{\pi}$ to the equilibrium, we define the (average) sub-optimality gap of policy $\boldsymbol{\pi}$ w.r.t. the initial distribution $\rho$ as

$$\mathsf{Gap}(\boldsymbol{\pi}) := \frac{1}{N}\sum_{n=1}^{N}\left(V_n^{\star,\boldsymbol{\pi}^{-n}}(\rho) - V_n^{\boldsymbol{\pi}}(\rho)\right).$$
(18)

A *product* policy $\boldsymbol{\pi}$ is said to be an $\varepsilon$-approximate NE (abbreviated as $\varepsilon$-NE) if $\mathsf{Gap}(\boldsymbol{\pi}) \leq \varepsilon$, and a *joint* policy $\boldsymbol{\pi}$ is said to be an $\varepsilon$-approximate CCE (abbreviated as $\varepsilon$-CCE) if $\mathsf{Gap}(\boldsymbol{\pi}) \leq \varepsilon$.

We aim to design a model-based framework that find the approximate NE or CCE of the Markov game $\mathcal{M}_{\mathbb{P}}$ in a provably efficient manner. Similar to the matrix game setting, we consider the following regret measure:

$$\mathsf{Regret}(T) := \sum_{t=1}^{T}\mathsf{Gap}(\boldsymbol{\pi}_t)$$
$$= \sum_{t=1}^{T}\frac{1}{N}\sum_{n=1}^{N}\left(V_n^{\star,\boldsymbol{\pi}_t^{-n}}(\rho) - V_n^{\boldsymbol{\pi}_t}(\rho)\right), \quad (19)$$

where $\boldsymbol{\pi}_t$ is the policy profile at time $t$. Our goal is to achieve a sublinear regret with respect to the number of rounds $T$, by carefully balancing the trade-off between exploration and exploitation, even under function approximation of the model class.

### 3.2. Algorithm development

For simplicity, we will focus on the function approximation over the transition kernel of the Markov game assuming the reward function is fixed and deterministic, while it is straightforwardly to also incorporate the reward function approximation. We let $\mathcal{F}$ denote the function class of the estimators of the transition kernel of the Markov game, and we denote the parameterized transition kernel as

$$\mathbb{P}_f = (\mathbb{P}_{f,1}, \cdots, \mathbb{P}_{f,H}) \in \mathcal{F} = \mathcal{F}_1 \times \cdots \mathcal{F}_H,$$

where $\mathcal{F}$ is the function class and $f$ is its parameterization. We define the value function $V_{f,h,n}^{\boldsymbol{\pi}}$ under the transition kernel $\mathbb{P}_f$ as

$$\forall s \in \mathcal{S}, h \in [H] \quad V_{f,h,n}^{\boldsymbol{\pi}}(s) := \quad (20)$$

$$\mathbb{E}_{\mathbb{P}_f, \boldsymbol{\pi}} \left[ \sum_{i=h}^{H} \left( r_i^n(s_i, \boldsymbol{a}_i) - \beta \log \frac{\pi^n(a_i^n | s_i)}{\pi_{\mathsf{ref}}^n(a_i^n | s_i)} \right) \middle| s_h = s \right],$$

and the Q-function $Q_{f,h,n}^{\boldsymbol{\pi}}$ is defined likewise.

Akin to the matrix game case, VMG alternates between updating the model updates based on all the transitions observed so far, and collecting new trajectories using the updated policies. Suppose that at the $t$-th iteration, the current estimate of the transition kernel is $\mathbb{P}_{f_{t-1}}$, and its corresponding NE or CCE is $\boldsymbol{\pi}_t$. VMG alternates between the following two steps.

- *Value-incentivized model updates.* Given all the collected transitions $\mathcal{D}_{t-1,h}$ at each step $h$ and the equilibrium policy $\boldsymbol{\pi}_t$, VMG updates the model parameter $f_t$ via solving a regularized maximum likelihood estimation (MLE) problem as (22), favoring models that *minimizes* the negative log-likelihood $\mathcal{L}_t(f)$ of the model, i.e.

$$\mathcal{L}_t(f) := \sum_{h=1}^{H} \sum_{(s_h, \boldsymbol{a}_h, s_{h+1}) \in \mathcal{D}_{t-1,h}} - \log \mathbb{P}_{f,h}(s_{h+1} | s_h, \boldsymbol{a}_h), \tag{21}$$

and *maximizes* the sum of the *best-response* values of each player when the other player's strategy is fixed at $\boldsymbol{\pi}_t^{-n}$. In words, the regularizer tries to encourage models that incentive the players to deviate from their current policy, resulting in better exploration.

- *Trajectory collection from best-response policy updates.* Using the updated model $\mathbb{P}_{f_t}$, VMG updates the best-response policy $\widetilde{\pi}_t^n$ of each player while fixing the policy $\boldsymbol{\pi}_t^{-n}$ of the other player via (23). VMG then collects new trajectories by following policy $\boldsymbol{\pi}_t$ and $(\widetilde{\pi}_t^n, \boldsymbol{\pi}_t^{-n})$ for all $n \in [N]$, and update the dataset.

The complete procedure of VMG is summarized in Algorithm 2, where the function $\mathsf{Equilibrium}(\mathcal{M}_f)$ returns the NE or CCE of the Markov game $\mathcal{M}_f$ by calling off-the-shelf solvers, e.g., Cai et al. (2024); Zhang et al. (2022). Note that we are primarily interested in finding the NE for two-player zero-sum Markov games, and the CCE for multi-player general-sum Markov games, due to computational tractability.

**Comparison with MEX.** Liu et al. (2024, Algorithm 2) proposed the MEX framework, which also considered using value functions as a means to incentive exploration for two-player zero-sum Markov games. Their algorithm requires asymmetric updates — and two sets of model parameters as a result — of the max and min players, where the model update of the max player is regularized by the optimal value $V_f^\star = \max_{\pi^1} \min_{\pi^2} V_{1,1}^{\boldsymbol{\pi}}(\rho)$ of the Markov game, which is an expensive saddle-point optimization problem, and the

model update of the min player is regularized by the best-response value function. In contrast, VMG only leverages best-response value functions as a regularization, which is much easier to solve. VMG also permits simultaneous updates for all the players, making it amenable to multi-player general-sum Markov games. In contrast, the extension of MEX (Xiong et al., 2024) to this more general setting is significantly more involved with a computational complexity that scales exponentially with the number of agents.

---

**Algorithm 2** Value-incentivized Online Markov Game (VMG)

---

1: **Input:** reference policies $\boldsymbol{\pi}_{\mathsf{ref}}$, initial transition kernel estimate $f_0 \in \mathcal{F}$, regularization coefficient $\alpha > 0$, iteration number $T$.
2: **Initialization:** dataset $\mathcal{D}_{0,h} := \emptyset, \forall h \in [H]$.
3: **for** $t = 1, \cdots, T$ **do**
4:      $\boldsymbol{\pi}_t \leftarrow \mathsf{Equilibrium}(\mathcal{M}_{f_{t-1}})$.      ▷ $\mathsf{Equilibrium}(\mathcal{M}_f)$ returns a CCE or NE of game $\mathcal{M}_f$.
5:      Model update: Update the estimator $f_t$ by minimizing the following objective:

$$f_t = \underset{f \in \mathcal{F}}{\arg\min} \; \mathcal{L}_t(f) - \alpha \sum_{n=1}^{N} V_{f,n}^{\star, \boldsymbol{\pi}_t^{-n}}(\rho). \tag{22}$$

6:      Compute best-response policies $\{\widetilde{\pi}_t^n\}_{n \in [N]}$:

$$\forall n \in [N]: \quad \widetilde{\pi}_t^n = \underset{\pi^n : \mathcal{S} \times [H] \to \Delta(\mathcal{A}_n)}{\arg\max} V_{f_t,n}^{\pi^n, \boldsymbol{\pi}_t^{-n}}(\rho). \tag{23}$$

7:      Data collection: sample a trajectory with transition tuples $\{(s_{t,h}, \boldsymbol{a}_{t,h}, s_{t,h+1})\}_{h=1}^{H}$ by executing $\boldsymbol{\pi}_t$, and sample a trajectory with transition tuples $\{(s_{t,h}^n, \boldsymbol{a}_{t,h}^n, s_{t,h+1}^n)\}_{h=1}^{H}$ by executing $(\widetilde{\pi}_t^n, \boldsymbol{\pi}_t^{-n})$ for each $n \in [N]$. Update the dataset $\mathcal{D}_{t,h} = \mathcal{D}_{t-1,h} \cup_{n=1}^{N} \{(s_{t,h}, \boldsymbol{a}_{t,h}, s_{t,h+1}), (s_{t,h}^n, \boldsymbol{a}_{t,h}^n, s_{t,h+1}^n)\}$, $\forall h \in [H]$.
8: **end for**

---

**Reduction to the single-agent MDP case.** VMG can be reduced to the Markov decision process (MDP) setting via either setting the number of players $N = 1$ in the multi-player general-sum Markov game, or setting the action space of the min player to a singleton in the two-player zero-sum Markov game. Interestingly, the former leads to the value regularization $V_f^\star(\rho)$ studied in MEX (Liu et al., 2024), while the latter leads to a new form of regularizer $V_f^\star(\rho) - V_f^{\pi_t}(\rho)$, adding friction from the current policy $\pi_t$.

## 3.3. Theoretical guarantee

We demonstrate that VMG achieves near-optimal regret under the following linear mixture model of the transition kernel for Markov games.

**Assumption 3.1** (linear mixture model). *The function class $\mathcal{F} = \mathcal{F}_1 \times \cdots \mathcal{F}_H$ is*

$$\forall h \in [H]: \quad \mathcal{F}_h := \Big\{ f_h | f_h(s'|s, \boldsymbol{a}) = \phi_h(s, \boldsymbol{a}, s')^\top \theta_h,$$
$$\forall (s, \boldsymbol{a}, s') \in \mathcal{S} \times \mathcal{A} \times \mathcal{S}, \theta_h \in \Theta_h \Big\},$$

*where $\phi_h = (\phi_h^1, \cdots, \phi_h^d) : \mathcal{S} \times \mathcal{A} \times \mathcal{S} \to \mathbb{R}^d$ are the known feature maps with $\phi_h^i : \mathcal{S} \times \mathcal{A} \to \Delta(\mathcal{S})$ being transition kernels for all $i \in [d]$. $\|\phi_h(s, \boldsymbol{a}, s')\|_2 \leq 1$ for all $(s, \boldsymbol{a}, s')$, and $\Theta_h \subseteq \mathbb{B}_2^d(\sqrt{d})$, $\forall h \in [H]$. For each $f_h \in \mathcal{F}_h$ and $(s, \boldsymbol{a}) \in \mathcal{S} \times \mathcal{A}$, $f_h(\cdot|s, \boldsymbol{a}) \in \Delta(\mathcal{S})$, $\forall h \in [H]$.*

The linear mixture model is a common assumption in the RL literature, see, for example, Ayoub et al. (2020); Modi et al. (2020); Cai et al. (2020) for single-agent RL, and Chen et al. (2022); Liu et al. (2024) for Markov games. We also assume the function class $\mathcal{F}$ is expressive enough to describe the true transition kernel of the Markov game.

**Assumption 3.2** (realizability). *There exists $f^\star \in \mathcal{F}$ such that $\mathbb{P}_{f^\star} = \mathbb{P}$.*

**Regret guarantee.** We now present our main result for the regret of the online Markov game, whose proof is deferred to Appendix B.3.

**Theorem 3.3.** *Under Assumptions 3.1 and 3.2, if setting the regularization coefficient $\alpha$ as*

$$\alpha = \sqrt{\frac{T}{Hd \log\left(1 + \frac{T^{3/2}H^2}{\sqrt{d}}\right)} \left(\log\left(\frac{HN}{\delta}\right) + d \log\left(d|\mathcal{S}|T\right)\right)},$$

*then for any $\beta \geq 0$, with any initial state distribution $\rho$, transition kernel estimator $f_0 \in \mathcal{F}$ and reference policy $\pi_{\mathsf{ref}}$, the regret of Algorithm 2 satisfies the following bound with probability at least $1 - \delta$ for any $\delta \in (0, 1)$ and for all $T \in \mathbb{N}_+$ :*

$$\mathsf{Regret}(T) \leq \widetilde{\mathcal{O}} \left( d\sqrt{H^3 T} \cdot \sqrt{\frac{1}{d} \log\left(\frac{NH}{\delta}\right) + \log\left(d|\mathcal{S}|T\right)} \right). \tag{24}$$

Theorem 3.3 establishes that by setting $\alpha$ on the order of $\widetilde{O}(\sqrt{T/H})$, with high probability, the regret of VMG is no larger than an order of

$$\widetilde{\mathcal{O}}\left(d\sqrt{H^3 T}\right)$$

for general-sum Markov games, which improves the dependency on the horizon length $H$ compared with the regret

$\widetilde{\mathcal{O}}\left(dH^5\sqrt{T}\right)$ of (Xiong et al., 2024). When reducing to two-player zero-sum Markov games, our regret bound — established for both players — matches that of MEX (Liu et al., 2024), which only covers the max player.

In addition, since $\min_{t \in [T]} \mathsf{Gap}\left(\boldsymbol{\pi}_t\right) \leq \mathsf{Regret}(T)/T$ and each iteration collects $N + 1$ trajectories, VMG is guaranteed to find an $\varepsilon$-NE ($\varepsilon$-CCE) of $\mathcal{M}_{\mathbb{P}}$ for any $\varepsilon > 0$ within $\widetilde{\mathcal{O}}\left(\frac{Nd^2H^3}{\varepsilon^2}\right)$ trajectories or $\widetilde{\mathcal{O}}\left(\frac{Nd^2H^4}{\varepsilon^2}\right)$ samples. Compared to the minimax sample complexity (Chen et al., 2022), our sample complexity is near-optimal up to a factor of $H$ when the number of players $N$ is fixed.

## 4. Conclusion

In this paper, we introduced VMG, a provably-efficient model-based algorithm for online MARL that balances exploration and exploitation without requiring explicit uncertainty quantification. The key innovation lies in incentivizing the model estimation to maximize the best-response value functions across all players to implicitly drive exploration. In addition, VMG is readily compatible with modern deep reinforcement learning architectures using function approximation, and is demonstrated to achieve a near-optimal regret under linear function approximation of the model class. We believe this work takes an important step toward making MARL more practical and scalable for real-world applications.

Several promising directions remain for future work. For example, designing a model-free counterpart of VMG that can be used in conjunction with function approximation could be a valuable extension. Additionally, it will be interesting to develop the performance guarantee of VMG under alternative assumptions of function approximation, such as general function approximation and independent function approximation across the players to tame the curse of dimensionality and multi-agency. Last but not least, it will be of interest to study the performance of VMG under adversarial environments.

## Acknowledgement

This work is supported in part by the grants NSF DMS-2134080, CCF-2106778, ONR N00014-19-1-2404, ONR N00014-25-1-2173, NSF ECCS-2401391, and NSF IIS-2403240.

## Impact Statement

This paper presents work whose goal is to advance the field of Machine Learning. There are many potential societal consequences of our work, none which we feel must be specifically highlighted here.

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

## A. Special Cases

**Symmetric matrix game.** One important special class of matrix games is the symmetric matrix game (Cheng et al., 2004), with $A = -A^\top$, $\mu_{\text{ref}} = \nu_{\text{ref}}$, and $m = n$. In this case, we assume the parameter space $\Omega$ preserves anti-symmetry of $A$, i.e., $A_\omega = -A_\omega^\top$ for any $\omega \in \Omega$. For a symmetric matrix game, it admits a symmetric Nash $(\mu^\star, \mu^\star)$, and Algorithm 1 reduces to a single-player algorithm by only tracking a single policy $\mu_t$, recognizing $\mu_t = \nu_t$ and $\widetilde{\mu}_t = \widetilde{\nu}_t$ due to $f^{\mu,\nu}(A) = -f^{\nu,\mu}(A)$. In addition, VMG only needs to collect one sample from the policy pair $(\widetilde{\mu}_t, \mu_t)$ in each iteration. Altogether, these lead to a simplified algorithm summarized in Algorithm 3.

---

**Algorithm 3** Value-incentivized Online Symmetric Matrix Game (VMG)

---

1: **Input:** initial parameter $\omega_0$, regularization coefficient $\alpha > 0$, iteration number $T$.

2: **Initialization:** dataset $\mathcal{D}_0 := \emptyset$.

3: **for** $t = 1, \cdots, T$ **do**

4:   Compute $\mu_t$ by solving the matrix game with the current parameter $\omega_{t-1}$:

$$\mu_t = \arg\max_{\mu \in \Delta^m} \min_{\nu \in \Delta^n} f^{\mu,\nu}(A_{\omega_{t-1}}). \tag{25}$$

5:   Model update: Update the parameter $\omega_t$ by minimizing the following objective:

$$\omega_t = \arg\min_{\omega \in \Omega} \sum_{(i,j,\widehat{A}(i,j)) \in \mathcal{D}_{t-1}} \left( A_\omega(i,j) - \widehat{A}(i,j) \right)^2 + \alpha f^{\mu_t,\star}(A_\omega). \tag{26}$$

6:   Compute $\widetilde{\mu}_t$ by solving the following optimization problem:

$$\widetilde{\mu}_t = \arg\max_{\mu \in \Delta^m} f^{\mu,\mu_t}(A_{\omega_t}). \tag{27}$$

7:   Data collection: Sample $(i_t, j_t) \sim (\widetilde{\mu}_t, \mu_t)$ and get the noisy feedback $\widehat{A}(i_t, j_t)$ following the oracle (2). Update the dataset $\mathcal{D}_t = \mathcal{D}_{t-1} \cup \left\{ (i_t, j_t, \widehat{A}(i_t, j_t)) \right\}$.

8: **end for**

---

**Bandit setting.** By setting $n = 1$, we can reduce the matrix game to the bandit setting, where the payoff matrix becomes a reward vector $A \in \mathbb{R}^m$, leading to a simplified algorithm in Algorithm 4. Here, we let $f^\mu(A) = \mu^\top A - \beta \text{KL}(\mu \| \mu_{\text{ref}})$ and $f^\star(A) := \max_{\mu \in \Delta^m} f^\mu(A)$. Interestingly, to encourage exploration, the regularization term favors a reward estimate that maximizes its regret $f^\star(A_\omega) - f^{\mu_t}(A_\omega)$ on the current policy $\mu_t$, which is *different from* the reward-biasing framework that only regularizes against $f^\star(A_\omega)$ (Cen et al., 2024; Liu et al., 2020; Xie et al., 2024).

**MDP setting.** VMG can be reduced to the single-agent setting via either setting the number of players $N = 1$ in the multi-player general-sum Markov game, or setting the action space of the min player to a singleton, i.e., $|\mathcal{A}_2| = 1$, in the two-player zero-sum Markov game. Interestingly, the former (option I) leads to the value regularization $V_f^\star(\rho)$ studied in MEX (c.f., Algorithm 1 in Liu et al. (2024)), while the latter (option II) leads to a new form of regularizer $V_f^\star(\rho) - V_f^{\pi_t}(\rho)$, adding friction from the current policy $\pi_t$. The latter regularizer is also the MDP counterpart of the bandit algorithm in Algorithm 4. We summarize both variants in Algorithm 5.

## B. Proofs of Main Theorems

### B.1. Auxiliary lemmas

We provide some technical lemmas that will be used in our proofs.

**Lemma B.1** (Freedman's inequality, Lemma D.2 in Liu et al. (2024)). *Let $\{X_t\}_{t \leq T}$ be a real-valued martingale difference sequence adapted to filtration $\{\mathcal{F}_t\}_{t \leq T}$. If $|X_t| \leq R$ almost surely, then for any $\eta \in (0, 1/R)$ it holds that with probability at least $1 - \delta$,*

$$\sum_{t=1}^T X_t \leq \mathcal{O}\left( \eta \sum_{t=1}^T \mathbb{E}[X_t^2 | \mathcal{F}_{t-1}] + \frac{\log(1/\delta)}{\eta} \right).$$

---

**Algorithm 4** Value-incentivized Online Bandit (VMG)

---

1: **Input:** initial parameter $\omega_0$, regularization coefficient $\alpha > 0$, iteration number $T$.
2: **Initialization:** dataset $\mathcal{D}_0 := \emptyset$.
3: **for** $t = 1, \cdots, T$ **do**
4:     Policy update: Compute $\mu_t$ with the current parameter $\omega_{t-1}$:

$$\mu_t = \arg\max_{\mu \in \Delta^m} f^\mu(A_{\omega_{t-1}}) \propto \mu_{\mathsf{ref}} \exp\left(\frac{A_{\omega_{t-1}}}{\beta}\right). \tag{28}$$

5:     Data collection: Sample $i_t \sim \mu_t$ and get the noisy feedback $\widehat{A}(i_t)$ following the oracle (2). Update the dataset $\mathcal{D}_t = \mathcal{D}_{t-1} \cup \left\{(i_t, \widehat{A}(i_t))\right\}$.
6:     Model update: Update the parameter $\omega_t$ by minimizing the following objective:

$$\omega_t = \arg\min_{\omega \in \Omega} \sum_{(i, \widehat{A}(i)) \in \mathcal{D}_t} \left(A_\omega(i) - \widehat{A}(i)\right)^2 - \alpha f^\star(A_\omega) + \alpha f^{\mu_t}(A_\omega). \tag{29}$$

7: **end for**

---

---

**Algorithm 5** Value-incentivized Online Single-agent MDP (VMG)

---

1: **Input:** initial transition kernel estimate $f_0 \in \mathcal{F}$, regularization coefficient $\alpha > 0$, iteration number $T$.
2: **Initialization:** dataset $\mathcal{D}_{0,h} := \emptyset$ for all $h \in [H]$.
3: **for** $t = 1, \cdots, T$ **do**
4:     Policy update: Compute $\pi_t$ with the current transition kernel estimator $f_{t-1}$:

$$\pi_t = \arg\max_{\pi \in \Delta(\mathcal{A}_1)} V_{f_{t-1}}^\pi(\rho). \tag{30}$$

5:     Data collection: sample a trajectory with transition tuples $\{(s_{t,h}, a_{t,h}, s_{t,h+1})\}_{h=1}^H$ following $\pi_t$. Update the dataset $\mathcal{D}_{t,h} = \mathcal{D}_{t-1,h} \cup \{(s_{t,h}, a_{t,h}, s_{t,h+1})\}$ for all $h \in [H]$.
6:     Model update: update the estimator $f_t$ by minimizing the following objective

$$f_t = \begin{cases} \arg\min_{f \in \mathcal{F}} \sum_{h=1}^H \sum_{(s_h, a_h, s_{h+1}) \in \mathcal{D}_{t,h}} -\log \mathbb{P}_{f,h}(s_{h+1}|s_h, a_h) - \alpha V_f^\star(\rho) & \text{(option I)} \\ \arg\min_{f \in \mathcal{F}} \sum_{h=1}^H \sum_{(s_h, a_h, s_{h+1}) \in \mathcal{D}_{t,h}} -\log \mathbb{P}_{f,h}(s_{h+1}|s_h, a_h) - \alpha V_f^\star(\rho) + \alpha V_f^{\pi_t}(\rho) & \text{(option II)} \end{cases} . \tag{31}$$

7: **end for**

---

**Lemma B.2** (Lemma 11 in Abbasi-Yadkori et al. (2011)). *Let $\{x_s\}_{s \in [T]}$ be a sequence of vectors with $x_s \in \mathcal{V}$ for some Hilbert space $\mathcal{V}$. Let $\Lambda_0$ be a positive definite matrix and define $\Lambda_t = \Lambda_0 + \sum_{s=1}^t x_s x_s^\top$. Then it holds that*

$$\sum_{s=1}^T \min\left\{1, \|x_s\|_{\Lambda_{s-1}^{-1}}\right\} \le 2\log\left(\frac{\det(\Lambda_T)}{\det(\Lambda_0)}\right).$$

**Lemma B.3** (Lemma F.3 in Du et al. (2021)). *Let $\mathcal{X} \subset \mathbb{R}^d$ and $\sup_{x \in \mathcal{X}} \|x\|_2 \le B_X$. Then for any $n \in \mathbb{N}_+$, we have*

$$\forall \lambda > 0: \quad \max_{x_1, \cdots, x_n \in \mathcal{X}} \log\det\left(I_d + \frac{1}{\lambda}\sum_{i=1}^n x_i x_i^\top\right) \le d\log\left(1 + \frac{nB_X^2}{d\lambda}\right).$$

**Lemma B.4** (Martingale exponential concentration, Lemma D.1 in Liu et al. (2024)). *Let $\delta \in (0, 1)$. For a sequence of real-valued random variables $\{X_t\}_{t \in [T]}$ adapted to filtration $\{\mathcal{F}_t\}_{t \in [T]}$, the following holds with probability at least $1 - \delta$:*

$$\forall t \in [T]: \quad -\sum_{s=1}^t X_s \le \sum_{s=1}^t \log \mathbb{E}[\exp(-X_s)|\mathcal{F}_{s-1}] + \log(1/\delta).$$

**Lemma B.5** (Covering number of $\ell_2$ ball, Lemma D.5 in Jin et al. (2020)). *For any $\epsilon > 0$ and $d \in \mathbb{N}_+$, the $\epsilon$-covering number of the $\ell_2$ ball of radius $R$ in $\mathbb{R}^d$ is bounded by $(1 + 2R/\epsilon)^d$.*

### B.2. Proof of Theorem 2.4

In the proof, for any sequence $\{x_i\}_{i\in\mathbb{Z}}$ and any integers $a, b \in \mathbb{Z}$ where $a > b$, we define $\sum_{i=a}^b x_i := 0$.

We begin by decomposing the regret as

$$
\begin{aligned}
\mathsf{Regret}(T) &= \sum_{t=1}^T f^{\star,\nu_t}(A) - f^{\mu_t,\star}(A) \\
&= \sum_{t=1}^T \left( f^{\star,\nu_t}(A) - f^{\mu_t,\star}(A) - (f^{\star,\nu_t}(A_{\omega_t}) - f^{\mu_t,\star}(A_{\omega_t})) \right) \\
&\quad + \sum_{t=1}^T \left( f^{\star,\nu_t}(A_{\omega_t}) - f^{\widetilde{\mu}_t,\nu_t}(A) \right) + \sum_{t=1}^T \left( f^{\mu_t,\widetilde{\nu}_t}(A) - f^{\mu_t,\star}(A_{\omega_t}) \right) \\
&\quad + \sum_{t=1}^T \left( f^{\widetilde{\mu}_t,\nu_t}(A) - f^{\widetilde{\mu}_t,\nu_t}(A_{\omega_{t-1}}) \right) + \sum_{t=1}^T \left( f^{\mu_t,\widetilde{\nu}_t}(A_{\omega_{t-1}}) - f^{\mu_t,\widetilde{\nu}_t}(A) \right) \\
&\quad + \sum_{t=1}^T \left( f^{\widetilde{\mu}_t,\nu_t}(A_{\omega_{t-1}}) - f^{\mu_t,\widetilde{\nu}_t}(A_{\omega_{t-1}}) \right).
\end{aligned}
\tag{32}
$$

Recall that $(\mu_t, \nu_t)$ is the Nash equilibrium of the matrix game with the pay-off matrix $A_{\omega_{t-1}}$ (see (6)), we have

$$
\forall t \in [T]: \quad f^{\widetilde{\mu}_t,\nu_t}(A_{\omega_{t-1}}) \leq f^{\mu_t,\nu_t}(A_{\omega_{t-1}}) \leq f^{\mu_t,\widetilde{\nu}_t}(A_{\omega_{t-1}}).
\tag{33}
$$

This implies the last term in the regret decomposition is non-positive, i.e.,

$$
\sum_{t=1}^T \left( f^{\widetilde{\mu}_t,\nu_t}(A_{\omega_{t-1}}) - f^{\mu_t,\widetilde{\nu}_t}(A_{\omega_{t-1}}) \right) \leq 0.
\tag{34}
$$

Moreover, by the definition of $\widetilde{\mu}_t$ and $\widetilde{\nu}_t$ in (8), we have

$$
f^{\star,\nu_t}(A_{\omega_t}) = f^{\widetilde{\mu}_t,\nu_t}(A_{\omega_t}) \quad \text{and} \quad f^{\mu_t,\star}(A_{\omega_t}) = f^{\mu_t,\widetilde{\nu}_t}(A_{\omega_t}).
\tag{35}
$$

Combining (34), (35) with (32), we have

$$
\begin{aligned}
\mathsf{Regret}(T) &\leq \underbrace{\sum_{t=1}^T \left( f^{\star,\nu_t}(A) - f^{\mu_t,\star}(A) - (f^{\star,\nu_t}(A_{\omega_t}) - f^{\mu_t,\star}(A_{\omega_t})) \right)}_{(i)} \\
&\quad + \underbrace{\sum_{t=1}^T \left( f^{\widetilde{\mu}_t,\nu_t}(A_{\omega_t}) - f^{\widetilde{\mu}_t,\nu_t}(A) \right) + \sum_{t=1}^T \left( f^{\mu_t,\widetilde{\nu}_t}(A) - f^{\mu_t,\widetilde{\nu}_t}(A_{\omega_t}) \right)}_{(ii)} \\
&\quad + \underbrace{\sum_{t=1}^T \left( f^{\widetilde{\mu}_t,\nu_t}(A) - f^{\widetilde{\mu}_t,\nu_t}(A_{\omega_{t-1}}) \right) + \sum_{t=1}^T \left( f^{\mu_t,\widetilde{\nu}_t}(A_{\omega_{t-1}}) - f^{\mu_t,\widetilde{\nu}_t}(A) \right)}_{(iii)}.
\end{aligned}
\tag{36}
$$

We will upper bound the three terms in the right-hand side of (36) separately.

**Step 1: bounding term (i).** Define the squared loss function $L_t(\omega)$ over the dataset $\mathcal{D}_{t-1}$ as

$$L_t(\omega) := \sum_{(i,j,\widehat{A}(i,j)) \in \mathcal{D}_{t-1}} \left( A_\omega(i,j) - \widehat{A}(i,j) \right)^2. \tag{37}$$

Then by the optimality of $\omega_t$ for (7), we know that

$$L_t(\omega_t) - \alpha f^{\star,\nu_t}(A_{\omega_t}) + \alpha f^{\mu_t,\star}(A_{\omega_t}) \le L_t(\omega^\star) - \alpha f^{\star,\nu_t}(A) + \alpha f^{\mu_t,\star}(A),$$

where we use Assumption 2.2, which implies $A_{\omega^\star} = A$. Reorganizing the terms, we have

$$\text{(i)} \le \frac{1}{\alpha} \sum_{t=1}^{T} \left( L_t(\omega^\star) - L_t(\omega_t) \right). \tag{38}$$

Thus, it is sufficient to bound the term $\sum_{t=1}^{T} \left( L_t(\omega^\star) - L_t(\omega_t) \right)$. For any $t \in [T]$, we denote

$$\widehat{A}(i_t, j_t) = A(i_t, j_t) + \xi_t \quad and \quad \widehat{A}(i_t', j_t') = A(i_t, j_t) + \xi_t'.$$

It follows that we can rewrite $L_t(\omega)$ as

$$L_t(\omega) = \sum_{s=1}^{t-1} \left( A_\omega(i_s, j_s) - A(i_s, j_s) - \xi_s \right)^2 + \sum_{s=1}^{t-1} \left( A_\omega(i_s', j_s') - A(i_s', j_s') - \xi_s' \right)^2,$$

from which we deduce

$$L_t(\omega^\star) - L_t(\omega) = -\sum_{s=1}^{t-1} \left[ \left( A_\omega(i_s, j_s) - A(i_s, j_s) - \xi_s \right)^2 - \xi_s^2 \right] - \sum_{s=1}^{t-1} \left[ \left( A_\omega(i_s', j_s') - A(i_s', j_s') - \xi_s' \right)^2 - (\xi_s')^2 \right]$$

$$= -\sum_{s=1}^{t-1} \underbrace{\left[ \left( A_\omega(i_s, j_s) - A(i_s, j_s) \right) \left( A_\omega(i_s, j_s) - A(i_s, j_s) - 2\xi_s \right) \right]}_{:= X_s^\omega}$$

$$- \sum_{s=1}^{t-1} \underbrace{\left[ \left( A_\omega(i_s', j_s') - A(i_s', j_s') \right) \left( A_\omega(i_s', j_s') - A(i_s', j_s') - 2\xi_s' \right) \right]}_{:= Y_s^\omega}. \tag{39}$$

It is then sufficient to bound $-\sum_{s=1}^{t-1} X_s^\omega$ and $-\sum_{s=1}^{t-1} Y_s^\omega$, which is supplied by the following lemma.

**Lemma B.6.** *When Assumption 2.1, 2.2, 2.3 hold, for any $\delta \in (0,1)$, with probability at least $1 - \delta$, it holds*

$$\forall t \in [T], \omega \in \Omega: \quad -\sum_{s=1}^{t-1} X_s^\omega \le -\frac{1}{2} \sum_{s=1}^{t-1} \mathbb{E}_{i \sim \widetilde{\mu}_s, j \sim \nu_s} \left[ \left( A_\omega(i,j) - A(i,j) \right)^2 \right]$$

$$+ CB_l \left( B_l + \sigma \sqrt{2 \log(8T/\delta)} \right) \left( \log\left( \frac{4T}{\delta} \right) + d \log\left( 1 + 2T\sqrt{d} \right) \right) \tag{40}$$

*and*

$$\forall t \in [T], \omega \in \Omega: \quad -\sum_{s=1}^{t-1} Y_s^\omega \le -\frac{1}{2} \sum_{s=1}^{t-1} \mathbb{E}_{i \sim \mu_s, j \sim \widetilde{\nu}_s} \left[ \left( A_\omega(i,j) - A(i,j) \right)^2 \right]$$

$$+ CB_l \left( B_l + \sigma \sqrt{2 \log(8T/\delta)} \right) \left( \log\left( \frac{4T}{\delta} \right) + d \log\left( 1 + 2T\sqrt{d} \right) \right) \tag{41}$$

*where $C > 0$ is some universal constant.*

Combining (39), (38) and Lemma B.6 leads to a bound of term (i):

$$\text{(i)} \le \frac{1}{\alpha} \left\{ -\frac{1}{2} \sum_{t=1}^{T} \sum_{s=1}^{t-1} \mathbb{E}_{i \sim \widetilde{\mu}_s, j \sim \nu_s} \left[ (A_{\omega_t}(i,j) - A(i,j))^2 \right] - \frac{1}{2} \sum_{t=1}^{T} \sum_{s=1}^{t-1} \mathbb{E}_{i \sim \mu_s, j \sim \widetilde{\nu}_s} \left[ (A_{\omega_t}(i,j) - A(i,j))^2 \right] \right.$$

$$\left. + 2T \cdot CB_l \left( B_l + \sigma \sqrt{2 \log(8T/\delta)} \right) \left( \log \left( \frac{4T}{\delta} \right) + d \log \left( 1 + 2T\sqrt{d} \right) \right) \right\}. \tag{42}$$

**Step 2: bounding terms (ii) and (iii).** To bound (ii) and (iii), we first prove the following lemma.

**Lemma B.7.** *For any* $\{(\widehat{\mu}_t, \widehat{\nu}_t)\}_{t \in [T]} \subset \Delta^m \times \Delta^n$ *and any* $\{\widehat{\omega}_t\}_{t \in [T]} \subset \Omega$, *we have*

$$\sum_{t=1}^{T} \left| f^{\widehat{\mu}_t, \widehat{\nu}_t}(A_{\widehat{\omega}_t}) - f^{\widehat{\mu}_t, \widehat{\nu}_t}(A) \right| \le \frac{d(\lambda)}{2\eta} + \frac{\eta}{2} \sum_{t=1}^{T} \sum_{s=1}^{t-1} \mathbb{E}_{i \sim \widehat{\mu}_s, j \sim \widehat{\nu}_s} [(A_{\widehat{\omega}_t}(i,j) - A(i,j))^2]$$

$$+ (\sqrt{d} + 2B_l) \min\{d(\lambda), T\} + \sqrt{d}\lambda \tag{43}$$

*for any* $\lambda, \eta > 0$, *where* $d(\lambda) := 2d \log \left( 1 + \frac{T}{d\lambda} \right)$.

By letting $\widehat{\mu}_t = \widetilde{\mu}_t$, $\widehat{\nu}_t = \nu_t$ and $\widehat{\omega}_t = \omega_t$ in Lemma B.7, we have

$$\sum_{t=1}^{T} \left( f^{\widetilde{\mu}_t, \nu_t}(A_{\omega_t}) - f^{\widetilde{\mu}_t, \nu_t}(A) \right) \le \frac{d(\lambda)}{2\eta} + \frac{\eta}{2} \sum_{t=1}^{T} \sum_{s=1}^{t-1} \mathbb{E}_{i \sim \widetilde{\mu}_s, j \sim \nu_s} [(A_{\omega_t}(i,j) - A(i,j))^2]$$

$$+ (\sqrt{d} + 2B_l) \min\{d(\lambda), T\} + \sqrt{d}\lambda T. \tag{44}$$

By letting $\widehat{\mu}_t = \mu_t$, $\widehat{\nu}_t = \widetilde{\nu}_t$ and $\widehat{\omega}_t = \omega_t$ in Lemma B.7, we have

$$\sum_{t=1}^{T} \left( f^{\mu_t, \widetilde{\nu}_t}(A) - f^{\mu_t, \widetilde{\nu}_t}(A_{\omega_t}) \right) \le \frac{d(\lambda)}{2\eta} + \frac{\eta}{2} \sum_{t=1}^{T} \sum_{s=1}^{t-1} \mathbb{E}_{i \sim \mu_s, j \sim \widetilde{\nu}_s} [(A_{\omega_t}(i,j) - A(i,j))^2]$$

$$+ (\sqrt{d} + 2B_l) \min\{d(\lambda), T\} + \sqrt{d}\lambda T. \tag{45}$$

Similarly, we have

$$\sum_{t=1}^{T} \left( f^{\widetilde{\mu}_t, \nu_t}(A) - f^{\widetilde{\mu}_t, \nu_t}(A_{\omega_{t-1}}) \right) \le \frac{d(\lambda)}{2\eta} + \frac{\eta}{2} \sum_{t=1}^{T} \sum_{s=1}^{t-2} \mathbb{E}_{i \sim \widetilde{\mu}_s, j \sim \nu_s} [(A_{\omega_{t-1}}(i,j) - A(i,j))^2]$$

$$+ (\sqrt{d} + 2B_l) \min\{d(\lambda), T\} + \sqrt{d}\lambda T + 2B_l^2 \eta T, \tag{46}$$

which uses the fact

$$\mathbb{E}_{i \sim \widetilde{\mu}_{t-1}, j \sim \nu_{t-1}} [(A_{\omega_{t-1}}(i,j) - A(i,j))^2] \le 4B_l^2.$$

Notice that the second term in (46) can be further bounded by

$$\sum_{t=1}^{T} \sum_{s=1}^{t-2} \mathbb{E}_{i \sim \widetilde{\mu}_s, j \sim \nu_s} [(A_{\omega_{t-1}}(i,j) - A(i,j))^2] = \sum_{t=0}^{T-1} \sum_{s=1}^{t-1} \mathbb{E}_{i \sim \widetilde{\mu}_s, j \sim \nu_s} [(A_{\omega_t}(i,j) - A(i,j))^2]$$

$$\le \sum_{t=0}^{T} \sum_{s=1}^{t-1} \mathbb{E}_{i \sim \widetilde{\mu}_s, j \sim \nu_s} [(A_{\omega_t}(i,j) - A(i,j))^2]$$

$$= \sum_{t=1}^{T} \sum_{s=1}^{t-1} \mathbb{E}_{i \sim \widetilde{\mu}_s, j \sim \nu_s} [(A_{\omega_t}(i,j) - A(i,j))^2], \tag{47}$$

where the first line shifts the index of $t$ by 1, and the last equality holds because the term is 0 when $t = 0$. Plugging the above inequality back to (46) leads to

$$\sum_{t=1}^{T} \left( f^{\widetilde{\mu}_t, \nu_t}(A) - f^{\widetilde{\mu}_t, \nu_t}(A_{\omega_{t-1}}) \right) = \frac{d(\lambda)}{2\eta} + \frac{\eta}{2} \sum_{t=1}^{T} \sum_{s=1}^{t-1} \mathbb{E}_{i \sim \widetilde{\mu}_s, j \sim \nu_s} [(A_{\omega_t}(i,j) - A(i,j))^2]$$

$$+ (\sqrt{d} + 2B_l) \min\{d(\lambda), T\} + \sqrt{d}\lambda T + 2B_l^2 \eta T. \tag{48}$$

Analogously, we have

$$\sum_{t=1}^{T} \left( f^{\mu_t, \widetilde{\nu}_t}(A_{\omega_{t-1}}) - f^{\mu_t, \widetilde{\nu}_t}(A) \right) \leq \frac{d(\lambda)}{2\eta} + \frac{\eta}{2} \sum_{t=1}^{T} \sum_{s=1}^{t-1} \mathbb{E}_{i \sim \mu_s, j \sim \widetilde{\nu}_s}[(A_{\omega_t}(i,j) - A(i,j))^2]$$
$$+ (\sqrt{d} + 2B_l) \min\{d(\lambda), T\} + \sqrt{d}\lambda T + 2B_l^2 \eta T. \tag{49}$$

Combining (44), (45), (48), and (49), we have

$$\text{(ii)} + \text{(iii)} \leq \eta \left\{ \sum_{t=1}^{T} \sum_{s=1}^{t-1} \mathbb{E}_{i \sim \widetilde{\mu}_s, j \sim \nu_s}[(A_{\omega_t}(i,j) - A(i,j))^2] + \sum_{t=1}^{T} \sum_{s=1}^{t-1} \mathbb{E}_{i \sim \mu_s, j \sim \widetilde{\nu}_s}[(A_{\omega_t}(i,j) - A(i,j))^2] \right\}$$
$$+ \frac{2d(\lambda)}{\eta} + 4(\sqrt{d} + 2B_l) \min\{d(\lambda), T\} + 4\sqrt{d}\lambda T + 4B_l^2 \eta T. \tag{50}$$

**Step 3: combining the bounds.** Letting $\eta = \frac{1}{2\alpha}$ in (50), the first line of (42) could cancel out the first line of (50), which leads to

$$\text{Regret}(T) = \text{(i)} + \text{(ii)} + \text{(iii)}$$
$$= \frac{T}{\alpha} \cdot 2CB_l \left( B_l + \sigma\sqrt{2\log(8T/\delta)} \right) \left( \log\left(\frac{4T}{\delta}\right) + d\log\left(1 + 2T\sqrt{d}\right) \right)$$
$$+ 4\alpha d(\lambda) + 4(\sqrt{d} + 2B_l) \min\{d(\lambda), T\} + 4\sqrt{d}\lambda T + \frac{2B_l^2 T}{\alpha} \tag{51}$$

with probability at least $1 - \delta$.

By choosing

$$\alpha = \sqrt{\frac{T\left(\log(4T/\delta) + d\log\left(1 + 2\sqrt{d}T\right)\right)}{d\log\left(1 + (T/d)^{3/2}\right)}} \quad \text{and} \quad \lambda = \sqrt{\frac{d}{T}},$$

we have with probability at least $1 - \delta$,

$$\text{Regret}(T)$$
$$\leq 2\left( CB_l \left( B_l + \sigma\sqrt{2\log(8T/\delta)} \right) + 1 \right) d\sqrt{T} \cdot \sqrt{\left( \frac{1}{d}\log(4T/\delta) + \log\left(1 + 2\sqrt{d}T\right) \right) \log\left(1 + (T/d)^{3/2}\right)}$$
$$+ 2B_l^2 \sqrt{T} \sqrt{\frac{d\log\left(1 + (T/d)^{3/2}\right)}{\log(4T/\delta) + d\log\left(1 + 2\sqrt{d}T\right)}} + 4(\sqrt{d} + 2B_l)d\log\left(1 + (T/d)^{3/2}\right) + 4d\sqrt{T} \tag{52}$$

for some absolute constant $C > 0$, and thus the regret could be bounded by (13) by simplifying the logarithmic terms.

### B.2.1. PROOF OF LEMMA B.6

To begin, by Assumption 2.3 together with the sub-Gaussian concentration inequality, we have that with probability at least $1 - \frac{\delta}{2}$, for any $s \in [T]$ and $\omega \in \Omega$,

$$\mathbb{P}(|\xi_s| \geq a) \leq 2\exp\left(-\frac{a^2}{2\sigma^2}\right) \quad \text{and} \quad \mathbb{P}(|\xi_s'| \geq a) \leq 2\exp\left(-\frac{a^2}{2\sigma^2}\right) \quad \text{for all } a > 0,$$

which implies that with probability at least $1 - \frac{\delta}{2}$,

$$|\xi_s| \leq \sigma\sqrt{2\log(8T/\delta)}, \quad |\xi_s'| \leq \sigma\sqrt{2\log(8T/\delta)}, \quad \forall s \in [T]. \tag{53}$$

We let $\mathcal{E}$ be the event that (53) holds for all $s \in [T]$, which satisfies $\mathbb{P}(\mathcal{E}) \geq 1 - \frac{\delta}{2}$.

Next, we define filtrations $\mathcal{F}_t := \sigma(\mathcal{D}_t)$ for all $t \in [T]$. By Assumption 2.3, we have for all $s \in [T]$ and $\omega \in \Omega$,

$$\mathbb{E}[X_s^\omega | \mathcal{F}_{s-1}] = \mathbb{E}_{i \sim \widetilde{\mu}_s, j \sim \nu_s} \left[ (A_\omega(i,j) - A(i,j))^2 \right], \tag{54}$$

$$\mathbb{E}[Y_s^\omega | \mathcal{F}_{s-1}] = \mathbb{E}_{i \sim \mu_s, j \sim \widetilde{\nu}_s} \left[ (A_\omega(i,j) - A(i,j))^2 \right]. \tag{55}$$

We also have

$$\begin{aligned}
\mathsf{Var}\left[X_s^\omega | \mathcal{F}_{s-1}\right] &\leq \mathbb{E}\left[(X_s^\omega)^2 | \mathcal{F}_{s-1}\right] \\
&= \mathbb{E}\left[ (A_\omega(i_s, j_s) - A(i_s, j_s))^2 (A_\omega(i_s, j_s) - A(i_s, j_s) - 2\xi_s)^2 | \mathcal{F}_{s-1} \right] \\
&\leq 4(B_l^2 + \sigma^2) \mathbb{E}\left[ (A_\omega(i_s, j_s) - A(i_s, j_s))^2 | \mathcal{F}_{s-1} \right] \\
&= 4(B_l^2 + \sigma^2) \mathbb{E}_{i \sim \widetilde{\mu}_s, j \sim \nu_s} \left[ (A_\omega(i,j) - A(i,j))^2 \right],
\end{aligned} \tag{56}$$

where we use Assumptions 2.1, 2.2 and 2.3 in the last inequality: Assumption 2.2 guarantees that $A_{\omega^\star} = A$, and Assumption 2.1 indicates that $\|A(i,j)\|_\infty := \max_{i \in [m], j \in [n]} |A(i,j)| \leq B_l$ and $\|A_\omega(i,j)\|_\infty \leq B_l$ for all $\omega \in \Omega$; moreover, Assumption 2.3 implies $\mathbb{E}\xi_s^2 \leq \sigma^2$. Conditioned on event $\mathcal{E}$, we can bound $|X_s^\omega - \mathbb{E}[X_s^\omega | \mathcal{F}_{s-1}]|$ using (39) and (54) as follows:

$$\begin{aligned}
&|X_s^\omega - \mathbb{E}[X_s^\omega | \mathcal{F}_{s-1}]| \\
&= \left| (A_\omega(i_s, j_s) - A(i_s, j_s)) (A_\omega(i_s, j_s) - A(i_s, j_s) - 2\xi_s) - \mathbb{E}_{i \sim \widetilde{\mu}_s, j \sim \nu_s} \left[ (A_\omega(i,j) - A(i,j))^2 \right] \right| \\
&\leq \left| (A_\omega(i_s, j_s) - A(i_s, j_s))^2 - \mathbb{E}_{i \sim \widetilde{\mu}_s, j \sim \nu_s} \left[ (A_\omega(i,j) - A(i,j))^2 \right] \right| + 2|\xi_s| |A_\omega(i_s, j_s) - A(i_s, j_s)| \\
&\leq 4B_l \left( B_l + \sigma\sqrt{2\log(8T/\delta)} \right).
\end{aligned} \tag{57}$$

In what follows, we apply a standard covering argument together with the Freedman's inequality to prove the desired bound, conditioned on event $\mathcal{E}$. First, for any $\mathcal{X} \subset \mathbb{R}^d$, let $\mathcal{N}(\mathcal{X}, \epsilon, \|\cdot\|)$ be the $\epsilon$-covering number of $\mathcal{X}$ with respect to the norm $\|\cdot\|$. By Assumption 2.1 we know that $\Omega \subset \mathbb{B}_2^d(\sqrt{d})$. Thus by Lemma B.5 we have

$$\log \mathcal{N}(\Omega, \epsilon, \|\cdot\|_2) \leq \log \mathcal{N}(\mathbb{B}_2^d(\sqrt{d}), \epsilon, \|\cdot\|_2) \leq d \log \left( 1 + \frac{2\sqrt{d}}{\epsilon} \right) \tag{58}$$

for any $\epsilon > 0$. In other words, for any $\epsilon > 0$, there exists an $\epsilon$-net $\Omega_\epsilon \subset \Omega$ such that $\log |\Omega_\epsilon| \lesssim d \log \left( 1 + \frac{2\sqrt{d}}{\epsilon} \right)$.

Applying Freedman's inequality (c.f. Lemma B.1) to the martingale difference sequence $\{\mathbb{E}[X_s^\omega | \mathcal{F}_{s-1}] - X_s^\omega\}_{s \in [T]}$ and making use of (54), (56) and (57) we have under event $\mathcal{E}$, with probability at least $1 - \frac{\delta}{4}$,

$$\begin{aligned}
\forall t \in [T], \omega \in \Omega_\epsilon: \quad &\sum_{s=1}^t \left( \mathbb{E}_{i \sim \widetilde{\mu}_s, j \sim \nu_s} \left[ (A_\omega(i,j) - A(i,j))^2 \right] - X_s^\omega \right) \\
&\leq \frac{1}{2} \sum_{s=1}^t \mathbb{E}_{i \sim \widetilde{\mu}_s, j \sim \nu_s} \left[ (A_\omega(i,j) - A(i,j))^2 \right] \\
&\quad + 4CB_l \left( B_l + \sigma\sqrt{2\log(8T/\delta)} \right) \left( \log\left( \frac{4T}{\delta} \right) + d \log \left( 1 + \frac{2\sqrt{d}}{\epsilon} \right) \right),
\end{aligned}$$

where $C > 0$ is an absolute constant.

In addition, conditioned on event $\mathcal{E}$, for any $\omega, \omega' \in \Omega$, $\|\omega - \omega'\|_2 \leq \epsilon$, we have

$$\left| \left( \frac{1}{2} \mathbb{E}_{i \sim \widetilde{\mu}_s, j \sim \nu_s} \left[ (A_\omega(i,j) - A(i,j))^2 \right] - X_s^\omega \right) - \left( \frac{1}{2} \mathbb{E}_{i \sim \widetilde{\mu}_s, j \sim \nu_s} \left[ (A_{\omega'}(i,j) - A(i,j))^2 \right] - X_s^{\omega'} \right) \right|$$

$$\leq \frac{1}{2} \mathbb{E}_{i \sim \widetilde{\mu}_s, j \sim \nu_s} \left| (A_\omega(i,j) - A(i,j))^2 - (A_{\omega'}(i,j) - A(i,j))^2 \right| + |X_s^\omega - X_s^{\omega'}|$$

$$\leq \left( 6B_l + 2\sigma\sqrt{2\log(8T/\delta)} \right) \epsilon.$$

Thus combining the above two expressions and set $\epsilon = \frac{1}{T}$, we have that under event $\mathcal{E}$, with probability at least $1 - \frac{\delta}{4}$,

$$\forall t \in [T], \omega \in \Omega: \quad \sum_{s=1}^{t} \left( \mathbb{E}_{i \sim \widetilde{\mu}_s, j \sim \nu_s} \left[ (A_\omega(i,j) - A(i,j))^2 \right] - X_s^\omega \right)$$

$$\leq \frac{1}{2} \sum_{s=1}^{t} \mathbb{E}_{i \sim \widetilde{\mu}_s, j \sim \nu_s} \left[ (A_\omega(i,j) - A(i,j))^2 \right]$$

$$+ 4CB_l \left( B_l + \sigma\sqrt{2\log(8T/\delta)} \right) \left( \log\left( \frac{4T}{\delta} \right) + d\log\left( 1 + 2T\sqrt{d} \right) \right) \tag{59}$$

for sufficiently large constant $C$.

Rearranging terms, we have with probability at least $1 - \frac{\delta}{4}$,

$$\forall t \in [T], \omega \in \Omega: \quad -\sum_{s=1}^{t-1} X_s^\omega \leq -\frac{1}{2} \sum_{s=1}^{t-1} \mathbb{E}_{i \sim \widetilde{\mu}_s, j \sim \nu_s} \left[ (A_\omega(i,j) - A(i,j))^2 \right]$$

$$+ 4CB_l \left( B_l + \sigma\sqrt{2\log(8T/\delta)} \right) \left( \log\left( \frac{4T}{\delta} \right) + d\log\left( 1 + 2T\sqrt{d} \right) \right). \tag{60}$$

Similar to (40), conditioned on event $\mathcal{E}$, we could upper bound $-\sum_{s=1}^{t-1} Y_s^\omega$ as follows with probability at least $1 - \frac{\delta}{4}$:

$$\forall t \in [T], \omega \in \Omega: \quad -\sum_{s=1}^{t-1} Y_s^\omega \leq -\frac{1}{2} \sum_{s=1}^{t-1} \mathbb{E}_{i \sim \mu_s, j \sim \widetilde{\nu}_s} \left[ (A_\omega(i,j) - A(i,j))^2 \right]$$

$$+ 4CB_l \left( B_l + \sigma\sqrt{2\log(8T/\delta)} \right) \left( \log\left( \frac{4T}{\delta} \right) + d\log\left( 1 + 2T\sqrt{d} \right) \right). \tag{61}$$

Applying union bound completes the proof of Lemma B.6.

### B.2.2. PROOF OF LEMMA B.7

For any $\mu, \nu \in \Delta^m \times \Delta^n$ and any $\omega \in \Omega$, notice that

$$f^{\mu,\nu}(A_\omega) - f^{\mu,\nu}(A) = \langle \underbrace{\mathbb{E}_{i \sim \mu, j \sim \nu}[\phi(i,j)]}_{=:x(\mu,\nu)}, \omega - \omega^\star \rangle, \tag{62}$$

where we denote $\mathbb{E}_{i \sim \mu, j \sim \nu}[\phi(i,j)]$ as $x(\mu,\nu)$ for simplicity. By Assumption 2.1, it guarantees that $\|x(\mu,\nu)\|_\infty \leq 1$ for all $\mu, \nu$. For each $t \in [T]$, we define $\Lambda_t \in \mathbb{R}^{d \times d}$ as

$$\Lambda_t := \lambda I_d + \sum_{s=1}^{t-1} x(\widehat{\mu}_s, \widehat{\nu}_s) x(\widehat{\mu}_s, \widehat{\nu}_s)^\top \tag{63}$$

for any $\lambda > 0$. By Lemma B.2 and Lemma B.3, we have

$$\sum_{s=1}^{t} \min\left\{ \|x(\widehat{\mu}_s, \widehat{\nu}_s)\|_{\Lambda_s^{-1}}, 1 \right\} \leq 2d\log\left( 1 + \frac{T}{d\lambda} \right) := d(\lambda), \tag{64}$$

which will be used repeatedly in the proof.

We decompose $\sum_{t=1}^{T} \left| f^{\widehat{\mu}_t, \widehat{\nu}_t}(A_{\widehat{\omega}_t}) - f^{\widehat{\mu}_t, \widehat{\nu}_t}(A) \right|$ into two terms:

$$\sum_{t=1}^{T} \left| f^{\widehat{\mu}_t, \widehat{\nu}_t}(A_{\widehat{\omega}_t}) - f^{\widehat{\mu}_t, \widehat{\nu}_t}(A) \right| = \underbrace{\sum_{t=1}^{T} |\langle x(\widehat{\mu}_t, \widehat{\nu}_t), \widehat{\omega}_t - \omega^\star \rangle| \, \mathbb{1} \left\{ \|x(\widehat{\mu}_t, \widehat{\nu}_t)\|_{\Lambda_t^{-1}} \leq 1 \right\}}_{(a)}$$

$$+ \underbrace{\sum_{t=1}^{T} |\langle x(\widehat{\mu}_t, \widehat{\nu}_t), \widehat{\omega}_t - \omega^\star \rangle| \, \mathbb{1} \left\{ \|x(\widehat{\mu}_t, \widehat{\nu}_t)\|_{\Lambda_t^{-1}} > 1 \right\}}_{(b)}. \tag{65}$$

To prove Lemma B.7, below we bound (a) and (b) separately.

**Step 1: bounding term (a).** To bound term (a), it follows that

$$(a) = \sum_{t=1}^{T} |\langle x(\widehat{\mu}_t, \widehat{\nu}_t), \widehat{\omega}_t - \omega^\star \rangle| \, \mathbb{1} \left\{ \|x(\widehat{\mu}_t, \widehat{\nu}_t)\|_{\Lambda_t^{-1}} \leq 1 \right\}$$

$$\leq \sum_{t=1}^{T} \|\widehat{\omega}_t - \omega^\star\|_{\Lambda_t} \|x(\widehat{\mu}_t, \widehat{\nu}_t)\|_{\Lambda_t^{-1}} \, \mathbb{1} \left\{ \|x(\widehat{\mu}_t, \widehat{\nu}_t)\|_{\Lambda_t^{-1}} \leq 1 \right\}$$

$$\leq \sum_{t=1}^{T} \|\widehat{\omega}_t - \omega^\star\|_{\Lambda_t} \min \left\{ \|x(\widehat{\mu}_t, \widehat{\nu}_t)\|_{\Lambda_t^{-1}}, 1 \right\}, \tag{66}$$

where the first inequality uses the Cauchy-Schwarz inequality, and the second inequality uses the fact that

$$\|x(\widehat{\mu}_t, \widehat{\nu}_t)\|_{\Lambda_t^{-1}} \, \mathbb{1} \left\{ \|x(\widehat{\mu}_t, \widehat{\nu}_t)\|_{\Lambda_t^{-1}} \leq 1 \right\} \leq \min \left\{ \|x(\widehat{\mu}_t, \widehat{\nu}_t)\|_{\Lambda_t^{-1}}, 1 \right\}.$$

Also, by Assumption 2.1 and the definition of $\Lambda_t$ in (63), we have

$$\|\widehat{\omega}_t - \omega^\star\|_{\Lambda_t} \leq 2\sqrt{\lambda d} + \left( \sum_{s=1}^{t-1} |\langle \widehat{\omega}_t - \omega^\star, x(\widehat{\mu}_s, \widehat{\nu}_s) \rangle|^2 \right)^{1/2}, \tag{67}$$

which gives

$$\sum_{t=1}^{T} \|\widehat{\omega}_t - \omega^\star\|_{\Lambda_t} \min \left\{ \|x(\widehat{\mu}_t, \widehat{\nu}_t)\|_{\Lambda_t^{-1}}, 1 \right\}$$

$$\leq \sum_{t=1}^{T} \left( 2\sqrt{\lambda d} + \left( \sum_{s=1}^{t-1} |\langle \widehat{\omega}_t - \omega^\star, x(\widehat{\mu}_s, \widehat{\nu}_s) \rangle|^2 \right)^{1/2} \right) \cdot \min \left\{ \|x(\widehat{\mu}_t, \widehat{\nu}_t)\|_{\Lambda_t^{-1}}, 1 \right\}$$

$$\leq \left( \sum_{t=1}^{T} 4\lambda d \right)^{1/2} \left( \sum_{t=1}^{T} \min \left\{ \|x(\widehat{\mu}_t, \widehat{\nu}_t)\|_{\Lambda_t^{-1}}, 1 \right\} \right)^{1/2}$$

$$+ \left( \sum_{t=1}^{T} \sum_{s=1}^{t-1} |\langle \widehat{\omega}_t - \omega^\star, x(\widehat{\mu}_s, \widehat{\nu}_s) \rangle|^2 \right)^{1/2} \left( \sum_{t=1}^{T} \min \left\{ \|x(\widehat{\mu}_t, \widehat{\nu}_t)\|_{\Lambda_t^{-1}}, 1 \right\} \right)^{1/2}$$

$$\leq 2\sqrt{\lambda d T \min\{d(\lambda), T\}} + \left( d(\lambda) \sum_{t=1}^{T} \sum_{s=1}^{t-1} |\langle \widehat{\omega}_t - \omega^\star, x(\widehat{\mu}_s, \widehat{\nu}_s) \rangle|^2 \right)^{1/2}, \tag{68}$$

where the first inequality uses (67) and the second inequality uses the Cauchy-Schwarz inequality and the fact that

$$\min \left\{ \|x(\widehat{\mu}_t, \widehat{\nu}_t)\|_{\Lambda_t^{-1}}, 1 \right\}^2 \leq \min \left\{ \|x(\widehat{\mu}_t, \widehat{\nu}_t)\|_{\Lambda_t^{-1}}, 1 \right\},$$

and the last inequality uses (64).

Plugging (68) into (66), we have

$$(a) \leq 2\sqrt{d} \cdot \sqrt{\lambda T \min\{d(\lambda), T\}} + \left( d(\lambda) \sum_{t=1}^{T} \sum_{s=1}^{t-1} |\langle \widehat{\omega}_t - \omega^\star, x(\widehat{\mu}_s, \widehat{\nu}_s) \rangle|^2 \right)^{1/2}. \tag{69}$$

**Step 2: bounding term (b).** It follows that

$$(b) = \sum_{t=1}^{T} |\langle x(\widehat{\mu}_t, \widehat{\nu}_t), \widehat{\omega}_t - \omega^\star \rangle| \, \mathbb{1}\left\{ \|x(\widehat{\mu}_t, \widehat{\nu}_t)\|_{\Lambda_t^{-1}} > 1 \right\}$$

$$\leq \sum_{t=1}^{T} |\langle x(\widehat{\mu}_t, \widehat{\nu}_t), \widehat{\omega}_t - \omega^\star \rangle| \min\left\{ \|x(\widehat{\mu}_t, \widehat{\nu}_t)\|_{\Lambda_t^{-1}}, 1 \right\} \leq 2B_l \min\{T, d(\lambda)\}, \tag{70}$$

where the first inequality uses the fact that

$$\mathbb{1}\left\{ \|x(\widehat{\mu}_t, \widehat{\nu}_t)\|_{\Lambda_t^{-1}} > 1 \right\} \leq \min\left\{ \|x(\widehat{\mu}_t, \widehat{\nu}_t)\|_{\Lambda_t^{-1}}, 1 \right\},$$

and the last inequality uses Assumption 2.1 and (64).

**Step 3: combining (a) and (b).** Plugging (69) and (70) into (65), we have

$$\sum_{t=1}^{T} \left| f^{\widehat{\mu}_t, \widehat{\nu}_t}(A_{\widehat{\omega}_t}) - f^{\widehat{\mu}_t, \widehat{\nu}_t}(A) \right|$$

$$\leq 2\sqrt{d} \cdot \sqrt{\lambda T \min\{d(\lambda), T\}} + \left( d(\lambda) \sum_{t=1}^{T} \sum_{s=1}^{t-1} |\langle \widehat{\omega}_t - \omega^\star, x(\widehat{\mu}_s, \widehat{\nu}_s) \rangle|^2 \right)^{1/2} + 2B_l \min\{T, d(\lambda)\}$$

$$\leq \left( \frac{d(\lambda)}{\eta} \cdot \eta \sum_{t=1}^{T} \sum_{s=1}^{t-1} |\langle \widehat{\omega}_t - \omega^\star, x(\widehat{\mu}_s, \widehat{\nu}_s) \rangle|^2 \right)^{1/2} + (\sqrt{d} + 2B_l) \min\{d(\lambda), T\} + \sqrt{d}\lambda T$$

$$\leq \frac{d(\lambda)}{2\eta} + \frac{\eta}{2} \sum_{t=1}^{T} \sum_{s=1}^{t-1} |\langle \widehat{\omega}_t - \omega^\star, x(\widehat{\mu}_s, \widehat{\nu}_s) \rangle|^2 + (\sqrt{d} + 2B_l) \min\{d(\lambda), T\} + \sqrt{d}\lambda T \tag{71}$$

for any $\eta > 0$, where the second and third inequalities both use the fact that $\sqrt{ab} \leq \frac{a+b}{2}$ for any $a, b \geq 0$. The proof is completed by plugging in the following fact into the above relation: for any $\mu, \nu \in \Delta^m \times \Delta^n$ and any $\omega \in \Omega$, we have

$$|\langle x(\mu, \nu), \omega - \omega^\star \rangle|^2 = |\mathbb{E}_{i \sim \mu, j \sim \nu}[A_\omega(i, j) - A(i, j)]|^2 \leq \mathbb{E}_{i \sim \mu, j \sim \nu}[(A_\omega(i, j) - A(i, j))^2]. \tag{72}$$

### B.3. Proof of Theorem 3.3

For notation simplicity, we define

$$\widetilde{\boldsymbol{\pi}}_{t,n} := (\widetilde{\pi}_t^n, \boldsymbol{\pi}_t^{-n}), \quad \forall n \in [N]. \tag{73}$$

Analogous to (32), here we decompose the regret as

$$\text{Regret}(T) = \sum_{t=1}^{T} \frac{1}{N} \sum_{n=1}^{N} \left( V_n^{\star, \boldsymbol{\pi}_t^{-n}}(\rho) - V_n^{\boldsymbol{\pi}_t}(\rho) \right),$$

$$= \sum_{t=1}^{T} \frac{1}{N} \sum_{n=1}^{N} \left( V_n^{\star, \boldsymbol{\pi}_t^{-n}}(\rho) - V_{f_t, n}^{\star, \boldsymbol{\pi}_t^{-n}}(\rho) \right) + \sum_{t=1}^{T} \frac{1}{N} \sum_{n=1}^{N} \left( V_{f_t, n}^{\star, \boldsymbol{\pi}_t^{-n}}(\rho) - V_n^{\widetilde{\pi}_t^n, \boldsymbol{\pi}_t^{-n}}(\rho) \right)$$

$$+ \sum_{t=1}^{T} \frac{1}{N} \sum_{n=1}^{N} \left( V_n^{\widetilde{\pi}_t^n, \boldsymbol{\pi}_t^{-n}}(\rho) - V_{f_{t-1},n}^{\widetilde{\pi}_t^n, \boldsymbol{\pi}_t^{-n}}(\rho) \right) + \sum_{t=1}^{T} \frac{1}{N} \sum_{n=1}^{N} \left( V_{f_{t-1},n}^{\widetilde{\pi}_t^n, \boldsymbol{\pi}_t^{-n}}(\rho) - V_{f_{t-1},n}^{\boldsymbol{\pi}_t}(\rho) \right)$$

$$+ \sum_{t=1}^{T} \frac{1}{N} \sum_{n=1}^{N} \left( V_{f_{t-1},n}^{\boldsymbol{\pi}_t}(\rho) - V_n^{\boldsymbol{\pi}_t}(\rho) \right). \tag{74}$$

By line 4 in Algorithm 2 we know that the second term in the third line of (74) is non-positive. Besides, (23) indicates

$$\forall n \in [N]: \quad V_{f_t,n}^{\star, \boldsymbol{\pi}_t^{-n}}(\rho) = V_{f_t,n}^{\widetilde{\pi}_t^n, \boldsymbol{\pi}_t^{-n}}(\rho). \tag{75}$$

Combining these two facts, we have

$$\mathsf{Regret}(T) \le \underbrace{\sum_{t=1}^{T} \frac{1}{N} \sum_{n=1}^{N} \left( V_n^{\star, \boldsymbol{\pi}_t^{-n}}(\rho) - V_{f_t,n}^{\star, \boldsymbol{\pi}_t^{-n}}(\rho) \right)}_{(i)} + \underbrace{\sum_{t=1}^{T} \frac{1}{N} \sum_{n=1}^{N} \left( V_{f_t,n}^{\widetilde{\pi}_t^n, \boldsymbol{\pi}_t^{-n}}(\rho) - V_n^{\widetilde{\pi}_t^n, \boldsymbol{\pi}_t^{-n}}(\rho) \right)}_{(ii)}$$

$$+ \underbrace{\sum_{t=1}^{T} \frac{1}{N} \sum_{n=1}^{N} \left( V_n^{\widetilde{\pi}_t^n, \boldsymbol{\pi}_t^{-n}}(\rho) - V_{f_{t-1},n}^{\widetilde{\pi}_t^n, \boldsymbol{\pi}_t^{-n}}(\rho) \right)}_{(iii)} + \underbrace{\sum_{t=1}^{T} \frac{1}{N} \sum_{n=1}^{N} \left( V_{f_{t-1},n}^{\boldsymbol{\pi}_t}(\rho) - V_n^{\boldsymbol{\pi}_t}(\rho) \right)}_{(iv)}. \tag{76}$$

In the following we upper bound each term in (76) separately.

**Step 1: bounding term (i).** By Assumption 3.2 we know that there exists $f^\star \in \mathcal{F}$ such that $f^\star := \mathbb{P} = \mathbb{P}_{f^\star}$. By the model update rule (22) in Algorithm 2 and the definition of the loss function (21), we have

$$\mathcal{L}_t(f_t) - \alpha \sum_{n=1}^{N} V_{f_t,n}^{\star, \boldsymbol{\pi}_t^{-n}}(\rho) \le \mathcal{L}_t(f^\star) - \alpha \sum_{n=1}^{N} V_n^{\star, \boldsymbol{\pi}_t^{-n}}(\rho)$$

from which we deduce

$$(i) \le \frac{1}{N\alpha} \sum_{t=1}^{T} \left( \mathcal{L}_t(f^\star) - \mathcal{L}_t(f_t) \right). \tag{77}$$

It then boils down to bounding the right-hand side of (77).

We first define random variables $X_{t,h}^f$ and $Y_{t,h,n}^f$ as

$$X_{t,h}^f := \log \left( \frac{\mathbb{P}_h(s_{t,h+1}|s_{t,h}, \boldsymbol{a}_{t,h})}{\mathbb{P}_{f,h}(s_{t,h+1}|s_{t,h}, \boldsymbol{a}_{t,h})} \right) \quad \text{and} \quad Y_{t,h,n}^f := \log \left( \frac{\mathbb{P}_h(s_{t,h+1}^n|s_{t,h}^n, \boldsymbol{a}_{t,h}^n)}{\mathbb{P}_{f,h}(s_{t,h+1}^n|s_{t,h}^n, \boldsymbol{a}_{t,h}^n)} \right), \quad \forall n \in [N]. \tag{78}$$

By the definition of the loss function (21), we have

$$\mathcal{L}_t(f^\star) - \mathcal{L}_t(f) = -\sum_{i=1}^{t-1} \sum_{h=1}^{H} \sum_{n=1}^{N} \left( X_{i,h}^f + Y_{i,h,n}^f \right). \tag{79}$$

Let $D_{\mathsf{H}}^2(\cdot\|\cdot)$ denote the Hellinger divergence defined as:

$$D_{\mathsf{H}}^2(P\|Q) := \frac{1}{2} \int_{\mathcal{X}} \left( \sqrt{P(x)} - \sqrt{Q(x)} \right)^2 dx \tag{80}$$

for any probability measures $P$ and $Q$ on $\mathcal{X}$, and define

$$\ell(f_h, s, \boldsymbol{a}) := D_{\mathsf{H}}^2 \left( \mathbb{P}_{f,h}(\cdot|s, \boldsymbol{a}) \| \mathbb{P}_h(\cdot|s, \boldsymbol{a}) \right). \tag{81}$$

In the following lemma we provide a concentration result for the random variables $X_{t,h}^f$ and $Y_{t,h,n}^f$ in (79) (recall we define $\widetilde{\boldsymbol{\pi}}_{t,n} := (\widetilde{\pi}_t^n, \boldsymbol{\pi}_t^{-n})$ in (73)), where the state-action visitation distribution $d_h^{\boldsymbol{\pi}}(\rho) \in \Delta(\mathcal{S} \times \mathcal{A})$ at step $h$ under the policy $\boldsymbol{\pi}$ and the initial state distribution $\rho$ is defined as

$$d_h^{\boldsymbol{\pi}}(s, a; \rho) := \mathbb{E}_{s_1 \sim \rho} \mathbb{P}^{\boldsymbol{\pi}}(s_h = s, \boldsymbol{a}_h = \boldsymbol{a}|s_1). \tag{82}$$

**Lemma B.8.** *When Assumptions 3.1 and 3.2 hold, for any $\delta \in (0,1)$, with probability at least $1 - \delta$, we have for all $t \in [T]$, $f \in \mathcal{F}$ and $n \in [N]$:*

$$-\sum_{i=1}^{t-1}\sum_{h=1}^{H} X_{i,h}^f \le -2\sum_{i=1}^{t-1}\sum_{h=1}^{H}\mathbb{E}_{(s_{i,h},\boldsymbol{a}_{i,h})\sim d_h^{\boldsymbol{\pi}_i}(\rho)}\left[\ell(f_h, s_{i,h}, \boldsymbol{a}_{i,h})\right]$$
$$+ 2\sqrt{2}H + 2H\log\left(\frac{(N+1)HT}{\delta}\right) + 2dH\log\left(1 + 2\sqrt{d}|\mathcal{S}|^2 T^2\right). \tag{83}$$

$$-\sum_{i=1}^{t-1}\sum_{h=1}^{H} Y_{i,h,n}^f \le -2\sum_{i=1}^{t-1}\sum_{h=1}^{H}\mathbb{E}_{(s_{i,h}^n,\boldsymbol{a}_{i,h}^n)\sim d_h^{\tilde{\boldsymbol{\pi}}_{i,n}}(\rho)}\left[\ell(f_h, s_{i,h}^n, \boldsymbol{a}_{i,h}^n)\right]$$
$$+ 2\sqrt{2}H + 2H\log\left(\frac{(N+1)HT}{\delta}\right) + 2dH\log\left(1 + 2\sqrt{d}|\mathcal{S}|^2 T^2\right). \tag{84}$$

Combining (77), (79), (83), (84), we have with probability at least $1 - \delta$:

$$\text{(i)} \le -\frac{2}{N\alpha}\sum_{n=1}^{N}\bigg\{\sum_{t=1}^{T}\sum_{i=1}^{t-1}\sum_{h=1}^{H}\mathbb{E}_{(s_{i,h},\boldsymbol{a}_{i,h})\sim d_h^{\boldsymbol{\pi}_i}(\rho)}\left[\ell(f_{t,h}, s_{i,h}, \boldsymbol{a}_{i,h})\right]$$
$$+ \sum_{t=1}^{T}\sum_{i=1}^{t-1}\sum_{h=1}^{H}\mathbb{E}_{(s_{i,h}^n,\boldsymbol{a}_{i,h}^n)\sim d_h^{\tilde{\boldsymbol{\pi}}_{i,n}}(\rho)}\left[\ell(f_{t,h}, s_{i,h}^n, \boldsymbol{a}_{i,h}^n)\right]\bigg\}$$
$$+ \frac{4HT}{\alpha}\left(\sqrt{2} + \log\left(\frac{(N+1)HT}{\delta}\right) + d\log\left(1 + 2\sqrt{d}|\mathcal{S}|^2 T^2\right)\right). \tag{85}$$

**Step 2: bounding terms (ii), (iii) and (iv).** To bound (ii), (iii) and (iv), we introduce the following lemma.

**Lemma B.9.** *Under Assumptions 3.1 and 3.2, for any $n \in [N]$, $\beta \ge 0$, $\{\widehat{\boldsymbol{\pi}}_t : \mathcal{S} \times [H] \to \Delta(\mathcal{A})\}_{t \in [T]}$ and $\{\widehat{f}_t\}_{t \in [T]} \subset \mathcal{F}$, we have*

$$\sum_{t=1}^{T}\left|V_{\widehat{f}_t,n}^{\widehat{\boldsymbol{\pi}}_t}(\rho) - V_n^{\widehat{\boldsymbol{\pi}}_t}(\rho)\right| \le \frac{\eta}{2}\sum_{t=1}^{T}\sum_{i=1}^{t-1}\sum_{h=1}^{H}\mathbb{E}_{(s,\boldsymbol{a})\sim d_h^{\widehat{\boldsymbol{\pi}}_i}(\rho)}\ell(\widehat{f}_{t,h}, s, \boldsymbol{a})$$
$$+ H\left(\frac{4d_H(\lambda)H}{\eta} + \left(\sqrt{d} + H\right)\min\{d_H(\lambda), T\} + \sqrt{d}\lambda T\right) \tag{86}$$

*for any $\eta > 0$ and $\lambda > 0$, where $d_H(\lambda)$ is defined as*

$$d_H(\lambda) := 2d\log\left(1 + \frac{H^2 T}{\lambda}\right).$$

Now we are ready to bound (ii), (iii) and (iv). To bound (ii), letting $\widehat{f}_t = f_t$ and $\widehat{\boldsymbol{\pi}}_t = \widetilde{\boldsymbol{\pi}}_{t,n}$ for each $n \in [N]$ in Lemma B.9 (recall we define $\widetilde{\boldsymbol{\pi}}_{t,n} := (\widetilde{\pi}_t^n, \boldsymbol{\pi}_t^{-n})$ in (73)), we have for any $\eta > 0$:

$$\text{(ii)} \le \frac{\eta}{2N}\sum_{h=1}^{H}\sum_{n=1}^{N}\sum_{t=1}^{T}\sum_{i=1}^{t-1}\mathbb{E}_{(s,\boldsymbol{a})\sim d_h^{\tilde{\boldsymbol{\pi}}_{i,n}}(\rho)}\ell(f_{t,h}, s, \boldsymbol{a})$$
$$+ H\left(\frac{4d_H(\lambda)H}{\eta} + \left(\sqrt{d} + H\right)\min\{d_H(\lambda), T\} + \sqrt{d}\lambda T\right). \tag{87}$$

Letting $\widehat{f}_{t,h} = f_{t-1}$ and $\widehat{\boldsymbol{\pi}}_{t,h} = \widetilde{\boldsymbol{\pi}}_{t,n}$ for each $n \in [N]$ in Lemma B.9, we can bound (iii) as follows:

$$\text{(iii)} \le \frac{\eta}{2N}\sum_{h=1}^{H}\sum_{n=1}^{N}\sum_{t=1}^{T}\sum_{i=1}^{t-1}\mathbb{E}_{(s,\boldsymbol{a})\sim d_h^{\tilde{\boldsymbol{\pi}}_{i,n}}(\rho)}\ell(f_{t-1,h}, s, \boldsymbol{a})$$

$$+ H \left( \frac{4d_H(\lambda)H}{\eta} + \left( \sqrt{d} + H \right) \min\{d_H(\lambda), T\} + \sqrt{d}\lambda T \right). \tag{88}$$

To continue to bound the first term, note that

$$\sum_{h=1}^{H} \sum_{t=1}^{T} \sum_{i=1}^{t-1} \mathbb{E}_{(s,\boldsymbol{a}) \sim d_h^{\tilde{\pi}_{i,n}}(\rho)} \ell(f_{t-1,h}, s, \boldsymbol{a}) \leq \sum_{h=1}^{H} \sum_{t=1}^{T} \sum_{i=1}^{t-2} \mathbb{E}_{(s,\boldsymbol{a}) \sim d_h^{\tilde{\pi}_{i,n}}(\rho)} \ell(f_{t-1,h}, s, \boldsymbol{a}) + HT$$

$$= \sum_{h=1}^{H} \sum_{t=0}^{T-1} \sum_{i=1}^{t-1} \mathbb{E}_{(s,\boldsymbol{a}) \sim d_h^{\tilde{\pi}_{i,n}}(\rho)} \ell(f_{t,h}, s, \boldsymbol{a}) + HT$$

$$\leq \sum_{h=1}^{H} \sum_{t=1}^{T} \sum_{i=1}^{t-1} \mathbb{E}_{(s,\boldsymbol{a}) \sim d_h^{\tilde{\pi}_{i,n}}(\rho)} \ell(f_{t,h}, s, \boldsymbol{a}) + HT,$$

where the first inequality uses the fact that

$$\ell(f_h, s, \boldsymbol{a}) = D_{\mathsf{H}}^2 \left( \mathbb{P}_{f,h}(\cdot|s, \boldsymbol{a}) \| \mathbb{P}_h(\cdot|s, \boldsymbol{a}) \right) \leq 1, \tag{89}$$

the second line shifts the index of $t$ by 1, and the last line follows by noticing the first summand is 0 at $t = 0$. Plugging the above relation back to (88) leads to

$$\text{(iii)} \leq \frac{\eta}{2N} \sum_{h=1}^{H} \sum_{n=1}^{N} \sum_{t=1}^{T} \sum_{i=1}^{t-1} \mathbb{E}_{(s,\boldsymbol{a}) \sim d_h^{\tilde{\pi}_{i,n}}(\rho)} \ell(f_{t,h}, s, \boldsymbol{a})$$

$$+ H \left( \frac{4d_H(\lambda)H}{\eta} + \left( \sqrt{d} + H \right) \min\{d_H(\lambda), T\} + \sqrt{d}\lambda T + \frac{\eta}{2}T \right). \tag{90}$$

Finally, similar to (90), letting $\widehat{f}_{t,h} = f_{t-1}, \widehat{\boldsymbol{\pi}}_{t,h} = \boldsymbol{\pi}_t$ for each $n \in [N]$ and replace $\eta$ by $2\eta$ in Lemma B.9, we can bound (iv) as follows:

$$\text{(iv)} \leq \frac{\eta}{N} \sum_{h=1}^{H} \sum_{n=1}^{N} \sum_{t=1}^{T} \sum_{i=1}^{t-1} \mathbb{E}_{(s,\boldsymbol{a}) \sim d_h^{\pi_i}(\rho)} \ell(f_{t,h}, s, \boldsymbol{a})$$

$$+ H \left( \frac{2Hd_H(\lambda)}{\eta} + \left( \sqrt{d} + H \right) \min\{d_H(\lambda), T\} + \sqrt{d}\lambda T + \eta T \right). \tag{91}$$

**Step 3: combining the bounds.** Letting $\eta = \frac{2}{\alpha}$ in (87), (90) and (91), and adding (85), (87), (90) and (91) together, we have with probability at least $1 - \delta$:

$$\mathsf{Regret}(T) \leq \frac{4HT}{\alpha} \left( \sqrt{2} + \log \left( \frac{(N+1)HT}{\delta} \right) + d \log \left( 1 + 2\sqrt{d}|\mathcal{S}|^2 T^2 \right) \right)$$

$$+ H \left( 5\alpha d_H(\lambda)H + 3 \left( \sqrt{d} + H \right) \min\{d_H(\lambda), T\} + 3\sqrt{d}\lambda T + \frac{3}{\alpha}T \right).$$

By setting

$$\lambda = \sqrt{\frac{d}{T}}, \quad \alpha = \sqrt{\frac{\log \left( \frac{(N+1)HT}{\delta} \right) + d \log \left( 1 + 2\sqrt{d}|\mathcal{S}|^2 T^2 \right)}{Hd \log \left( 1 + \frac{H^2 T^{3/2}}{\sqrt{d}} \right)}} T, \tag{92}$$

in the above expression, we have with probability at least $1 - \delta$:

$$\mathsf{Regret}(T) \leq 4(1 + \sqrt{2}) \sqrt{\frac{d \log \left( 1 + \frac{H^2 T^{3/2}}{\sqrt{d}} \right)}{\log \left( \frac{(N+1)HT}{\delta} \right) + d \log \left( 1 + 2\sqrt{d}|\mathcal{S}|^2 T^2 \right)}} \cdot \sqrt{HT}$$

$$14d\sqrt{H^3 T} \cdot \sqrt{\left(\frac{1}{d}\log\left(\frac{(N+1)HT}{\delta}\right) + \log\left(1 + \sqrt{d}|\mathcal{S}|^2 T^2\right)\right)\log\left(1 + \frac{H^2 T^{3/2}}{\sqrt{d}}\right)}$$

$$+ 6H\left(\sqrt{d} + H\right)d\log\left(1 + \frac{H^2 T^{3/2}}{\sqrt{d}}\right) + 3dH\sqrt{T}, \tag{93}$$

which gives the desired result after simplifying the expression.

### B.3.1. PROOF OF LEMMA B.8

Same as in (58), for the parameter spaces $\Theta_h$, $h \in [H]$, by Assumption 3.1 and Lemma B.5 we have

$$\forall h \in [H]: \quad \log\mathcal{N}(\Theta_h, \epsilon, \|\cdot\|_2) \le d\log\left(1 + \frac{2\sqrt{d}}{\epsilon}\right) \tag{94}$$

for any $\epsilon > 0$. Thus for any $\epsilon > 0$, there exists an $\epsilon$-net $\Theta_{h,\epsilon}$ of $\Theta_h$ ($\Theta_{h,\epsilon} \subset \Theta_h$) such that $\log|\Theta_{h,\epsilon}| \le d\log\left(1 + \frac{2\sqrt{d}}{\epsilon}\right)$, $\forall h \in [H]$. Define

$$\mathcal{F}_{h,\epsilon} := \left\{ f_h \in \mathcal{F}_h : f_h(s, \boldsymbol{a}, s_{h+1}) = \phi_h(s, \boldsymbol{a}, s_{h+1})^\top \theta_h, \theta_h \in \Theta_{h,\epsilon} \right\}.$$

For any $f \in \mathcal{F}$, there exists $\theta_h \in \Theta_h$ such that $f_h(s, \boldsymbol{a}, s_{h+1}) = \phi_h(s, \boldsymbol{a}, s_{h+1})^\top \theta_h$. In addition, there exists $\theta_{h,\epsilon} \in \Theta_{h,\epsilon}$ such that $\|\theta_h - \theta_{h,\epsilon}\|_2 \le \epsilon$. We let $f_\epsilon(s, \boldsymbol{a}, s_{h+1}) = \phi_h(s, \boldsymbol{a}, s_{h+1})^\top \theta_{h,\epsilon}$. Then $f_\epsilon \in \mathcal{F}_{h,\epsilon}$, and we have

$$|\mathbb{P}_{f,h}(s_{h+1}|s, \boldsymbol{a}) - \mathbb{P}_{f_\epsilon,h}(s_{h+1}|s, \boldsymbol{a})| = |\phi_h(s, \boldsymbol{a}, s_{h+1})^\top (\theta_h - \theta_{h,\epsilon})| \le \epsilon, \tag{95}$$

from which we deduce

$$\forall t \in [T], h \in [H]: \quad -X_{t,h}^f \le -\log\left(\frac{\mathbb{P}_h(s_{t,h+1}|s_{t,h}, \boldsymbol{a}_{t,h})}{\mathbb{P}_{f_\epsilon,h}(s_{t,h+1}|s_{t,h}, \boldsymbol{a}_{t,h}) + \epsilon}\right) := -X_{t,h}^{f_\epsilon}(\epsilon). \tag{96}$$

Let $\mathcal{F}_t := \sigma(\mathcal{D}_t)$ be the $\sigma$-algebra generated by the dataset $\mathcal{D}_t$. By Lemma B.4 we have with probability at least $1 - \frac{\delta}{N+1}$:

$$\forall t \in [T], h \in [H], f_{h,\epsilon} \in \mathcal{F}_{h,\epsilon}: \quad -\frac{1}{2}\sum_{i=1}^{t-1} X_{i,h}^{f_\epsilon}(\epsilon) \le \sum_{i=1}^{t-1}\log\mathbb{E}\left[\exp\left(-\frac{1}{2}X_{i,h}^{f_\epsilon}(\epsilon)\right)\bigg|\mathcal{F}_{i-1}\right]$$

$$+ \log\left(\frac{(N+1)HT}{\delta}\right) + d\log\left(1 + \frac{2\sqrt{d}}{\epsilon}\right). \tag{97}$$

Then we have for all $t \in [T]$, $h \in [H]$ and $f \in \mathcal{F}$:

$$-\frac{1}{2}\sum_{i=1}^{t-1}\sum_{h=1}^{H} X_{i,h}^f \le -\frac{1}{2}\sum_{i=1}^{t-1}\sum_{h=1}^{H} X_{i,h}^{f_\epsilon}(\epsilon)$$

$$\le \sum_{i=1}^{t}\log\mathbb{E}\left[\exp\left(-\frac{1}{2}X_{i,h}^{f_\epsilon}(\epsilon)\right)\bigg|\mathcal{F}_{i-1}\right] + H\log\left(\frac{(N+1)HT}{\delta}\right) + dH\log\left(1 + \frac{2\sqrt{d}}{\epsilon}\right), \tag{98}$$

where the first line follows (96), and the second line follows from (97). The first term in the last line of (98) can be further bounded as follows:

$$\sum_{i=1}^{t}\log\mathbb{E}\left[\exp\left(-\frac{1}{2}X_{i,h}^{f_\epsilon}(\epsilon)\right)\bigg|\mathcal{F}_{i-1}\right]$$

$$= \sum_{i=1}^{t-1}\sum_{h=1}^{H}\log\mathbb{E}\left[\sqrt{\frac{\mathbb{P}_{f_\epsilon,h}(s_{i,h+1}|s_{i,h}, \boldsymbol{a}_{i,h}) + \epsilon}{\mathbb{P}_h(s_{i,h+1}|s_{i,h}, \boldsymbol{a}_{i,h})}}\bigg|\mathcal{F}_{s-1}\right]$$

$$
= \sum_{i=1}^{t-1} \sum_{h=1}^{H} \log \mathbb{E}_{\substack{(s_{i,h}, \boldsymbol{a}_{i,h}) \sim d_h^{\pi_i}(\rho), \\ s_{i,h+1} \sim \mathbb{P}_h(\cdot|s_{i,h}, \boldsymbol{a}_{i,h})}} \left[ \sqrt{\frac{\mathbb{P}_{f_\epsilon,h}(s_{i,h+1}|s_{i,h}, \boldsymbol{a}_{i,h}) + \epsilon}{\mathbb{P}_h(s_{i,h+1}|s_{i,h}, \boldsymbol{a}_{i,h})}} \right]
$$

$$
= \sum_{i=1}^{t-1} \sum_{h=1}^{H} \log \mathbb{E}_{(s_{i,h}, \boldsymbol{a}_{i,h}) \sim d_h^{\pi_i}(\rho)} \left[ \int_{\mathcal{S}} \sqrt{(\mathbb{P}_{f_\epsilon,h}(s_{i,h+1}|s_{i,h}, \boldsymbol{a}_{i,h}) + \epsilon) \mathbb{P}_h(s_{i,h+1}|s_{i,h}, \boldsymbol{a}_{i,h})} ds_{i,h+1} \right]
$$

$$
\leq \sum_{i=1}^{t-1} \sum_{h=1}^{H} \log \mathbb{E}_{(s_{i,h}, \boldsymbol{a}_{i,h}) \sim d_h^{\pi_i}(\rho)} \left[ \int_{\mathcal{S}} \sqrt{(\mathbb{P}_{f,h}(s_{i,h+1}|s_{i,h}, \boldsymbol{a}_{i,h}) + 2\epsilon) \mathbb{P}_h(s_{i,h+1}|s_{i,h}, \boldsymbol{a}_{i,h})} ds_{i,h+1} \right], \tag{99}
$$

where the last inequality uses (95). Furthermore, we have

$$
\mathbb{E}_{(s_{i,h}, \boldsymbol{a}_{i,h}) \sim d_h^{\pi_i}(\rho)} \left[ \int_{\mathcal{S}} \sqrt{(\mathbb{P}_{f,h}(s_{i,h+1}|s_{i,h}, \boldsymbol{a}_{i,h}) + 2\epsilon) \mathbb{P}_h(s_{i,h+1}|s_{i,h}, \boldsymbol{a}_{i,h})} ds_{i,h+1} \right]
$$

$$
\leq \mathbb{E}_{(s_{i,h}, \boldsymbol{a}_{i,h}) \sim d_h^{\pi_i}(\rho)} \left[ \int_{\mathcal{S}} \sqrt{\mathbb{P}_{f,h}(s_{i,h+1}|s_{i,h}, \boldsymbol{a}_{i,h}) \mathbb{P}_h(s_{i,h+1}|s_{i,h}, \boldsymbol{a}_{i,h})} ds_{i,h+1} \right]
$$

$$
+ \mathbb{E}_{(s_{i,h}, \boldsymbol{a}_{i,h}) \sim d_h^{\pi_i}(\rho)} \left[ \int_{\mathcal{S}} \sqrt{2\epsilon \mathbb{P}_h(s_{i,h+1}|s_{i,h}, \boldsymbol{a}_{i,h})} ds_{i,h+1} \right]
$$

$$
\leq 1 - \frac{1}{2} \mathbb{E}_{(s_{i,h}, \boldsymbol{a}_{i,h}) \sim d_h^{\pi_i}(\rho)} \left[ \int_{\mathcal{S}} \left( \sqrt{\mathbb{P}_{f,h}(s_{i,h+1}|s_{i,h}, \boldsymbol{a}_{i,h})} - \sqrt{\mathbb{P}_h(s_{i,h+1}|s_{i,h}, \boldsymbol{a}_{i,h})} \right)^2 ds_{i,h+1} \right] + \sqrt{2\epsilon}|\mathcal{S}|
$$

$$
= 1 - \mathbb{E}_{(s_{i,h}, \boldsymbol{a}_{i,h}) \sim d_h^{\pi_i}(\rho)} \left[ D_{\mathsf{H}}^2 \left( \mathbb{P}_{f,h}(\cdot|s_{i,h}, \boldsymbol{a}_{i,h}) \| \mathbb{P}_h(\cdot|s_{i,h}, \boldsymbol{a}_{i,h}) \right) \right] + \sqrt{2\epsilon}|\mathcal{S}|, \tag{100}
$$

where in the first inequality we use the fact that $\sqrt{a+b} \leq \sqrt{a} + \sqrt{b}$ for any $a, b \geq 0$, and the last line uses the definition of the Hellinger distance in (80).

Plugging (100) into (99), we have

$$
\sum_{i=1}^{t} \log \mathbb{E} \left[ \exp\left( -\frac{1}{2} X_{i,h}^{f_\epsilon}(\epsilon) \right) \Big| \mathcal{F}_{i-1} \right]
$$

$$
\leq \sum_{i=1}^{t-1} \sum_{h=1}^{H} \log \left( 1 - \mathbb{E}_{(s_{i,h}, \boldsymbol{a}_{i,h}) \sim d_h^{\pi_i}(\rho)} \left[ D_{\mathsf{H}}^2 \left( \mathbb{P}_{f,h}(\cdot|s_{i,h}, \boldsymbol{a}_{i,h}) \| \mathbb{P}_h(\cdot|s_{i,h}, \boldsymbol{a}_{i,h}) \right) \right] + \sqrt{2\epsilon}|\mathcal{S}| \right)
$$

$$
\leq -\sum_{i=1}^{t-1} \sum_{h=1}^{H} \mathbb{E}_{(s_{i,h}, \boldsymbol{a}_{i,h}) \sim d_h^{\pi_i}(\rho)} \left[ D_{\mathsf{H}}^2 \left( \mathbb{P}_{f,h}(\cdot|s_{i,h}, \boldsymbol{a}_{i,h}) \| \mathbb{P}_h(\cdot|s_{i,h}, \boldsymbol{a}_{i,h}) \right) \right] + \sqrt{2\epsilon}|\mathcal{S}|
$$

$$
= -\sum_{i=1}^{t-1} \sum_{h=1}^{H} \mathbb{E}_{(s_{i,h}, \boldsymbol{a}_{i,h}) \sim d_h^{\pi_i}(\rho)} \left[ \ell(f_h, s_{i,h}, \boldsymbol{a}_{i,h}) \right] + TH\sqrt{2\epsilon}|\mathcal{S}|,
$$

where the second inequality follows from $\log(x) \leq x - 1$ for any $x > 0$, and the last line follows the definition (81).

Plugging the above inequality into (98), we have with probability at least $1 - \frac{\delta}{N+1}$:

$$
\forall t \in [T], f \in \mathcal{F}: \quad -\sum_{i=1}^{t-1} \sum_{h=1}^{H} X_{i,h}^f \leq -2 \sum_{i=1}^{t-1} \sum_{h=1}^{H} \mathbb{E}_{(s_{i,h}, \boldsymbol{a}_{i,h}) \sim d_h^{\pi_i}(\rho)} \left[ \ell(f_h, s_{i,h}, \boldsymbol{a}_{i,h}) \right]
$$

$$
+ 2TH\sqrt{2\epsilon}|\mathcal{S}| + 2H \log\left( \frac{(N+1)HT}{\delta} \right) + 2dH \log\left( 1 + \frac{2\sqrt{d}}{\epsilon} \right). \tag{101}
$$

Then analogous to (101), we can bound $-\sum_{i=1}^{t-1} \sum_{h=1}^{H} Y_{i,h,n}^f$ for all $n \in [N]$ with probability at least $1 - \frac{N\delta}{N+1}$ as follows:

$$
\forall t \in [T], f \in \mathcal{F}, n \in [N]: \quad -\sum_{i=1}^{t-1} \sum_{h=1}^{H} Y_{i,h,n}^f \leq -2 \sum_{i=1}^{t-1} \sum_{h=1}^{H} \mathbb{E}_{(s_{i,h}^n, \boldsymbol{a}_{i,h}^n) \sim d_h^{\tilde{\pi}_{i,n}}(\rho)} \left[ \ell(f_h, s_{i,h}^n, \boldsymbol{a}_{i,h}^n) \right]
$$

$$+ 2TH\sqrt{2\epsilon}|\mathcal{S}| + 2H \log\left(\frac{(N+1)HT}{\delta}\right) + 2dH \log\left(1 + \frac{2\sqrt{d}}{\epsilon}\right). \tag{102}$$

Letting $\epsilon = \frac{1}{T^2|\mathcal{S}|^2}$ in (101) and (102), we obtain (83) and (84) in Lemma B.8.

### B.3.2. PROOF LEMMA B.9

To prove Lemma B.9, we first express the value difference sum $\sum_{t=1}^T \left| V_{\widehat{f}_{t,n}}^{\widehat{\pi}_t}(\rho) - V_n^{\widehat{\pi}_t}(\rho) \right|$ on the left hand side of (86) as sum of the expectation of the model estimation errors $\mathcal{E}_n^{\widehat{\pi}_t}(\widehat{f}_{t,h}, s_h, \boldsymbol{a}_h)$.

**Step 1: reformulating the value difference sum.** For any $f \in \mathcal{F}$ and $\boldsymbol{\pi} = (\pi^1, \cdots, \pi^N) : \mathcal{S} \times [H] \rightarrow \Delta(\mathcal{A})$, we have (recall we defined the state-action visitation distribution $d_h^{\boldsymbol{\pi}}(\rho)$ in (82)) for $n \in [N]$:

$$
\begin{aligned}
V_{f,n}^{\boldsymbol{\pi}}(\rho) &= \mathbb{E}_{\substack{\forall h \in [H]:(s_h,\boldsymbol{a}_h) \sim d_h^{\boldsymbol{\pi}}(\rho), \\ s_{h+1} \sim \mathbb{P}_h(\cdot|s_h,\boldsymbol{a}_h)}} \left[ \sum_{h=1}^H \left( V_{f,h,n}^{\boldsymbol{\pi}}(s_h) - V_{f,h+1,n}^{\boldsymbol{\pi}}(s_{h+1}) \right) \right] \\
&= \mathbb{E}_{\substack{\forall h \in [H]:(s_h,\boldsymbol{a}_h) \sim d_h^{\boldsymbol{\pi}}(\rho), \\ s_{h+1} \sim \mathbb{P}_h(\cdot|s_h,\boldsymbol{a}_h)}} \left[ \sum_{h=1}^H \left( Q_{f,h,n}^{\boldsymbol{\pi}}(s_h,\boldsymbol{a}_h) - \beta \log \frac{\pi^n(a_h^n|s_h^n)}{\pi_{\mathsf{ref}}^n(a_h^n|s_h^n)} - V_{f,h+1,n}^{\boldsymbol{\pi}}(s_{h+1}) \right) \right],
\end{aligned} \tag{103}
$$

where in the second line we use the fact that

$$V_{f,h,n}^{\boldsymbol{\pi}}(s) = \mathbb{E}_{\boldsymbol{a} \sim \boldsymbol{\pi}(\cdot|s)} \left[ Q_{f,h,n}^{\boldsymbol{\pi}}(s,\boldsymbol{a}) - \beta \log \frac{\pi^n(a^n|s^n)}{\pi_{\mathsf{ref}}^n(a^n|s^n)} \right].$$

By the definition of $V_n^{\boldsymbol{\pi}}$ we have

$$\forall n \in [N]: \quad V_n^{\boldsymbol{\pi}}(\rho) = \mathbb{E}_{\substack{\forall h \in [H]:(s_h,\boldsymbol{a}_h) \sim d_h^{\boldsymbol{\pi}}(\rho), \\ s_{h+1} \sim \mathbb{P}_h(\cdot|s_h,\boldsymbol{a}_h)}} \sum_{h=1}^H \left[ r_h^n(s_h,\boldsymbol{a}_h) - \beta \log \frac{\pi^n(a_h^n|s_h^n)}{\pi_{\mathsf{ref}}^n(a_h^n|s_h^n)} \right]. \tag{104}$$

To simplify the notation, we define

$$\forall g \in \mathcal{F}, h \in [H]: \quad \mathbb{P}_{g,h} V_{f,h+1,n}^{\boldsymbol{\pi}}(s_h,\boldsymbol{a}_h) := \mathbb{E}_{s_{h+1} \sim \mathbb{P}_{g,h}(\cdot|s_h,\boldsymbol{a}_h)} \left[ V_{f,h+1,n}^{\boldsymbol{\pi}}(s_{h+1}) \right]. \tag{105}$$

Combining (103) and (104), we have

$$
\begin{aligned}
V_{f,n}^{\boldsymbol{\pi}}(\rho) - V_n^{\boldsymbol{\pi}}(\rho) &= \mathbb{E}_{\substack{\forall h \in [H]:(s_h,\boldsymbol{a}_h) \sim d_h^{\boldsymbol{\pi}}(\rho), \\ s_{h+1} \sim \mathbb{P}_h(\cdot|s_h,\boldsymbol{a}_h)}} \left[ \sum_{h=1}^H \left( Q_{f,n}^{\boldsymbol{\pi}}(s_h,\boldsymbol{a}_h) - r_h^n(s_h,\boldsymbol{a}_h) - V_{f,h+1,n}^{\boldsymbol{\pi}}(s_{h+1}) \right) \right] \\
&= \sum_{h=1}^H \mathbb{E}_{(s_h,\boldsymbol{a}_h) \sim d_h^{\boldsymbol{\pi}}(\rho)} [\underbrace{\mathbb{P}_{f,h} V_{f,h+1,n}^{\boldsymbol{\pi}}(s_h,\boldsymbol{a}_h) - \mathbb{P}_h V_{f,h+1,n}^{\boldsymbol{\pi}}(s_h,\boldsymbol{a}_h)}_{=: \mathcal{E}_n^{\boldsymbol{\pi}}(f_h, s_h, \boldsymbol{a}_h)}].
\end{aligned} \tag{106}
$$

Therefore, we can express the value difference sum $\sum_{t=1}^T \left| V_{\widehat{f}_{t,n}}^{\widehat{\pi}_t}(\rho) - V_n^{\widehat{\pi}_t}(\rho) \right|$ as sum of the expectation of the model estimation errors $\mathcal{E}_n^{\widehat{\pi}_t}(\widehat{f}_{t,h}, s_h, \boldsymbol{a}_h)$:

$$\sum_{t=1}^T \left| V_{\widehat{f}_{t,n}}^{\widehat{\pi}_t}(\rho) - V_n^{\widehat{\pi}_t}(\rho) \right| = \sum_{t=1}^T \sum_{h=1}^H \left| \mathbb{E}_{(s,\boldsymbol{a}) \sim d_h^{\widehat{\pi}_t}(\rho)} \left[ \mathcal{E}_n^{\widehat{\pi}_t}(\widehat{f}_{t,h}, s, \boldsymbol{a}) \right] \right|. \tag{107}$$

Thus we only need to bound the right-hand side of (107).

**Step 2: bounding the sum of model estimation errors.**

By Assumption 3.1, there exist $\theta_{f,h}$ and $\theta_h^\star$ in $\Theta_h$ such that $f_h(s_{h+1}|s_h,\boldsymbol{a}_h) = \phi_h(s_h,\boldsymbol{a}_h,s_{h+1})^\top \theta_{f,h}$ and $\mathbb{P}_h(s_{h+1}|s_h,\boldsymbol{a}_h) = \phi_h(s_h,\boldsymbol{a}_h,s_{h+1})^\top \theta_h^\star$ for all $h \in [H]$. Thus we have

$$\mathbb{E}_{(s_h,\boldsymbol{a}_h) \sim d_h^{\boldsymbol{\pi}}(\rho)} [\mathcal{E}_n^{\boldsymbol{\pi}}(f_h, s_h, \boldsymbol{a}_h)] = (\theta_{f,h} - \theta_h^\star)^\top \underbrace{\mathbb{E}_{(s_h,\boldsymbol{a}_h) \sim d_h^{\boldsymbol{\pi}}(\rho)} \left[ \int_{\mathcal{S}} \phi_h(s_h,\boldsymbol{a}_h,s_{h+1}) V_{f,h+1,n}^{\boldsymbol{\pi}}(s_{h+1}) ds_{h+1} \right]}_{=: x_{h,n}(f,\boldsymbol{\pi})}. \tag{108}$$

We let $x_{h,n}^i(f, \boldsymbol{\pi})$ denote the $i$-th component of $x_{h,n}(f, \boldsymbol{\pi})$, i.e.,

$$x_{h,n}^i(f, \boldsymbol{\pi}) = \mathbb{E}_{(s_h, \boldsymbol{a}_h) \sim d_h^{\boldsymbol{\pi}}(\rho)} \left[ \int_{\mathcal{S}} \phi_h^i(s_h, \boldsymbol{a}_h, s_{h+1}) V_{f,h+1,n}^{\boldsymbol{\pi}}(s_{h+1}) ds_{h+1} \right].$$

Then we have

$$\forall i \in [d]: \quad |x_{h,n}^i(f, \boldsymbol{\pi})| \leq H \tag{109}$$

(recall that by the definition of the linear mixture model (c.f. Assumption 3.1), $\phi_h^i(s, \boldsymbol{a}, \cdot) \in \Delta(\mathcal{S})$ for all $i \in [d]$ and $(s, \boldsymbol{a}) \in \mathcal{S} \times \mathcal{A}$), which gives

$$\|x_{h,n}(f, \boldsymbol{\pi})\|_2 \leq H\sqrt{d}. \tag{110}$$

For each $t \in [T]$, we define $\Lambda_{t,h} \in \mathbb{R}^{d \times d}$ as

$$\Lambda_{t,h} := \lambda I_d + \sum_{i=1}^{t-1} x_{h,n}(\widehat{f}_i, \widehat{\boldsymbol{\pi}}_i) x_{h,n}(\widehat{f}_i, \widehat{\boldsymbol{\pi}}_i)^\top. \tag{111}$$

We can decompose the sum of model estimation errors as follows:

$$\sum_{t=1}^T \left| \mathbb{E}_{(s,\boldsymbol{a}) \sim d_h^{\widehat{\boldsymbol{\pi}}_t}(\rho)} \left[ \mathcal{E}_n^{\widehat{\boldsymbol{\pi}}_t}(\widehat{f}_{t,h}, s, \boldsymbol{a}) \right] \right| = \underbrace{\sum_{t=1}^T \left| \langle x_{h,n}(\widehat{f}_t, \widehat{\boldsymbol{\pi}}_t), \widehat{\theta}_{t,h} - \theta_h^\star \rangle \right| \mathbb{1} \left\{ \left\| x_{h,n}(\widehat{f}_t, \widehat{\boldsymbol{\pi}}_t) \right\|_{\Lambda_{t,h}^{-1}} \leq 1 \right\}}_{(a)}$$

$$+ \underbrace{\sum_{t=1}^T \left| \langle x_{h,n}(\widehat{f}_t, \widehat{\boldsymbol{\pi}}_t), \widehat{\theta}_{t,h} - \theta_h^\star \rangle \right| \mathbb{1} \left\{ \left\| x_{h,n}(\widehat{f}_t, \widehat{\boldsymbol{\pi}}_t) \right\|_{\Lambda_{t,h}^{-1}} > 1 \right\}}_{(b)}. \tag{112}$$

Below we bound (a) and (b) respectively.

**Step 1: bounding term (a).** By the Cauchy-Schwarz inequality, we have

$$(a) \leq \sum_{t=1}^T \left\| \widehat{\theta}_{t,h} - \theta_h^\star \right\|_{\Lambda_{t,h}} \left\| x_{h,n}(\widehat{f}_t, \widehat{\boldsymbol{\pi}}_t) \right\|_{\Lambda_{t,h}^{-1}} \mathbb{1} \left\{ \left\| x_{h,n}(\widehat{f}_t, \widehat{\boldsymbol{\pi}}_t) \right\|_{\Lambda_{t,h}^{-1}} \leq 1 \right\}$$

$$\leq \sum_{t=1}^T \left\| \widehat{\theta}_{t,h} - \theta_h^\star \right\|_{\Lambda_{t,h}} \min \left\{ \left\| x_{h,n}(\widehat{f}_t, \widehat{\boldsymbol{\pi}}_t) \right\|_{\Lambda_{t,h}^{-1}}, 1 \right\}, \tag{113}$$

where the last inequality uses the fact that

$$\left\| x_{h,n}(\widehat{f}_t, \widehat{\boldsymbol{\pi}}_t) \right\|_{\Lambda_{t,h}^{-1}} \mathbb{1} \left\{ \left\| x_{h,n}(\widehat{f}_t, \widehat{\boldsymbol{\pi}}_t) \right\|_{\Lambda_{t,h}^{-1}} \leq 1 \right\} \leq \min \left\{ \left\| x_{h,n}(\widehat{f}_t, \widehat{\boldsymbol{\pi}}_t) \right\|_{\Lambda_{t,h}^{-1}}, 1 \right\}.$$

By the definition of $\Lambda_{t,h}$ (c.f. (111)) and Assumption 3.1 we have

$$\left\| \widehat{\theta}_{t,h} - \theta_h^\star \right\|_{\Lambda_{t,h}} \leq 2\sqrt{\lambda d} + \left( \sum_{i=1}^{t-1} |\langle \widehat{\theta}_{t,h} - \theta_h^\star, x_{h,n}(\widehat{f}_i, \widehat{\boldsymbol{\pi}}_i) \rangle|^2 \right)^{1/2}, \tag{114}$$

which gives

$$\sum_{t=1}^T \left\| \widehat{\theta}_{t,h} - \theta_h^\star \right\|_{\Lambda_{t,h}} \min \left\{ \left\| x_{h,n}(\widehat{f}_t, \widehat{\boldsymbol{\pi}}_t) \right\|_{\Lambda_{t,h}^{-1}}, 1 \right\}$$

$$\leq \sum_{t=1}^{T} \left( 2\sqrt{\lambda d} + \left( \sum_{i=1}^{t-1} |\langle \widehat{\theta}_{t,h} - \theta_h^\star, x_{h,n}(\widehat{f}_i, \widehat{\boldsymbol{\pi}}_i)\rangle|^2 \right)^{1/2} \right) \cdot \min \left\{ \left\| x_{h,n}(\widehat{f}_t, \widehat{\boldsymbol{\pi}}_t) \right\|_{\Lambda_{t,h}^{-1}}, 1 \right\}$$

$$\leq \left( \sum_{t=1}^{T} 4\lambda d \right)^{1/2} \left( \sum_{t=1}^{T} \min \left\{ \left\| x_{h,n}(\widehat{f}_t, \widehat{\boldsymbol{\pi}}_t) \right\|_{\Lambda_{t,h}^{-1}}, 1 \right\} \right)^{1/2}$$

$$+ \left( \sum_{t=1}^{T} \sum_{i=1}^{t-1} |\langle \widehat{\theta}_{t,h} - \theta_h^\star, x_{h,n}(\widehat{f}_i, \widehat{\boldsymbol{\pi}}_i)\rangle|^2 \right)^{1/2} \left( \sum_{t=1}^{T} \min \left\{ \left\| x_{h,n}(\widehat{f}_t, \widehat{\boldsymbol{\pi}}_t) \right\|_{\Lambda_{t,h}^{-1}}, 1 \right\} \right)^{1/2}, \tag{115}$$

where the first inequality uses (67) and the second inequality uses the Cauchy-Schwarz inequality and the fact that

$$\min \left\{ \left\| x_{h,n}(\widehat{f}_t, \widehat{\boldsymbol{\pi}}_t) \right\|_{\Lambda_{t,h}^{-1}}, 1 \right\}^2 \leq \min \left\{ \left\| x_{h,n}(\widehat{f}_t, \widehat{\boldsymbol{\pi}}_t) \right\|_{\Lambda_{t,h}^{-1}}, 1 \right\}.$$

By Lemma B.2, Lemma B.3 and (110), we have

$$\sum_{i=1}^{t} \min \left\{ \left\| x_{h,n}(\widehat{f}_i, \widehat{\boldsymbol{\pi}}_i) \right\|_{\Lambda_{i,h}^{-1}}, 1 \right\} \leq 2d \log \left( 1 + \frac{H^2 T}{\lambda} \right) := d_H(\lambda) \tag{116}$$

holds for any $\lambda > 0$ and $t \in [T]$. By (116), (115) and (113), we have

$$(a) \leq 2\sqrt{\lambda dT \min\{d_H(\lambda), T\}} + \left( d_H(\lambda) \sum_{t=1}^{T} \sum_{i=1}^{t-1} |\langle \widehat{\theta}_{t,h} - \theta_h^\star, x_{h,n}(\widehat{f}_i, \widehat{\boldsymbol{\pi}}_i)\rangle|^2 \right)^{1/2}. \tag{117}$$

To continue, we have

$$|\langle \widehat{\theta}_{t,h} - \theta_h^\star, x_{h,n}(\widehat{f}_i, \widehat{\boldsymbol{\pi}}_i)\rangle|^2 = \left| \mathbb{E}_{(s,\boldsymbol{a}) \sim d_h^{\widehat{\boldsymbol{\pi}}_i}(\rho)} \left[ \int_{\mathcal{S}} \left( \mathbb{P}_{\widehat{f}_{t,h}}(s_{h+1}|s,\boldsymbol{a}) - \mathbb{P}_h(s_{h+1}|s,\boldsymbol{a}) \right) V_{\widehat{f}_i,h+1,n}^{\widehat{\boldsymbol{\pi}}_i}(s_{h+1}) ds_{h+1} \right] \right|^2$$

$$\leq \mathbb{E}_{(s,\boldsymbol{a}) \sim d_h^{\widehat{\boldsymbol{\pi}}_i}(\rho)} \left[ \left( \int_{\mathcal{S}} \left( \mathbb{P}_{\widehat{f}_{t,h}}(s_{h+1}|s,\boldsymbol{a}) - \mathbb{P}_h(s_{h+1}|s,\boldsymbol{a}) \right) V_{\widehat{f}_i,h+1,n}^{\widehat{\boldsymbol{\pi}}_i}(s_{h+1}) ds_{h+1} \right)^2 \right]$$

$$\leq 4 \left\| V_{\widehat{f}_i,h+1,n}^{\widehat{\boldsymbol{\pi}}_i}(\cdot) \right\|_\infty \mathbb{E}_{(s,\boldsymbol{a}) \sim d_h^{\widehat{\boldsymbol{\pi}}_t}(\rho)} D_{\mathsf{TV}}^2 \left( \mathbb{P}_{\widehat{f}_{t,h}}(\cdot|s,\boldsymbol{a}) \| \mathbb{P}_h(\cdot|s,\boldsymbol{a}) \right)$$

$$\leq 8H \mathbb{E}_{(s,\boldsymbol{a}) \sim d_h^{\widehat{\boldsymbol{\pi}}_i}(\rho)} D_{\mathsf{H}}^2 \left( \mathbb{P}_{\widehat{f}_{t,h}}(\cdot|s,\boldsymbol{a}) \| \mathbb{P}_h(\cdot|s,\boldsymbol{a}) \right)$$

$$= 8H \mathbb{E}_{(s,\boldsymbol{a}) \sim d_h^{\widehat{\boldsymbol{\pi}}_i}(\rho)} \ell(\widehat{f}_{t,h}, s, \boldsymbol{a}), \tag{118}$$

where the second line uses the Cauchy-Schwarz inequality, the third line follows from Hölder's inequality, and $D_{\mathsf{TV}}$ denote the TV distance:

$$D_{\mathsf{TV}}(P\|Q) := \frac{1}{2} \int_{\mathcal{X}} |P(x) - Q(x)| dx. \tag{119}$$

The fourth line uses the following inequality:

$$D_{\mathsf{TV}}^2(P\|Q) \leq 2 D_{\mathsf{H}}^2(P\|Q),$$

and the fact that $\left\| V_{\widehat{f}_t,h+1,n}^{\widehat{\boldsymbol{\pi}}_{t,h}}(\cdot) \right\|_\infty \leq H$ (recall we assume $r(s,\boldsymbol{a}) \in [0,1]$). The last line uses (81).

Plugging (118) into (117), we have

$$(a) \leq 2\sqrt{d} \cdot \sqrt{\lambda T \min\{d_H(\lambda), T\}} + \left( 8H d_H(\lambda) \sum_{t=1}^{T} \sum_{i=1}^{t-1} \mathbb{E}_{(s,\boldsymbol{a}) \sim d_h^{\widehat{\boldsymbol{\pi}}_i}(\rho)} \ell(\widehat{f}_{t,h}, s, \boldsymbol{a}) \right)^{1/2}. \tag{120}$$

**Step 2: bounding term (b).** Now we bound (b) in (112). Note that

$$\mathbb{1}\left\{\left\|x_{h,n}(\widehat{f}_t, \widehat{\boldsymbol{\pi}}_t)\right\|_{\Lambda_{t,h}^{-1}} > 1\right\} \le \min\left\{\left\|x_{h,n}(\widehat{f}_t, \widehat{\boldsymbol{\pi}}_t)\right\|_{\Lambda_{t,h}^{-1}}, 1\right\},$$

which gives

$$(b) \le \sum_{t=1}^T \left|\langle x_{h,n}(\widehat{f}_t, \widehat{\boldsymbol{\pi}}_t), \widehat{\theta}_{t,h} - \theta_h^\star\rangle\right| \min\left\{\left\|x_{h,n}(\widehat{f}_t, \widehat{\boldsymbol{\pi}}_t)\right\|_{\Lambda_{t,h}^{-1}}, 1\right\}. \tag{121}$$

We also have

$$\left|\langle x_{h,n}(\widehat{f}_t, \widehat{\boldsymbol{\pi}}_t), \widehat{\theta}_{t,h} - \theta_h^\star\rangle\right| = \left|\mathbb{E}_{(s,\boldsymbol{a})\sim d_h^{\widehat{\boldsymbol{\pi}}_t}(\rho)}\left[\mathbb{P}_{\widehat{f}_{t,h}} V_{\widehat{f}_t,h+1,n}^{\widehat{\boldsymbol{\pi}}_{t,h}}(s,\boldsymbol{a}) - \mathbb{P}_h V_{\widehat{f}_t,h+1,n}^{\widehat{\boldsymbol{\pi}}_{t,h}}(s,\boldsymbol{a})\right]\right| \le H. \tag{122}$$

Combining the (122),(116) with (121), we have

$$(b) \le H \min\{T, d_H(\lambda)\}. \tag{123}$$

**Step 3: combining everything together.** Plugging (120) and (123) into (112), we have

$$\sum_{t=1}^T \left|\mathbb{E}_{(s,\boldsymbol{a})\sim d_h^{\widehat{\boldsymbol{\pi}}_t}(\rho)}\left[\mathcal{E}_n^{\widehat{\boldsymbol{\pi}}_t}(\widehat{f}_{t,h}, s, \boldsymbol{a})\right]\right|$$

$$\le 2\sqrt{d}\cdot\sqrt{\lambda T \min\{d_H(\lambda), T\}} + \left(8 H d_H(\lambda)\sum_{t=1}^T\sum_{i=1}^{t-1}\mathbb{E}_{(s,\boldsymbol{a})\sim d_h^{\widehat{\boldsymbol{\pi}}_i}(\rho)}\ell(\widehat{f}_{t,h}, s, \boldsymbol{a})\right)^{1/2} + H\min\{T, d_H(\lambda)\}$$

$$\le \left(\frac{8 H d_H(\lambda)}{\eta}\cdot\eta\sum_{t=1}^T\sum_{i=1}^{t-1}\mathbb{E}_{(s,\boldsymbol{a})\sim d_h^{\widehat{\boldsymbol{\pi}}_i}(\rho)}\ell(\widehat{f}_{t,h}, s, \boldsymbol{a})\right)^{1/2} + \left(\sqrt{d} + H\right)\min\{d_H(\lambda), T\} + \sqrt{d}\lambda T$$

$$\le \frac{4 d_H(\lambda) H}{\eta} + \frac{\eta}{2}\sum_{t=1}^T\sum_{i=1}^{t-1}\mathbb{E}_{(s,\boldsymbol{a})\sim d_h^{\widehat{\boldsymbol{\pi}}_i}(\rho)}\ell(\widehat{f}_{t,h}, s, \boldsymbol{a}) + \left(\sqrt{d} + H\right)\min\{d_H(\lambda), T\} + \sqrt{d}\lambda T$$

for any $\eta > 0$, where the second and third inequalities both use the fact that $\sqrt{ab} \le \frac{a+b}{2}$ for any $a, b \ge 0$.

Finally, combining (107) with the above inequality, we have (86).

## C. Extension to the Infinite-horizon Setting

In this section, we consider the $N$-player general-sum episodic Markov game with infinite horizon denoted as $\mathcal{M}_{\mathbb{P}} \coloneqq (\mathcal{S}, \mathcal{A}, \mathbb{P}, r, \gamma)$ as a generalization of the finite-horizon case in the main paper, where $\gamma \in [0, 1)$ is the discounted factor, and $\mathbb{P} : \mathcal{S} \times \mathcal{A} \to \Delta(\mathcal{S})$ is the homogeneous transition kernel: the probability of transitioning from state $s$ to state $s'$ by the action $\boldsymbol{a} = (a^1, \cdots, a^n)$ is $\mathbb{P}(s'|s, \boldsymbol{a})$. For the infinite horizon case, the KL-regularized value function is defined as

$$\forall s \in \mathcal{S}: \quad V_n^{\boldsymbol{\pi}}(s) \coloneqq \mathbb{E}_{\mathbb{P}, \boldsymbol{\pi}}\left[\sum_{h=0}^\infty \gamma^h\left(r^n(s_h, \boldsymbol{a}_h) - \beta\log\frac{\pi^n(a_h|s_h)}{\pi_{\text{ref}}^n(a_h|s_h)}\right)\Bigg| s_0 = s\right]$$

$$= \frac{1}{1-\gamma}\mathbb{E}_{(\bar{s}, \bar{\boldsymbol{a}})\sim d^{\boldsymbol{\pi}}(s)}\left[r^n(\bar{s}, \bar{\boldsymbol{a}}) - \beta\log\frac{\pi^n(\bar{a}^n|\bar{s})}{\pi_{\text{ref}}^n(\bar{a}^n|\bar{s})}\right], \tag{124}$$

where $s_h$ and $\boldsymbol{a}_h$ are the state and action at timestep $h$, respectively, $d^{\boldsymbol{\pi}}(s) \in \Delta(\mathcal{S} \times \mathcal{A})$ is the *discounted state-action visitation distribution* under policy $\boldsymbol{\pi}$ starting from state $s$:

$$d_{\bar{s}, \bar{\boldsymbol{a}}}^{\boldsymbol{\pi}}(s) \coloneqq (1-\gamma)\sum_{h=0}^\infty \gamma^h \mathbb{P}^{\boldsymbol{\pi}}(s_h = \bar{s}, \boldsymbol{a}_h = \bar{\boldsymbol{a}}|s_0 = s). \tag{125}$$

We assume $\rho \in \Delta(\mathcal{S})$ is the initial state distribution, i.e. $s_0 \sim \rho$. We define $d^{\boldsymbol{\pi}}(\rho) := \mathbb{E}_{s_0 \sim \rho}[d^{\boldsymbol{\pi}}(s_0)]$ as the discounted state-action visitation distribution under policy $\boldsymbol{\pi}$ starting from the initial state distribution $\rho$. The KL-regularized Q-function is defined as

$$\forall (s, \boldsymbol{a}) \in \mathcal{S} \times \mathcal{A}: \quad Q_n^{\boldsymbol{\pi}}(s, \boldsymbol{a}) := r^n(s, \boldsymbol{a}) + \gamma \mathbb{E}_{s' \sim \mathbb{P}(\cdot|s, \boldsymbol{a})}[V_n^{\boldsymbol{\pi}}(s')]. \tag{126}$$

We let $\mathcal{F}$ denote the function class of the estimators of the transition kernel of the Markov game, and we denote $f = \mathbb{P}_f \in \mathcal{F}$. Without otherwise specified, we assume the other notations and settings are the same as the finite-horizon case stated in Section 3.

## C.1. Algorithm development

The algorithm for solving the (KL-regularized) Markov game is shown in Algorithm 6, where in (128) we set the loss function at each iteration $t$ as the negative log-likelihood of the transition kernel estimator $f$:

$$\mathcal{L}_t(f) := \sum_{(s, \boldsymbol{a}, s') \in \mathcal{D}_{t-1}} -\log \mathbb{P}_f(s'|s, \boldsymbol{a}). \tag{127}$$

Except for the loss function, the main change in Algorithm 6 is that we need to sample the state-action pair $(s, \boldsymbol{a})$ from the discounted state-action visitation distribution $d^{\boldsymbol{\pi}}(\rho)$, and sample the next state $s'$ from the transition kernel $\mathbb{P}(\cdot|s, \boldsymbol{a})$, which can be done by calling Algorithm 7. Algorithm 7 is adapted from Algorithm 3 in Yuan et al. (2023), see also Algorithm 5 in Yang et al. (2024a). Algorithm 7 satisfies $\mathbb{E}[h + 1] = \frac{1}{1-\gamma}$, and $\mathbb{P}(s_h = s, \boldsymbol{a}_h = \boldsymbol{a}) = d^{\boldsymbol{\pi}}(\rho)$ (Yuan et al., 2023).

---

**Algorithm 6** Value-incentive Infinite-horizon Markov Game

1: **Input:** reference policies $\boldsymbol{\pi}_{\mathrm{ref}}$, KL coefficient $\beta$, initial state distribution $\rho$, initial transition kernel estimator $f_0 \in \mathcal{F}$, regularization coefficient $\alpha > 0$, iteration number $T$.
2: **Initialization:** dataset $\mathcal{D}_0^n := \emptyset, \forall n \in [N]$. $\mathcal{D}_0 = \cup_{n=1}^N \mathcal{D}_0^n$.
3: **for** $t = 1, \cdots, T$ **do**
4:     $\boldsymbol{\pi}_t \leftarrow \mathsf{Equilibrium}(\mathcal{M}_{f_{t-1}})$.                           ▷ $\mathsf{Equilibrium}(\mathcal{M}_f)$ returns a CCE or NE of game $\mathcal{M}_f$.
5:     Model update: Update the estimator $f_t$ by minimizing the following objective:

$$f_t = \underset{f \in \mathcal{F}}{\arg\min} \sum_{(s, \boldsymbol{a}, s') \in \mathcal{D}_{t-1}} -\log \mathbb{P}_f(s'|s, \boldsymbol{a}) - \alpha \sum_{n=1}^N V_{f,n}^{\star, \boldsymbol{\pi}_t^{-n}}(\rho). \tag{128}$$

6:     Compute best-response policies $\{\widetilde{\pi}_t^n\}_{n \in [N]}$:

$$\text{For all } n \in [N]: \quad \widetilde{\pi}_t^n = \underset{\pi^n \in \Delta(\mathcal{A}_n)}{\arg\max} V_{f_t, n}^{\pi^n, \boldsymbol{\pi}_t^{-n}}(\rho). \tag{129}$$

7:     Data collection: sample $(s_t, \boldsymbol{a}_t, s_t') \leftarrow \mathsf{Sampler}(\boldsymbol{\pi}_t, \rho)$. For all $n \in [N]$, sample $(s_t^n, \boldsymbol{a}_t^n, s_t^{n\prime}) \leftarrow \mathsf{Sampler}((\widetilde{\pi}_t^n, \boldsymbol{\pi}_t^{-n}), \rho)$, and update the dataset $\mathcal{D}_t^n = \mathcal{D}_{t-1}^n \cup \{(s_t, \boldsymbol{a}_t, s_t'), (s_t^n, \boldsymbol{a}_t^n, s_t^{n\prime})\}$. $\mathcal{D}_t = \cup_{n=1}^N \mathcal{D}_t^n$.    ▷ $\mathsf{Sampler}(\boldsymbol{\pi}, \rho)$ returns $(s, \boldsymbol{a}) \sim d^{\boldsymbol{\pi}}(\rho)$ and $s' \sim \mathbb{P}(\cdot|s, \boldsymbol{a})$, see Algorithm 7.
8: **end for**

---

## C.2. Theoretical guarantee

We first state our assumptions on the function class for Markov game with infinite horizon.

**Assumption C.1** (linear mixture model, infinite horizon). The function class $\mathcal{F}$ is

$$\mathcal{F} := \left\{ f | f(s, \boldsymbol{a}, s') = \phi(s, \boldsymbol{a}, s')^\top \theta, \forall (s, \boldsymbol{a}, s') \in \mathcal{S} \times \mathcal{A} \times \mathcal{S}, \theta \in \Theta \right\},$$

where $\phi : \mathcal{S} \times \mathcal{A} \times \mathcal{S} \to \mathbb{R}^d$ is the known feature map, $\|\phi(s, \boldsymbol{a}, s')\|_2 \leq 1$ for all $(s, \boldsymbol{a}, s')$, and $\Theta \subseteq \mathbb{B}_2^d(\sqrt{d})$. Moreover, for each $f \in \mathcal{F}$ and $(s, \boldsymbol{a}) \in \mathcal{S} \times \mathcal{A}$, $f(\cdot|s, \boldsymbol{a}) \in \Delta(\mathcal{S})$.

We also assume the function class $\mathcal{F}$ is expressive enough to describe the true transition kernel of the Markov game.

---

**Algorithm 7** Sampler for $(s, \boldsymbol{a}) \sim d^{\boldsymbol{\pi}}(\rho)$ and $s' \sim \mathbb{P}(\cdot|s, \boldsymbol{a})$

---

1: **Input:** policy $\boldsymbol{\pi}$, initial state distribution $\rho$, player index $n$.
2: **Initialization:** $s_0 \sim \rho$, $\boldsymbol{a}_0 \sim \boldsymbol{\pi}(\cdot|s_0)$, time step $h = 0$, variable $X \sim \text{Bernoulli}(\gamma)$.
3: **while** $X = 1$ **do**
4:    Sample $s_{h+1} \sim \mathbb{P}(\cdot|s_h, \boldsymbol{a}_h)$
5:    Sample $\boldsymbol{a}_{h+1} \sim \boldsymbol{\pi}(\cdot|s_{h+1})$
6:    $h \leftarrow h + 1$
7:    $X \sim \text{Bernoulli}(\gamma)$
8: **end while**
9: Sample $s_{h+1} \sim \mathbb{P}(\cdot|s_h, \boldsymbol{a}_h)$
10: Return $(s_h, \boldsymbol{a}_h, s_{h+1})$.

---

**Assumption C.2** (realizability). There exists $f^\star \in \mathcal{F}$ such that $\mathbb{P}_{f^\star} = \mathbb{P}$.

Theorem C.3 states our main result for the regret of the infinite-horizon online Markov game, whoes proof is deferred to Appendix C.3.

**Theorem C.3.** *Under Assumption C.1 and Assumption C.2, if setting the regularization coefficient $\alpha$ as*

$$\alpha = \frac{(1-\gamma)^{3/2}}{\gamma} \sqrt{\frac{\log\left(\frac{N}{\delta}\right) + d\log\left(d|\mathcal{S}|T\right)}{d\log\left(1 + \frac{T^{3/2}}{(1-\gamma)^2\sqrt{d}}\right)}} T,$$

*then for any $\beta \geq 0$, with any initial state distribution $\rho$, transition kernel estimator $f_0 \in \mathcal{F}$ and reference policy $\boldsymbol{\pi}_{\text{ref}}$, the regret of Algorithm 2 satisfies the following bound with probability at least $1 - \delta$ for any $\delta \in (0,1)$:*

$$\forall T \in \mathbb{N}_+ : \quad \text{Regret}(T) \leq \widetilde{\mathcal{O}}\left(\frac{\gamma d\sqrt{T}}{(1-\gamma)^{3/2}} \cdot \sqrt{\frac{1}{d}\log\left(\frac{N}{\delta}\right) + \log\left(d|\mathcal{S}|T\right)}\right). \tag{130}$$

Note that

$$\min_{t \in [T]} \text{Gap}\left(\boldsymbol{\pi}_t\right) \leq \frac{\text{Regret}(T)}{T}, \tag{131}$$

similar to earlier arguments, Theorem C.3 also implies an order of $\widetilde{\mathcal{O}}\left(\frac{\gamma^2 N d^2}{(1-\gamma)^4 \varepsilon^2}\right)$ sample complexity for Algorithm 2 to find an $\varepsilon$-NE or $\varepsilon$-CCE of $\mathcal{M}_{\mathbb{P}}$.

## C.3. Proof of Theorem C.3

The proof of Theorem C.3 resembles that of Theorem 3.3. We repeat some of the proof for clarity and completeness. Here we also have (76), and will upper bound each term in (76) separately.

**Step 1: bounding (i).** By Assumption C.2 we know that there exists $f^\star \in \mathcal{F}$ such that $f^\star := \mathbb{P} = \mathbb{P}_{f^\star}$

(77) also holds here, and we define random variables $X_t^f$ and $Y_t^f$ as

$$X_t^f := \log\left(\frac{\mathbb{P}(s_t'|s_t, \boldsymbol{a}_t)}{\mathbb{P}_f(s_t'|s_t, \boldsymbol{a}_t)}\right) \quad \text{and} \quad Y_{t,n}^f := \log\left(\frac{\mathbb{P}(s_t^{n\prime}|s_t^n, \boldsymbol{a}_t^n)}{\mathbb{P}_f(s_t^{n\prime}|s_t^n, \boldsymbol{a}_t^n)}\right), \quad \forall n \in [N]. \tag{132}$$

Then by the definition of the loss function (127), we have

$$\mathcal{L}_t(f^\star) - \mathcal{L}_t(f) = -\sum_{i=1}^{t-1}\sum_{n=1}^{N}\left(X_i^f + Y_{i,n}^f\right). \tag{133}$$

Same as in the proof of Theorem 3.3, we define

$$\widetilde{\boldsymbol{\pi}}_{t,n} := (\widetilde{\pi}_t^n, \boldsymbol{\pi}_t^{-n}), \quad \forall n \in [N]. \tag{134}$$

We also define

$$\ell(f, s, \boldsymbol{a}) := D_{\mathsf{H}}^2 \left( \mathbb{P}_f(\cdot|s, \boldsymbol{a}) \| \mathbb{P}(\cdot|s, \boldsymbol{a}) \right). \tag{135}$$

In the following lemma we provide a concentration result for the random variables $X_t^f$ and $Y_{t,n}^f$ in (133).

**Lemma C.4.** *When Assumption C.1,C.2 hold, for any $\delta \in (0, 1)$, with probability at least $1 - \delta$, we have*

$$\forall t \in [T], f \in \mathcal{F}: \quad -\sum_{i=1}^{t-1} X_i^f \leq -2 \sum_{i=1}^{t-1} \mathbb{E}_{(s_i, \boldsymbol{a}_i) \sim d^{\boldsymbol{\pi}_i}(\rho)} [\ell(f, s_i, \boldsymbol{a}_i)]$$
$$+ 2\sqrt{2} + 2 \log \left( \frac{(N+1)T}{\delta} \right) + 2d \log \left( 1 + 2\sqrt{d} T^2 |\mathcal{S}|^2 \right), \tag{136}$$

*and*

$$\forall t \in [T], f \in \mathcal{F}, n \in [N]: \quad -\sum_{i=1}^{t-1} Y_{i,n}^f \leq -2 \sum_{i=1}^{t-1} \mathbb{E}_{(s_i^n, \boldsymbol{a}_i^n) \sim d^{\tilde{\boldsymbol{\pi}}_{i,n}}(\rho)} [\ell(f, s_i^n, \boldsymbol{a}_i^n)]$$
$$+ 2\sqrt{2} + 2 \log \left( \frac{(N+1)T}{\delta} \right) + 2d \log \left( 1 + 2\sqrt{d} T^2 |\mathcal{S}|^2 \right). \tag{137}$$

The proof of Lemma C.4 is provided in Appendix C.3.1.

By (77), (133) and Lemma C.4, we have with probability at least $1 - \delta$:

$$\text{(i)} \leq -\frac{2}{N\alpha} \sum_{n=1}^{N} \left\{ \sum_{t=1}^{T} \sum_{i=1}^{t-1} \mathbb{E}_{(s_i, \boldsymbol{a}_i) \sim d^{\boldsymbol{\pi}_i}(\rho)} [\ell(f_t, s_i, \boldsymbol{a}_i)] + \sum_{t=1}^{T} \sum_{i=1}^{t-1} \mathbb{E}_{(s_i^n, \boldsymbol{a}_i^n) \sim d^{\tilde{\boldsymbol{\pi}}_{i,n}}(\rho)} [\ell(f_t, s_i^n, \boldsymbol{a}_i^n)] \right\}$$
$$+ \frac{4T}{\alpha} \left( \sqrt{2} + \log \left( \frac{(N+1)T}{\delta} \right) + d \log \left( 1 + \sqrt{d} |\mathcal{S}|^2 T^2 \right) \right). \tag{138}$$

**Step 2: bounding (ii), (iii) and (iv).** To bound (ii), (iii) and (iv), we introduce the following lemma.

**Lemma C.5.** *Under Assumption C.1 and Assumption C.2, for any $n \in [N]$, $\beta \geq 0$, $\{\widehat{\pi}_t : \mathcal{S} \to \Delta(\mathcal{A})\}_{t \in [T]}$ and $\{\widehat{f}_t\}_{t \in [T]} \subset \mathcal{F}$, we have*

$$\sum_{t=1}^{T} \left| V_{\widehat{f}_t, n}^{\widehat{\boldsymbol{\pi}}_t}(\rho) - V_n^{\widehat{\boldsymbol{\pi}}_t}(\rho) \right| \leq \frac{\gamma}{1 - \gamma} \left( \frac{\eta}{2} \sum_{t=1}^{T} \sum_{i=1}^{t-1} \mathbb{E}_{(s, \boldsymbol{a}) \sim d^{\widehat{\boldsymbol{\pi}}_i}(\rho)} \ell(\widehat{f}_t, s, \boldsymbol{a}) \right.$$
$$\left. + \frac{4 d_\gamma(\lambda)}{(1 - \gamma)\eta} + \left( \sqrt{d} + \frac{1}{1 - \gamma} \right) \min\{d_\gamma(\lambda), T\} + \sqrt{d} \lambda T \right) \tag{139}$$

*for any $\eta > 0$ and $\lambda > 0$, where $d_\gamma(\lambda)$ is defined as*

$$d_\gamma(\lambda) := 2d \log \left( 1 + \frac{T}{d\lambda} \right).$$

The proof of Lemma C.5 is provided in Appendix C.3.2.

Now we are ready to bound (ii), (iii) and (iv). To bound (ii), letting $\widehat{f}_t = f_t$ and $\widehat{\boldsymbol{\pi}}_t = \tilde{\boldsymbol{\pi}}_{t,n}$ for each $n$ in Lemma C.5 (recall we define $\tilde{\boldsymbol{\pi}}_{t,n} := (\tilde{\pi}_t^n, \boldsymbol{\pi}_t^{-n})$ in (73)), we have for any $\eta > 0$:

$$\text{(ii)} \leq \frac{\gamma}{1 - \gamma} \cdot \frac{\eta}{2N} \sum_{n=1}^{N} \sum_{t=1}^{T} \sum_{i=1}^{t-1} \mathbb{E}_{(s, \boldsymbol{a}) \sim d^{\tilde{\boldsymbol{\pi}}_{i,n}}(\rho)} \ell(f_t, s, \boldsymbol{a})$$
$$+ \frac{\gamma}{1 - \gamma} \left( \frac{4 d_\gamma(\lambda)}{(1 - \gamma)\eta} + \left( \sqrt{d} + \frac{1}{1 - \gamma} \right) \min\{d_\gamma(\lambda), T\} + \sqrt{d} \lambda T \right). \tag{140}$$

Letting $\widehat{f}_t = f_{t-1}$ and $\widehat{\pi}_t = \widetilde{\pi}_{t,n}$ for each $n$ in Lemma C.5, we can bound (iii) as follows:

$$
\text{(iii)} \le \frac{\gamma}{1-\gamma} \cdot \frac{\eta}{2N} \sum_{n=1}^{N} \sum_{t=1}^{T} \sum_{i=1}^{t-1} \mathbb{E}_{(s,\boldsymbol{a})\sim d^{\widetilde{\pi}_{i,n}}(\rho)} \ell(f_{t-1}, s, \boldsymbol{a})
$$
$$
+ \frac{\gamma}{1-\gamma} \left( \frac{4 d_\gamma(\lambda)}{(1-\gamma)\eta} + \left( \sqrt{d} + \frac{1}{1-\gamma} \right) \min\{d_\gamma(\lambda), T\} + \sqrt{d}\lambda T \right). \tag{141}
$$

To continue to bound the first term, note that

$$
\sum_{t=1}^{T} \sum_{i=1}^{t-1} \mathbb{E}_{(s,\boldsymbol{a})\sim d^{\widetilde{\pi}_{i,n}}(\rho)} \ell(f_{t-1}, s, \boldsymbol{a}) \le \sum_{t=1}^{T} \sum_{i=1}^{t-2} \mathbb{E}_{(s,\boldsymbol{a})\sim d^{\widetilde{\pi}_{i,n}}(\rho)} \ell(f_{t-1}, s, \boldsymbol{a}) + T
$$
$$
= \sum_{t=0}^{T-1} \sum_{i=1}^{t-1} \mathbb{E}_{(s,\boldsymbol{a})\sim d^{\widetilde{\pi}_{i,n}}(\rho)} \ell(f_{t-1}, s, \boldsymbol{a}) + T
$$
$$
\le \sum_{t=1}^{T} \sum_{i=1}^{t-1} \mathbb{E}_{(s,\boldsymbol{a})\sim d^{\widetilde{\pi}_{i,n}}(\rho)} \ell(f_{t-1}, s, \boldsymbol{a}) + T, \tag{142}
$$

where the first inequality uses the fact that

$$
\ell(f, s, \boldsymbol{a}) = D_{\mathsf{H}}^2 \left( \mathbb{P}_f(\cdot|s, \boldsymbol{a}) \| \mathbb{P}(\cdot|s, \boldsymbol{a}) \right) \le 1, \tag{143}
$$

the second line shifts the index of $t$ by 1, and the last line follows by noticing the first summand is 0 at $t = 0$.

Plugging the above relation back to (141) leads to

$$
\text{(iii)} \le \frac{\gamma}{1-\gamma} \cdot \frac{\eta}{2N} \sum_{n=1}^{N} \sum_{t=1}^{T} \sum_{i=1}^{t-1} \mathbb{E}_{(s,\boldsymbol{a})\sim d^{\widetilde{\pi}_{i,n}}(\rho)} \ell(f_t, s, \boldsymbol{a})
$$
$$
+ \frac{\gamma}{1-\gamma} \left( \frac{4 d_\gamma(\lambda)}{(1-\gamma)\eta} + \left( \sqrt{d} + \frac{1}{1-\gamma} \right) \min\{d_\gamma(\lambda), T\} + \sqrt{d}\lambda T + \frac{\eta}{2}T \right). \tag{144}
$$

Finally, similar to (144), letting $\widehat{f}_t = f_{t-1}$, $\widehat{\pi}_t = \pi_t$ for each $n$ and $\eta \leftarrow 2\eta$ in Lemma C.5, we can bound (iv) as follows:

$$
\text{(iv)} \le \frac{\gamma}{1-\gamma} \cdot \frac{\eta}{N} \sum_{n=1}^{N} \sum_{t=1}^{T} \sum_{i=1}^{t-1} \mathbb{E}_{(s,\boldsymbol{a})\sim d^{\pi_i}(\rho)} \ell(f_t, s, \boldsymbol{a})
$$
$$
+ \frac{\gamma}{1-\gamma} \left( \frac{2 d_\gamma(\lambda)}{(1-\gamma)\eta} + \left( \sqrt{d} + \frac{1}{1-\gamma} \right) \min\{d_\gamma(\lambda), T\} + \sqrt{d}\lambda T + \eta T \right). \tag{145}
$$

**Step 3: combining the bounds.** Letting $\eta = \frac{2(1-\gamma)}{\gamma\alpha}$ in (140), (144) and (145), we have with probability at least $1 - \delta$:

$$
\mathsf{Regret}(T) \le \frac{4T}{\alpha} \left( \sqrt{2} + \log\left( \frac{(N+1)T}{\delta} \right) + d\log\left( 1 + \sqrt{d}|\mathcal{S}|^2 T^2 \right) \right)
$$
$$
+ \frac{\gamma}{1-\gamma} \left( \frac{5\gamma\alpha d_\gamma(\lambda)}{(1-\gamma)^2} + 3\left( \sqrt{d} + \frac{1}{1-\gamma} \right) \min\{d_\gamma(\lambda), T\} + 3\sqrt{d}\lambda T + \frac{3(1-\gamma)}{\gamma\alpha}T \right).
$$

By setting

$$
\lambda = \sqrt{\frac{d}{T}}, \quad \alpha = \frac{(1-\gamma)^{3/2}}{\gamma} \sqrt{\frac{\log\left( \frac{(N+1)T}{\delta} \right) + d\log\left( 1 + \sqrt{d}|\mathcal{S}|^2 T^2 \right)}{d\log\left( 1 + \frac{T^{3/2}}{(1-\gamma)^2\sqrt{d}} \right)} T} \tag{146}
$$

in the above expression, we have with probability at least $1 - \delta$:

$$
\begin{aligned}
\mathsf{Regret}(T) \leq &\frac{4(1+\sqrt{2})\gamma}{(1-\gamma)^{3/2}} \sqrt{\frac{d\log\left(1 + \frac{T^{3/2}}{(1-\gamma)^2\sqrt{d}}\right)}{\log\left(\frac{(N+1)T}{\delta}\right) + d\log\left(1 + \sqrt{d}|\mathcal{S}|^2 T^2\right)}} \cdot \sqrt{T} \\
&+ \frac{\gamma d\sqrt{T}}{(1-\gamma)^{3/2}} \cdot 14\sqrt{\left(\frac{1}{d}\log\left(\frac{(N+1)T}{\delta}\right) + \log\left(1 + \sqrt{d}|\mathcal{S}|^2 T^2\right)\right)\log\left(1 + \frac{T^{3/2}}{(1-\gamma)^2\sqrt{d}}\right)} \\
&+ \frac{6\gamma}{1-\gamma}\left(\sqrt{d} + \frac{1}{1-\gamma}\right)d\log\left(1 + \frac{T^{3/2}}{(1-\gamma)^2\sqrt{d}}\right) + \frac{3\gamma}{1-\gamma}d\sqrt{T},
\end{aligned}
\tag{147}
$$

which gives the desired result.

### C.3.1. PROOF OF LEMMA C.4

Same as in (58), for the parameter space $\Theta$, by Assumption C.1 and Lemma B.5 we have

$$
\log \mathcal{N}(\Theta, \epsilon, \|\cdot\|_2) \leq d\log\left(1 + \frac{2\sqrt{d}}{\epsilon}\right)
\tag{148}
$$

for any $\epsilon > 0$. Thus there exists an $\epsilon$-net $\Theta_\epsilon$ of $\Theta$ ($\Theta_\epsilon \subset \Theta$) such that $\log|\Theta_\epsilon| \leq d\log\left(1 + \frac{2\sqrt{d}}{\epsilon}\right)$. Define

$$
\mathcal{F}_\epsilon := \left\{ f \in \mathcal{F} : f(s, \boldsymbol{a}, s') = \phi(s, \boldsymbol{a}, s')^\top \theta, \theta \in \Theta_\epsilon \right\}.
$$

For any $f \in \mathcal{F}$, there exists $\theta \in \Theta$ such that $f(s, \boldsymbol{a}, s') = \phi(s, \boldsymbol{a}, s')^\top \theta$. And there exists $\theta_\epsilon \in \Theta_\epsilon$ such that $\|\theta - \theta_\epsilon\|_2 \leq \epsilon$. We let $f_\epsilon(s, \boldsymbol{a}, s') = \phi(s, \boldsymbol{a}, s')^\top \theta_\epsilon$. Then $f_\epsilon \in \mathcal{F}_\epsilon$, and we have

$$
|\mathbb{P}_f(s'|s, \boldsymbol{a}) - \mathbb{P}_{f_\epsilon}(s'|s, \boldsymbol{a})| = |\phi(s, \boldsymbol{a}, s')^\top(\theta - \theta_\epsilon)| \leq \epsilon,
\tag{149}
$$

from which we deduce

$$
\forall t \in [T]: \quad -X_t^f \leq -\log\left(\frac{\mathbb{P}(s_t'|s_t, \boldsymbol{a}_t)}{\mathbb{P}_{f_\epsilon}(s_t'|s_t, \boldsymbol{a}_t) + \epsilon}\right) := -X_t^{f_\epsilon}(\epsilon).
\tag{150}
$$

Let $\mathcal{F}_t := \sigma(\mathcal{D}_t)$ be the $\sigma$-algebra generated by the data $\mathcal{D}_t$. By Lemma B.4 we have with probability at least $1 - \frac{\delta}{N+1}$:

$$
\begin{aligned}
\forall t \in [T], f_\epsilon \in \mathcal{F}_\epsilon: \quad -\frac{1}{2}\sum_{i=1}^{t-1} X_i^{f_\epsilon}(\epsilon) \leq &\sum_{i=1}^{t-1} \log \mathbb{E}\left[\exp\left(-\frac{1}{2}X_i^{f_\epsilon}(\epsilon)\right)\bigg|\mathcal{F}_{i-1}\right] \\
&+ \log\left(\frac{(N+1)T}{\delta}\right) + d\log\left(1 + \frac{2\sqrt{d}}{\epsilon}\right).
\end{aligned}
\tag{151}
$$

Thus we have

$$
\begin{aligned}
\forall t \in [T], f \in \mathcal{F}: \quad -\frac{1}{2}\sum_{i=1}^{t-1} X_i^f \overset{(150)}{\leq} &-\frac{1}{2}\sum_{i=1}^{t-1} X_i^{f_\epsilon}(\epsilon) \\
\overset{(151)}{\leq} &\sum_{i=1}^{t} \log \mathbb{E}\left[\exp\left(-\frac{1}{2}X_i^{f_\epsilon}(\epsilon)\right)\bigg|\mathcal{F}_{i-1}\right] + \log\left(\frac{(N+1)T}{\delta}\right) + d\log\left(1 + \frac{2\sqrt{d}}{\epsilon}\right).
\end{aligned}
\tag{152}
$$

We can further bound the first term in (152) as follows:

$$
\sum_{i=1}^{t} \log \mathbb{E}\left[\exp\left(-\frac{1}{2}X_i^{f_\epsilon}(\epsilon)\right)\bigg|\mathcal{F}_{i-1}\right]
$$

$$
\begin{aligned}
&= \sum_{i=1}^{t-1} \log \mathbb{E}_{\substack{(s_i, \boldsymbol{a}_i) \sim d^{\boldsymbol{\pi}_i}(\rho), \\ s_i' \sim \mathbb{P}(\cdot | s_i, \boldsymbol{a}_i)}} \left[ \sqrt{\frac{\mathbb{P}_{f_\epsilon}(s_i' | s_i, \boldsymbol{a}_i) + \epsilon}{\mathbb{P}(s_i' | s_i, \boldsymbol{a}_i)}} \right] \\
&= \sum_{i=1}^{t-1} \log \mathbb{E} \left[ \sqrt{\frac{\mathbb{P}_{f_\epsilon}(s_i' | s_i, \boldsymbol{a}_i) + \epsilon}{\mathbb{P}(s_i' | s_i, \boldsymbol{a}_i)}} \middle| \mathcal{F}_{s-1} \right] \\
&= \sum_{i=1}^{t-1} \log \mathbb{E}_{(s_i, \boldsymbol{a}_i) \sim d^{\boldsymbol{\pi}_i}(\rho)} \left[ \int_{\mathcal{S}} \sqrt{(\mathbb{P}_{f_\epsilon}(s_i' | s_i, \boldsymbol{a}_i) + \epsilon) \, \mathbb{P}(s_i' | s_i, \boldsymbol{a}_i)} ds_i' \right] \\
&\overset{(95)}{\leq} \sum_{i=1}^{t-1} \log \mathbb{E}_{(s_i, \boldsymbol{a}_i) \sim d^{\boldsymbol{\pi}_i}(\rho)} \left[ \int_{\mathcal{S}} \sqrt{(\mathbb{P}_f(s_i' | s_i, \boldsymbol{a}_i) + 2\epsilon) \, \mathbb{P}(s_i' | s_i, \boldsymbol{a}_i)} ds_i' \right].
\end{aligned}
\tag{153}
$$

Moreover, we have

$$
\begin{aligned}
&\mathbb{E}_{(s_i, \boldsymbol{a}_i) \sim d^{\boldsymbol{\pi}_i}(\rho)} \left[ \int_{\mathcal{S}} \sqrt{(\mathbb{P}_f(s_i' | s_i, \boldsymbol{a}_i) + 2\epsilon) \, \mathbb{P}(s_i' | s_i, \boldsymbol{a}_i)} ds_i' \right] \\
&\leq \mathbb{E}_{(s_i, \boldsymbol{a}_i) \sim d^{\boldsymbol{\pi}_i}(\rho)} \left[ \int_{\mathcal{S}} \sqrt{\mathbb{P}_f(s_i' | s_i, \boldsymbol{a}_i) \mathbb{P}(s_i' | s_i, \boldsymbol{a}_i)} ds_i' \right] + \mathbb{E}_{(s_i, \boldsymbol{a}_i) \sim d^{\boldsymbol{\pi}_i}(\rho)} \left[ \int_{\mathcal{S}} \sqrt{2\epsilon \mathbb{P}(s_i' | s_i, \boldsymbol{a}_i)} ds_i' \right] \\
&\leq 1 - \frac{1}{2} \mathbb{E}_{(s_i, \boldsymbol{a}_i) \sim d^{\boldsymbol{\pi}_i}(\rho)} \left[ \int_{\mathcal{S}} \left( \sqrt{\mathbb{P}_f(s_i' | s_i, \boldsymbol{a}_i)} - \sqrt{\mathbb{P}(s_i' | s_i, \boldsymbol{a}_i)} \right)^2 ds_i' \right] + \sqrt{2\epsilon} |\mathcal{S}| \\
&= 1 - \mathbb{E}_{(s_i, \boldsymbol{a}_i) \sim d^{\boldsymbol{\pi}_i}(\rho)} \left[ D_{\mathsf{H}}^2 \left( \mathbb{P}_f(\cdot | s_i, \boldsymbol{a}_i) \| \mathbb{P}(\cdot | s_i, \boldsymbol{a}_i) \right) \right] + \sqrt{2\epsilon} |\mathcal{S}|,
\end{aligned}
\tag{154}
$$

where the first inequality we use the fact that $\sqrt{a + b} \leq \sqrt{a} + \sqrt{b}$ for any $a, b \geq 0$.

Then combining (152), (154) and (135), we have

$$
\forall t \in [T], f \in \mathcal{F}: \quad -\frac{1}{2} \sum_{i=1}^{t-1} X_i^f \leq -\sum_{i=1}^{t-1} \mathbb{E}_{(s_i, \boldsymbol{a}_i) \sim d^{\boldsymbol{\pi}_i}(\rho)} \left[ \ell(f, s_i, \boldsymbol{a}_i) \right] \\
+ T\sqrt{2\epsilon} |\mathcal{S}| + \log \left( \frac{(N+1)T}{\delta} \right) + d \log \left( 1 + \frac{2\sqrt{d}}{\epsilon} \right),
$$

where we use the fact that $\log(x) \leq x - 1$ for any $x > 0$. Multiplying both sides by 2, we have with probability at least $1 - \frac{\delta}{N+1}$:

$$
\forall t \in [T], f \in \mathcal{F}: \quad -\sum_{i=1}^{t-1} X_i^f \leq -2 \sum_{i=1}^{t-1} \mathbb{E}_{(s_i, \boldsymbol{a}_i) \sim d^{\boldsymbol{\pi}_i}(\rho)} \left[ \ell(f, s_i, \boldsymbol{a}_i) \right] \\
+ 2T\sqrt{2\epsilon} |\mathcal{S}| + 2 \log \left( \frac{(N+1)T}{\delta} \right) + 2d \log \left( 1 + \frac{2\sqrt{d}}{\epsilon} \right).
\tag{155}
$$

Analogously, we can buond $-\sum_{i=1}^{t-1} Y_{i,n}^f$ for all $n \in [N]$ with probability at least $1 - \frac{N\delta}{N+1}$ as follows:

$$
\forall t \in [T], f \in \mathcal{F}, n \in [N]: \quad -\sum_{i=1}^{t-1} Y_{i,n}^f \leq -2 \sum_{i=1}^{t-1} \mathbb{E}_{(s_i^n, \boldsymbol{a}_i^n) \sim d^{\tilde{\boldsymbol{\pi}}_{i,n}}(\rho)} \left[ \ell(f, s_i^n, \boldsymbol{a}_i^n) \right] \\
+ 2T\sqrt{2\epsilon} |\mathcal{S}| + 2 \log \left( \frac{(N+1)T}{\delta} \right) + 2d \log \left( 1 + \frac{2\sqrt{d}}{\epsilon} \right).
\tag{156}
$$

By letting $\epsilon = \frac{1}{T^2 |\mathcal{S}|^2}$ in the above two inequalities, we have the desired result.

### C.3.2. PROOF OF LEMMA C.5

Similar as in the proof of Lemma B.9 in Appendix B.3.2, we first reformulate the value difference sequence $\sum_{t=1}^{T} \left| V_{\hat{f}_t, n}^{\hat{\boldsymbol{\pi}}_t}(\rho) - V_n^{\hat{\boldsymbol{\pi}}_t}(\rho) \right|$.

**Step 1: reformulation of the value difference sequence.** For any $f \in \mathcal{F}$ and $\boldsymbol{\pi} = (\pi^1, \cdots, \pi^N) : \mathcal{S} \to \Delta(\mathcal{A})$, we have

$$\forall n \in [N]: \quad V_{f,n}^{\boldsymbol{\pi}}(\rho) = \mathbb{E}_{\substack{s_0 \sim \rho, \boldsymbol{a}_h \sim \boldsymbol{\pi}(\cdot|s_h), \\ s_{h+1} \sim \mathbb{P}(\cdot|s_h, \boldsymbol{a}_h)}} \left[ \sum_{h=0}^{\infty} \gamma^h V_{f,n}^{\boldsymbol{\pi}}(s_h) - \gamma^{h+1} V_{f,n}^{\boldsymbol{\pi}}(s_{h+1}) \right]$$

$$= \mathbb{E}_{\substack{s_0 \sim \rho, \boldsymbol{a}_h \sim \boldsymbol{\pi}(\cdot|s_h), \\ s_{h+1} \sim \mathbb{P}(\cdot|s_h, \boldsymbol{a}_h)}} \left[ \sum_{h=0}^{\infty} \gamma^h \left( Q_{f,n}^{\boldsymbol{\pi}}(s_h, \boldsymbol{a}_h) - \beta \log \frac{\pi^n(a_h^n|s_h^n)}{\pi_{\mathsf{ref}}^n(a_h^n|s_h^n)} - \gamma V_{f,n}^{\boldsymbol{\pi}}(s_{h+1}) \right) \right], \quad (157)$$

where in the second line we use the fact that

$$V_{f,n}^{\boldsymbol{\pi}}(s) = \mathbb{E}_{\boldsymbol{a} \sim \boldsymbol{\pi}(\cdot|s)} \left[ Q_{f,n}^{\boldsymbol{\pi}}(s, \boldsymbol{a}) - \beta \log \frac{\pi^n(a^n|s^n)}{\pi_{\mathsf{ref}}^n(a^n|s^n)} \right].$$

And by the definition of $V_n^{\boldsymbol{\pi}}$ we have

$$\forall n \in [N]: \quad V_n^{\boldsymbol{\pi}}(\rho) = \mathbb{E}_{\substack{s_0 \sim \rho, \boldsymbol{a}_h \sim \boldsymbol{\pi}(\cdot|s_h), \\ s_{h+1} \sim \mathbb{P}(\cdot|s_h, \boldsymbol{a}_h)}} \left[ \sum_{h=0}^{\infty} \gamma^h \left( r^n(s_h, \boldsymbol{a}_h) - \beta \log \frac{\pi^n(a_h^n|s_h^n)}{\pi_{\mathsf{ref}}^n(a_h^n|s_h^n)} \right) \right]. \quad (158)$$

To simplify the notation, we define

$$\forall g \in \mathcal{F}: \quad \mathbb{P}_g V_{f,n}^{\boldsymbol{\pi}}(s, \boldsymbol{a}) := \mathbb{E}_{s' \sim \mathbb{P}_g(\cdot|s, \boldsymbol{a})} \left[ V_{f,n}^{\boldsymbol{\pi}}(s') \right]. \quad (159)$$

Combining (157) and (104), we have

$$V_{f,n}^{\boldsymbol{\pi}}(\rho) - V_n^{\boldsymbol{\pi}}(\rho) = \mathbb{E}_{\substack{s_0 \sim \rho, \boldsymbol{a}_h \sim \boldsymbol{\pi}(\cdot|s_h), \\ s_{h+1} \sim \mathbb{P}(\cdot|s_h, \boldsymbol{a}_h)}} \left[ \sum_{h=0}^{\infty} \gamma^h \left( Q_{f,n}^{\boldsymbol{\pi}}(s_h, \boldsymbol{a}_h) - r^n(s_h, \boldsymbol{a}_h) - \gamma V_{f,n}^{\boldsymbol{\pi}}(s_{h+1}) \right) \right]$$

$$= \frac{1}{1-\gamma} \mathbb{E}_{(s, \boldsymbol{a}) \sim d^{\boldsymbol{\pi}}(\rho)} \left[ Q_{f,n}^{\boldsymbol{\pi}}(s, \boldsymbol{a}) - r^n(s, \boldsymbol{a}) - \gamma \mathbb{P} V_{f,n}^{\boldsymbol{\pi}}(s, \boldsymbol{a}) \right]$$

$$= \frac{\gamma}{1-\gamma} \mathbb{E}_{(s, \boldsymbol{a}) \sim d^{\boldsymbol{\pi}}(\rho)} \big[ \underbrace{\mathbb{P}_f V_{f,n}^{\boldsymbol{\pi}}(s, \boldsymbol{a}) - \mathbb{P} V_{f,n}^{\boldsymbol{\pi}}(s, \boldsymbol{a})}_{:= \mathcal{E}_n^{\boldsymbol{\pi}}(f, s, \boldsymbol{a})} \big], \quad (160)$$

where the last relation follows from (126), and we define

$$\mathcal{E}_n^{\boldsymbol{\pi}}(f, s, \boldsymbol{a}) := \mathbb{P}_f V_{f,n}^{\boldsymbol{\pi}}(s, \boldsymbol{a}) - \mathbb{P} V_{f,n}^{\boldsymbol{\pi}}(s, \boldsymbol{a}). \quad (161)$$

Thus we have

$$\sum_{t=1}^{T} \left| V_{\widehat{f}_t, n}^{\widehat{\boldsymbol{\pi}}_t}(\rho) - V_n^{\widehat{\boldsymbol{\pi}}_t}(\rho) \right| = \frac{\gamma}{1-\gamma} \sum_{t=1}^{T} \left| \mathbb{E}_{(s, \boldsymbol{a}) \sim d^{\widehat{\boldsymbol{\pi}}_t}(\rho)} \left[ \mathcal{E}_n^{\widehat{\boldsymbol{\pi}}_t}(\widehat{f}_t, s, \boldsymbol{a}) \right] \right|. \quad (162)$$

Therefore, to bound $\sum_{t=1}^{T} \left| V_{\widehat{f}_t, n}^{\widehat{\boldsymbol{\pi}}_t}(\rho) - V_n^{\widehat{\boldsymbol{\pi}}_t}(\rho) \right|$, it suffices to bound the sum of model estimation errors $\sum_{t=1}^{T} \mathbb{E}_{(s, \boldsymbol{a}) \sim d^{\widehat{\boldsymbol{\pi}}_t}(\rho)} \left[ \mathcal{E}_n^{\widehat{\boldsymbol{\pi}}_t}(\widehat{f}_t, s, \boldsymbol{a}) \right]$.

**Step 2: bounding the sum of model estimation errors.** By Assumption 3.1, there exist $\theta_f$ and $\theta^\star$ in $\Theta$ such that $f(s'|s, \boldsymbol{a}) = \phi(s, \boldsymbol{a}, s')^\top \theta_f$ and $\mathbb{P}(s'|s, \boldsymbol{a}) = \phi(s, \boldsymbol{a}, s')^\top \theta^\star$. Thus we have

$$\mathbb{E}_{(s, \boldsymbol{a}) \sim d^{\boldsymbol{\pi}}(\rho)} \left[ \mathcal{E}_n^{\boldsymbol{\pi}}(f, s, \boldsymbol{a}) \right] = (\theta_f - \theta^\star)^\top \underbrace{\mathbb{E}_{(s, \boldsymbol{a}) \sim d^{\boldsymbol{\pi}}(\rho)} \left[ \int_{\mathcal{S}} \phi(s, \boldsymbol{a}, s') V_{f,n}^{\boldsymbol{\pi}}(s') ds' \right]}_{:= x_n(f, \boldsymbol{\pi})}. \quad (163)$$

We let $x_n^i(f, \boldsymbol{\pi})$ denote the $i$-th component of $x_n(f, \boldsymbol{\pi})$, i.e.,

$$x_n^i(f, \boldsymbol{\pi}) = \mathbb{E}_{(s, \boldsymbol{a}) \sim d^{\boldsymbol{\pi}}(\rho)} \left[ \int_{\mathcal{S}} \phi^i(s, \boldsymbol{a}, s') V_{f,n}^{\boldsymbol{\pi}}(s') ds' \right].$$

Then we have

$$\forall i \in [d]: \quad |x_n^i(f, \boldsymbol{\pi})| \le \frac{1}{1 - \gamma} \tag{164}$$

(recall that by the definition of linear mixture model (c.f. Assumption C.1), $\phi^i(s, \boldsymbol{a}, \cdot) \in \Delta(\mathcal{S})$ for each $i \in [d]$), which gives

$$\|x_n(f, \boldsymbol{\pi})\|_2 \le \frac{1}{1 - \gamma} \sqrt{d}. \tag{165}$$

For each $t \in [T]$, we define $\Lambda_t \in \mathbb{R}^{d \times d}$ as

$$\Lambda_t := \lambda I_d + \sum_{i=1}^{t-1} x_n(\widehat{f}_i, \widehat{\boldsymbol{\pi}}_i) x_n(\widehat{f}_i, \widehat{\boldsymbol{\pi}}_i)^\top. \tag{166}$$

We write $\widehat{\theta}_t$ as the parameter of $\widehat{f}_t$. Then we have the following decomposition:

$$\sum_{t=1}^T \left| \mathbb{E}_{(s,\boldsymbol{a}) \sim d^{\widehat{\boldsymbol{\pi}}_t}(\rho)} \left[ \mathcal{E}_n^{\widehat{\boldsymbol{\pi}}_t}(\widehat{f}_t, s, \boldsymbol{a}) \right] \right| = \underbrace{\sum_{t=1}^T \left| \langle x_n(\widehat{f}_t, \widehat{\boldsymbol{\pi}}_t), \widehat{\theta}_t - \theta^\star \rangle \right| \mathbb{1} \left\{ \left\| x_n(\widehat{f}_t, \widehat{\boldsymbol{\pi}}_t) \right\|_{\Lambda_t^{-1}} \le 1 \right\}}_{(a)}$$

$$+ \underbrace{\sum_{t=1}^T \left| \langle x_n(\widehat{f}_t, \widehat{\boldsymbol{\pi}}_t), \widehat{\theta}_t - \theta^\star \rangle \right| \mathbb{1} \left\{ \left\| x_n(\widehat{f}_t, \widehat{\boldsymbol{\pi}}_t) \right\|_{\Lambda_t^{-1}} > 1 \right\}}_{(b)}. \tag{167}$$

Below we bound (a) and (b) separately.

**Bounding (a).**

By the Cauchy-Schwarz inequality, we have

$$(a) \le \sum_{t=1}^T \left\| \widehat{\theta}_t - \theta^\star \right\|_{\Lambda_t} \left\| x_n(\widehat{f}_t, \widehat{\boldsymbol{\pi}}_t) \right\|_{\Lambda_t^{-1}} \mathbb{1} \left\{ \left\| x_n(\widehat{f}_t, \widehat{\boldsymbol{\pi}}_t) \right\|_{\Lambda_t^{-1}} \le 1 \right\}$$

$$\le \sum_{t=1}^T \left\| \widehat{\theta}_t - \theta^\star \right\|_{\Lambda_t} \min \left\{ \left\| x_n(\widehat{f}_t, \widehat{\boldsymbol{\pi}}_t) \right\|_{\Lambda_t^{-1}}, 1 \right\}, \tag{168}$$

where the last inequality follows from the fact that

$$\left\| x_n(\widehat{f}_t, \widehat{\boldsymbol{\pi}}_t) \right\|_{\Lambda_t^{-1}} \mathbb{1} \left\{ \left\| x_n(\widehat{f}_t, \widehat{\boldsymbol{\pi}}_t) \right\|_{\Lambda_t^{-1}} \le 1 \right\} \le \min \left\{ \left\| x_n(\widehat{f}_t, \widehat{\boldsymbol{\pi}}_t) \right\|_{\Lambda_t^{-1}}, 1 \right\}.$$

By Lemma B.2, Lemma B.3 and (165), we have

$$\sum_{i=1}^t \min \left\{ \left\| x_n(\widehat{f}_i, \widehat{\boldsymbol{\pi}}_i) \right\|_{\Lambda_i^{-1}}, 1 \right\} \le 2d \log \left( 1 + \frac{T}{(1 - \gamma)^2 \lambda} \right) := d_\gamma(\lambda). \tag{169}$$

holds for any $\lambda > 0$ and $t \in [T]$.

Further, by the definition of $\Lambda_t$ (c.f. (166)) and Assumption C.1 we have

$$\left\| \widehat{\theta}_t - \theta^\star \right\|_{\Lambda_t} \le 2\sqrt{\lambda d} + \left( \sum_{i=1}^{t-1} |\langle \widehat{\theta}_t - \theta^\star, x_n(\widehat{f}_i, \widehat{\boldsymbol{\pi}}_i) \rangle|^2 \right)^{1/2}, \tag{170}$$

which gives

$$
\begin{aligned}
\sum_{t=1}^{T} & \left\| \widehat{\theta}_t - \theta^\star \right\|_{\Lambda_t} \min \left\{ \left\| x_n(\widehat{f}_t, \widehat{\boldsymbol{\pi}}_t) \right\|_{\Lambda_t^{-1}}, 1 \right\} \\
& \leq \sum_{t=1}^{T} \left( 2\sqrt{\lambda d} + \left( \sum_{i=1}^{t-1} |\langle \widehat{\theta}_t - \theta^\star, x_n(\widehat{f}_i, \widehat{\boldsymbol{\pi}}_i) \rangle|^2 \right)^{1/2} \right) \cdot \min \left\{ \left\| x_n(\widehat{f}_t, \widehat{\boldsymbol{\pi}}_t) \right\|_{\Lambda_t^{-1}}, 1 \right\} \\
& \leq \left( \sum_{t=1}^{T} 4\lambda d \right)^{1/2} \left( \sum_{t=1}^{T} \min \left\{ \left\| x_n(\widehat{f}_t, \widehat{\boldsymbol{\pi}}_t) \right\|_{\Lambda_t^{-1}}, 1 \right\} \right)^{1/2} \\
& \quad + \left( \sum_{t=1}^{T} \sum_{i=1}^{t-1} |\langle \widehat{\theta}_t - \theta^\star, x_n(\widehat{f}_i, \widehat{\boldsymbol{\pi}}_i) \rangle|^2 \right)^{1/2} \left( \sum_{t=1}^{T} \min \left\{ \left\| x_n(\widehat{f}_t, \widehat{\boldsymbol{\pi}}_t) \right\|_{\Lambda_t^{-1}}, 1 \right\} \right)^{1/2} \\
& \leq 2\sqrt{\lambda d T \min\{d_\gamma(\lambda), T\}} + \left( d_\gamma(\lambda) \sum_{t=1}^{T} \sum_{i=1}^{t-1} |\langle \widehat{\theta}_t - \theta^\star, x_n(\widehat{f}_i, \widehat{\boldsymbol{\pi}}_i) \rangle|^2 \right)^{1/2},
\end{aligned}
\tag{171}
$$

where the first inequality uses (170) and the second inequality uses the Cauchy-Schwarz inequality and the fact that

$$
\min \left\{ \left\| x_n(\widehat{f}_t, \widehat{\boldsymbol{\pi}}_t) \right\|_{\Lambda_t^{-1}}, 1 \right\}^2 \leq \min \left\{ \left\| x_n(\widehat{f}_t, \widehat{\boldsymbol{\pi}}_t) \right\|_{\Lambda_t^{-1}}, 1 \right\},
$$

and the last inequality uses (169).

Furthermore, we have

$$
\begin{aligned}
|\langle \widehat{\theta}_t - \theta^\star, x_n(\widehat{f}_i, \widehat{\boldsymbol{\pi}}_i) \rangle|^2 &= \left| \mathbb{E}_{(s,\boldsymbol{a}) \sim d^{\widehat{\boldsymbol{\pi}}_i}(\rho)} \left[ \int_{\mathcal{S}} \left( \mathbb{P}_{\widehat{f}_t}(s'|s,\boldsymbol{a}) - \mathbb{P}(s'|s,\boldsymbol{a}) \right) V_{\widehat{f}_i,n}^{\widehat{\boldsymbol{\pi}}_i}(s') ds' \right] \right|^2 \\
&\leq \mathbb{E}_{(s,\boldsymbol{a}) \sim d^{\widehat{\boldsymbol{\pi}}_i}(\rho)} \left[ \left( \int_{\mathcal{S}} \left( \mathbb{P}_{\widehat{f}_t}(s'|s,\boldsymbol{a}) - \mathbb{P}(s'|s,\boldsymbol{a}) \right) V_{\widehat{f}_i,n}^{\widehat{\boldsymbol{\pi}}_i}(s') ds' \right)^2 \right] \\
&\leq 4 \left\| V_{\widehat{f}_i,n}^{\widehat{\boldsymbol{\pi}}_i}(\cdot) \right\|_\infty \mathbb{E}_{(s,\boldsymbol{a}) \sim d^{\widehat{\boldsymbol{\pi}}_t}(\rho)} D_{\text{TV}}^2 \left( \mathbb{P}_{\widehat{f}_t}(\cdot|s,\boldsymbol{a}) \| \mathbb{P}(\cdot|s,\boldsymbol{a}) \right) \\
&\leq \frac{8}{1-\gamma} \mathbb{E}_{(s,\boldsymbol{a}) \sim d^{\widehat{\boldsymbol{\pi}}_i}(\rho)} D_{\text{H}}^2 \left( \mathbb{P}_{\widehat{f}_t}(\cdot|s,\boldsymbol{a}) \| \mathbb{P}(\cdot|s,\boldsymbol{a}) \right) \\
&= \frac{8}{1-\gamma} \mathbb{E}_{(s,\boldsymbol{a}) \sim d^{\widehat{\boldsymbol{\pi}}_i}(\rho)} \ell(\widehat{f}_t, s, \boldsymbol{a}),
\end{aligned}
\tag{172}
$$

where the second line uses the Cauchy-Schwarz inequality, the third line follows from Höder's inequality, the fourth line uses the inequality $D_{\text{TV}}^2(P\|Q) \leq 2D_{\text{H}}^2(P\|Q)$ and the fact that $\left\| V_{\widehat{f}_t,n}^{\widehat{\boldsymbol{\pi}}_t}(\cdot) \right\|_\infty \leq \frac{1}{1-\gamma}$. The last line uses (135).

Plugging (172) into (171), we have

$$
(a) \leq 2\sqrt{d} \cdot \sqrt{\lambda T \min\{d_\gamma(\lambda), T\}} + \left( \frac{8 d_\gamma(\lambda)}{1-\gamma} \sum_{t=1}^{T} \sum_{i=1}^{t-1} \mathbb{E}_{(s,\boldsymbol{a}) \sim d^{\widehat{\boldsymbol{\pi}}_i}(\rho)} \ell(\widehat{f}_t, s, \boldsymbol{a}) \right)^{1/2}.
\tag{173}
$$

**Bounding (b).**

$$
\begin{aligned}
(b) &= \sum_{t=1}^{T} \left| \langle x_n(\widehat{f}_t, \widehat{\boldsymbol{\pi}}_t), \widehat{\theta}_t - \theta^\star \rangle \right| \mathbb{1} \left\{ \left\| x_n(\widehat{f}_t, \widehat{\boldsymbol{\pi}}_t) \right\|_{\Lambda_t^{-1}} > 1 \right\} \\
&\leq \sum_{t=1}^{T} \left| \langle x_n(\widehat{f}_t, \widehat{\boldsymbol{\pi}}_t), \widehat{\theta}_t - \theta^\star \rangle \right| \min \left\{ \left\| x_n(\widehat{f}_t, \widehat{\boldsymbol{\pi}}_t) \right\|_{\Lambda_t^{-1}}, 1 \right\},
\end{aligned}
\tag{174}
$$

where the inequality follows from the fact that

$$\mathbb{1}\left\{\left\|x_n(\widehat{f}_t, \widehat{\boldsymbol{\pi}}_t)\right\|_{\Lambda_t^{-1}} > 1\right\} \leq \min\left\{\left\|x_n(\widehat{f}_t, \widehat{\boldsymbol{\pi}}_t)\right\|_{\Lambda_t^{-1}}, 1\right\}.$$

Note that

$$\left\|x_n(\widehat{f}_t, \widehat{\boldsymbol{\pi}}_t)\right\|_{\Lambda_t^{-1}} \mathbb{1}\left\{\left\|x_n(\widehat{f}_t, \widehat{\boldsymbol{\pi}}_t)\right\|_{\Lambda_t^{-1}} \leq 1\right\} \leq \min\left\{\left\|x(\widehat{\mu}_t, \widehat{\nu}_t)\right\|_{\Lambda_t^{-1}}, 1\right\}$$

Thus by (169) and (174), we have

$$(b) \leq \frac{1}{1-\gamma}\min\{T, d_\gamma(\lambda)\}. \tag{175}$$

Plugging (173), (175) into (167), we have

$$\sum_{t=1}^{T}\left|\mathbb{E}_{(s,\boldsymbol{a})\sim d^{\widehat{\boldsymbol{\pi}}_t}(\rho)}\left[\mathcal{E}_n^{\widehat{\boldsymbol{\pi}}_t}(\widehat{f}_t, s, \boldsymbol{a})\right]\right|$$

$$\leq 2\sqrt{d}\cdot\sqrt{\lambda T\min\{d_\gamma(\lambda), T\}} + \left(\frac{8d_\gamma(\lambda)}{1-\gamma}\sum_{t=1}^{T}\sum_{i=1}^{t-1}\mathbb{E}_{(s,\boldsymbol{a})\sim d^{\widehat{\boldsymbol{\pi}}_i}(\rho)}\ell(\widehat{f}_t, s, \boldsymbol{a})\right)^{1/2} + \frac{1}{1-\gamma}\min\{T, d_\gamma(\lambda)\}$$

$$\leq \left(\frac{8d_\gamma(\lambda)}{(1-\gamma)\eta}\cdot\eta\sum_{t=1}^{T}\sum_{i=1}^{t-1}\mathbb{E}_{(s,\boldsymbol{a})\sim d^{\widehat{\boldsymbol{\pi}}_i}(\rho)}\ell(\widehat{f}_t, s, \boldsymbol{a})\right)^{1/2} + \left(\sqrt{d} + \frac{1}{1-\gamma}\right)\min\{d_\gamma(\lambda), T\} + \sqrt{d}\lambda T$$

$$\leq \frac{4d_\gamma(\lambda)}{(1-\gamma)\eta} + \frac{\eta}{2}\sum_{t=1}^{T}\sum_{i=1}^{t-1}\mathbb{E}_{(s,\boldsymbol{a})\sim d^{\widehat{\boldsymbol{\pi}}_i}(\rho)}\ell(\widehat{f}_t, s, \boldsymbol{a}) + \left(\sqrt{d} + \frac{1}{1-\gamma}\right)\min\{d_\gamma(\lambda), T\} + \sqrt{d}\lambda T \tag{176}$$

for any $\eta > 0$, where the second and third inequalities both use the fact that $\sqrt{ab} \leq \frac{a+b}{2}$ for any $a, b \geq 0$.

Finally, combining (162) with the above inequality, we have (139).

