# OpenReview forum: "Incentivize without Bonus: Provably Efficient Model-based Online Multi-agent RL for Markov Games"
_ICML.cc/2025/Conference — ICML 2025 poster_

### Official Review · Reviewer_n2KT · 2025-03-02

**Overall Recommendation:** 2

**Summary:**

This paper develops a value-incentivized model-based method for computing epsilon-optimal NE and CCE in matrix games and Markov games. The key idea of value-incentivized model-based method is that, conditioned on any policy at certain iterate, it finds the most adversarial game model that both fits well with the collected data via MLE and encourages the game value to be small under the current policy. This encourages that in the next iterate, it finds a better policy to maximize the game value conditioned on the current game model. This mechanism of updating the game model and policy alternatively implicitly encourages exploration without designing any specific bonus function. The paper gives a detailed algorithmic framework for matrix games, general-sum multi-player Markov games with general function approximation. The paper then gives theoretical guarantees for linear function function approximation, nearly recovering minimax optimal bounds in these cases.

**Claims And Evidence:**

The claims made in the submission were supported by clear and convincing evidence

**Essential References Not Discussed:**

See "Relation To Broader Scientific Literature"

**Experimental Designs Or Analyses:**

NA

**Methods And Evaluation Criteria:**

The proposed methods make sense.

**Other Comments Or Suggestions:**

See "Relation To Broader Scientific Literature"

**Other Strengths And Weaknesses:**

See "Relation To Broader Scientific Literature"

**Questions For Authors:**

See "Relation To Broader Scientific Literature"

**Relation To Broader Scientific Literature:**

The paper gives a nice overview of the development of reward-biased MLE methods, from bandits to MDP to Markov games (which the present paper addresses).


Though not being called reward-biased MLE, its idea has been extensively recognized and developed into RL and Markov games that the present paper seems lack of proper discussion. We have known well that one can use some form of regularization to encourage exploration in online learning setting (and pessimism in offline learning setting), without the need to construct any explicit bonus function. A typical example for that is the series of results for the estimation-decision coefficient framework by Dylan Foster at al., where the their algorithmic framework is essentially the bi-level optimization between constructing an adversarial game model and extracting its optimal policy, using an regularized objective without any bonus function.

Given that, I think the present paper needs to discuss such line of works as well.

At the same time, I am concerned with the significance of the results developed in the present paper given the similar idea and algorithmic framework is already developed and is even more general than the present paper, e.g., https://arxiv.org/pdf/2305.00684.

**Theoretical Claims:**

The theoretical claims make sense.

---

> ### Author Rebuttal · Authors · 2025-03-30
>
> # Response to Reviewer n2KT
>
> Thank you for your valuable feedback. If the clarifications below address your primary concerns, we'd appreciate your consideration of increasing your score. Certainly, please don't hesitate to request any further clarification.
>
> > **relationship with reward-biased MLE and the framework by Foster et al.**
>
> Thank you for your comment, and we will be happy to further discuss these related works. We fully acknowledge that our work is deeply related to the existing works on reward-biased MLE, which were discussed in the related work section in the paragraph "Uncertainty estimation in online RL". We will be more than happy to further discuss the line of work by Foster et al on Decision-Estimation Coefficient (DEC), which pioneered a unified complexity notion for many RL problems with both lower and upper bounds. There are a few algorithms proposed in the sequence of papers by Foster et al., and while some algorithmic variants still leverage explicit uncertainty estimation, the variants leveraging the so-called "optimimistic estimation" [Zhang, 2022] bears close connection to the reward-biasing idea explored in our work. Our main contribution, however, is towards the design of practical algorithms for multi-agent RL with provable guarantees, and we will elaborate further how our algorithmic design is more desirable compared to those existing in the literature. While the development is relatively extensive and mature in the single-agent setting, how to develop efficient and practical algorithms with provable guarantees in the MARL context is still highly under-developed.
>
> > **significance of the results developed in the present paper given the similar idea and algorithmic framework is already developed and is even more general than the present paper, e.g., https://arxiv.org/pdf/2305.00684.**
>
> Thank you for bringing up this paper to our attention! We agree https://arxiv.org/pdf/2305.00684 ([Foster et al, 2023a]) provided a general framework for many multi-agent RL settings that is highly inspiring. However, please allow us to elaborate why our results are still significant.
>
> - In [Foster et al, 2023a], theoretical guarantees for Markov games with function approximation are not explicitly provided. And as noted in their own paper below Thm. 1.6, their bounds do not lead to tight convergence guarantees for non-convex problem classes such as Markov games. In contrast, our algorithms have a near-optimal regret/sample complexity bound for the linear function approximation setting.
>
> - Our proposed algorithm is much more practical compared with that in [Foster et al, 2023a]. The first set of algorithm in [Foster et al, 2023a] reuses the algorithm proposed in [Foster et al, 2023b], which requires explicit uncertainty quantification of the model estimation in Helliger distance, as well as requiring solving a challenging constrained minimax optimization problem. The Algorithm 1 in [Foster et al, 2023a] requires storing all historical value estimators $\hat{f_k^{i}}$ for all $k\in[K]$ and $i\in [t-1]$ in each iteration $t$ in order to compute $q_k^t$ in line 4, leading to prohibitive memory requirements that scale with the number of iterations. Besides, the minmax objective $\Gamma_{q,\eta}$ in line 3 of Algorithm 1 involves optimization over four components simultaneously (two policies, one value estimator, and one transition kernel estimator), creating a computationally intensive procedure. In comparison, our algorithm only keeps a transition kernel estimator and is more efficient in both computation and memory requirements. To corroborate the tractability of our design, we have provided further numerical experiments, please see our response to Reviewer 4hiw. In contrast, there is no implementation for the algorithms in [Foster et al, 2023a].
>
> - In addtion, our VMG could be extended to the infinite-horizon setting, see Algorithm 6 in our paper. In contrast, [Foster et al, 2023a] only considers the finite-horizon setting.
>
> ---
>
> Zhang, T. (2022). Feel-good thompson sampling for contextual bandits and reinforcement learning. SIAM Journal on Mathematics of Data Science.
>
> Foster, D. J., et al (2023b). Tight Guarantees for Interactive Decision Making with the Decision-Estimation Coefficient. arXiv preprint arXiv:2301.08215.

---

### Official Review · Reviewer_9p5x · 2025-03-11

**Overall Recommendation:** 4

**Summary:**

The authors propose a novel algorithm for solving online general-sum $n$-player Markov games in finite and linear mixture MDPs. The proposed algorithm incentivizes exploration without introducing bonuses or constrained optimization to achieve optimism. Instead, it carefully applies regularization to the main objective in order to incentivize the algorithm to be more optimistic and, as a result, exploratory, without any need for sophisticated uncertainty quantification.
The algorithm achieves $\widetilde{O}(d\sqrt{T})$ regret bound for zero-sum matrix games, $\widetilde{O}(d\sqrt{TH^3})$ regret bound for general-sum Markov games, and the corresponding sample complexities for the NE and CCE identification correspondingly. Additionally, the authors generalize their result to the discounted setting.

**Claims And Evidence:**

**Claim 1.** The authors propose the algorithm for solving two-player zero-sum matrix game with regret $O(d\sqrt{T})$ under linear functional approximation.

The proof of this claim looks correct to me.

**Claim 2.** For finite-horizon multi-player general-sum Markov games, under the linear mixture model of the transition kernel, the proposed algorithm achieves near-optimal $O(d\sqrt[H^3 T})$ regret bound.

The proof of this claim looks correct to me.

**Claim 3.** The unified framework allows to cover many cases (such as symmetric games) that might be of independent interest.

This claim has evidence in terms of reformulations presented in Appendix A.

**Essential References Not Discussed:**

All the literature was discussed correctly.

**Experimental Designs Or Analyses:**

N/A

**Methods And Evaluation Criteria:**

N/A

**Other Comments Or Suggestions:**

- Step 1 in the proof of Lemma B.9 is a well-known performance difference lemma (up to adaptation to a regularized and finite-horizon case), I think the proper attribution could make it easier for the reader who already knows the performance-difference lemma.

**Other Strengths And Weaknesses:**

As an additional strength, I enjoyed the paper's clarity and the final algorithm's elegance.

**Questions For Authors:**

1. Is it possible to draw the connection between the proposed algorithm for symmetric matrix games and online modification of XPO (Xie et al., 2024)?


Xie, T., Foster, D. J., Krishnamurthy, A., Rosset, C., Awadallah, A., & Rakhlin, A. (2024). Exploratory preference optimization: Harnessing implicit q*-approximation for sample-efficient rlhf. arXiv preprint arXiv:2405.21046.

**Relation To Broader Scientific Literature:**

I found the main contribution of this paper to be aligned with the current line of research on implementable and provably efficient exploration. In particular, the idea of introducing optimism via regularization has already been shown to be practical, as is usual for single-agent RL (Liu et al., 2024b) and LLM alignment (see Xie et al., 2024).



Xie, T., Foster, D. J., Krishnamurthy, A., Rosset, C., Awadallah, A., & Rakhlin, A. (2024). Exploratory preference optimization: Harnessing implicit q*-approximation for sample-efficient rlhf. arXiv preprint arXiv:2405.21046.

**Theoretical Claims:**

I have carefully verified the proof for the matrix game setting and read the proof for the general Markov games in the finite-horizon setting and they look good to me.

---

> ### Author Rebuttal · Authors · 2025-03-30
>
> # Response to Reviewer 9p5x
>
> Thank you very much for your positive feedback and for carefully reviewing our proofs! Below we answer your questions.
>
> > **Step 1 in the proof of Lemma B.9 is a well-known performance difference lemma (up to adaptation to a regularized and finite-horizon case), I think the proper attribution could make it easier for the reader who already knows the performance-difference lemma.**
>
> Thank you for reading into the proof details and raising this point. We will add the following sentence at the beginning of Step 1 in Lemma B.9 in the revised paper:
>
> *In this step, we adapt the performance difference lemma [1] to our regularized and finite-horizon setting.*
>
> >**Is it possible to draw connection between the proposed algorithm for symmetric matrix games and online modification of XPO?**
>
> This is an interesting question. For symmetric matrix games, our approach simplifies to a single-player algorithm — as illustrated by Algorithm 3 in Appendix A. Interestingly, a similar reduction is observed in recent works on win-rate games for LLM alignment [2]-[4] (the win-rate matrix (refer to Eq.(2) in [4]) is indeed skew-symmetric). In those studies, the goal is to obtain a policy that has a higher win rate against any other policies by having the model engage in self-play. However, these works did not address the exploration challenge, and the integration of online exploration - through regularization strategies as proposed in our work — could enhance performance in online settings and is a promising future direction.
>
> In addition to symmetric matrix games, we build connection between VPO [5], a concurrent work of XPO, in the Bandit setting (see Algorithm 4 in Appendix A of our paper). Interestingly, the reduction bears similarity to VPO/XPO, but does not lead to exactly the same algorithm.
>
> ---
>
> [1] Sham Kakade and John Langford. Approximately Optimal Approximate Reinforcement Learning. In Proceedings of the 19th International Conference on Machine Learning, volume 2, pages 267–274, 2002.
>
> [2] Swamy et al., 2024. A minimaximalist approach to reinforcement learning from human feedback.
>
> [3] Wu et al., 2024. Self-play preference optimization for language model alignment.
>
> [4] Yang et al., 2025. Faster WIND: Accelerating Iterative Best-of-N Distillation for LLM Alignment.
>
> [5] Cen et al., 2024. Value-incentivized preference optimization: A unified approach to online and offline RLHF.

---

> > ### Comment · Reviewer_9p5x · 2025-04-02
> >
> > I would like to thank the authors for their answer, and I am happy to keep my score.

---

### Official Review · Reviewer_NUsj · 2025-03-14

**Overall Recommendation:** 3

**Summary:**

This paper propose the strategy of value-incentivized exploration  for online  Markov games. Specifically, this method use a regularizer term to incentive the players to deviate from their current policy, resulting in exploration.  Theoretical analysis shows that the proposed algorithms achieve a near-optimal regret on the order of
 $\tilde{O}(d\sqrt{T})$ for  two-player zero-sum matrix games and $ \tilde{O}(d\sqrt{H^3T})$ for multi-player general-sum Markov
games. This work also  considers the infinite  horizon setting, which achieves a sample complexity of $ \tilde{O}(Nd^2/((1-\gamma)^4)\epsilon^2)$.

**Claims And Evidence:**

The claims are all supported by clear and convincing evidence.

**Essential References Not Discussed:**

There are no essential references not discussed.

**Experimental Designs Or Analyses:**

This paper has no experiments.

**Methods And Evaluation Criteria:**

The proposed methods  make sense for the problem.

**Other Comments Or Suggestions:**

The description of Thompson Sampling in the related work is not accurate enough.
The Thompson Sampling method is generally easier to implement than the UCB method, as it only requires updating the posterior distribution (some approximation algorithms can be used). For nonlinear MDPs, function approximation methods are generally used. Thompson sampling is often considered a computationally more tractable alternative to UCB algorithms [5].

I suggest adding some literature that uses posterior methods [6,7] to study MARL in the related works.

**Other Strengths And Weaknesses:**

**Strengths**:
1. The method proposed in this paper does not require explicitly designing a bonus term to encourage exploration, as constructing the uncertainty sets becomes intractable  when using function approximation.  Compared to the literature [2] that adopts the same idea, this paper is somewhat easier to implement.
2. Compared to the information-theoretic lower bound obtained in [3], the regret upper bound in this paper is nearly optimal.
3. This paper also extend VMG to the infinite-horizon setting.

**Weaknesses:**
1. The core idea of this article is similar to MEX [2], and the author claims that one of the advantages compared to MEX is computational efficiency. However, every iteration  in Algorithm 2 requires calculating a Nash equilibrium (Line 4) for a multi-player Markov game, which might hinder computational efficiency.
2. Section 2 studies two-player zero-sum Markov games and provides an upper bound on regret as $\tilde{O}(d\sqrt{T})$ in Theorem 2.4. $d$ is the dimension of the linear approximation payoff matrix. If $d=m \times n$, then the upper bound is $\tilde{O}(\sqrt{m^2n^2T})$. This is larger than the upper bound $\tilde{O}(\sqrt{mnT})$ given in [4].

**Questions For Authors:**

1. Algorithm 2 requires calculating a Nash equilibrium for a multi-player game in each iteration. In your proof (page 21, line 1106), it is necessary to assume that the Line 4 of Algorithm 2 yields an exact equilibrium or at least a very close approximation. Therefore, Algorithm 2 requires sufficient iterations in Line 4  to achieve a sufficiently precise solution. Will this step affect computational efficiency? How does it compare with the computational efficiency of MEX?
2. Section 2.3 performs linear parameterization on matrix games. Given a feature vector $ \phi(i,j),i \in [m],j \in [n]$, assumption 2.1 and 2.2 hold is equivalent to a $mn \times d$-dimensional linear equations having a solution.
(1) If $d < mn$ (which is generally the case), this system of linear equations may not have a solution, thus assumptions 2.1 and 2.2 limit its applicability range.
(2) If $ d \geq mn$,  the upper bound in theorem 2.4 will increase. Could the author explain the issues brought by the linear parameterization?  What are the benefits of performing linear approximation on matrix games?

**Relation To Broader Scientific Literature:**

This paper is focus on the theoretical side  and does not have a significant connection with the broader scientific literature.

**Theoretical Claims:**

I did not thoroughly check every detail of the proof. The theoretical analysis in the appendix appears to be correct. Some analytical methods adopt the theoretical techniques from references [1] and [2], such as Lemmas B.8 and B.9. This is reasonable since the method proposed in this paper shares some similarities with that in [2], as both utilize value-incentivized exploration.

---

> ### Author Rebuttal · Authors · 2025-03-30
>
> # Response to Reviewer NUsj
>
> Thank you for your insightful comments! Since the references in your review are not specified, we addressed some questions through best guesses -- we are happy to provide further answers if there are gaps in our interpretation. If these clarifications address your concerns, we'd appreciate your consideration of increasing your score.
>
> > **W1: NE computation**
>
> We want to emphasize that the computation of the NE/CCE is a standard subroutine in many existing approaches, including MEX. In addition, there are many existing computationally efficient algorithms for computing the NE of two-player zero-sum Markov games or the CCE of multi-player general-sum Markov games, such as [Cen et al., 2022; Zhang et al., 2022]. They can be efficiently leveraged to ensure the computational efficiency the the proposed algorithm.
>
> > **regret bound worse than [4] in the tabular setting**
>
> We infer [4] is *O’Donoghue et al., 2021. Matrix games with bandit feedback.* The gap between our bound and their bound stems from function approximation considered in our paper. For the tabular setting, entrywise bonuses are allowed to be computed for each state-action pair individually, which is prohibitive with fucntion approximation. As we remark below Theorem 2.4, even for the simpler linear bandit setting (a special case of our problem), the established lower bound is $O(d\sqrt{T})$, indicating near-optimality of our result. See also [Chen et al., 2022], which gives a $O(d\sqrt{T})$ lower bound in the game setting with linear function approximation.
>
>
> > **Regarding description of Thompson Sampling**
>
> Thank you for the suggestion on Thompson Sampling. While you mentioned references [5-7], these weren't specified in your comments. We are happy to add the literature you mention once you provide them to us. In addition, we will include more discussion about Thompson Sampling in the revised paper.
>
> >**Q1: regarding NE computing**
>
> Thank you for your question.
>
> - MEX has substantially higher computational costs than our approach. MEX requires each agent to solve a bilevel optimization problem in each iteration, where the lower-level problem involves computing the equilibrium (which is also assumed to be exact computation), see line 3 and 5 in their Algorithm 2, as well as the discussion in Section 1 of our paper. This creates a nested optimization structure that is inherently more computationally intensive. In contrast, our approach is more efficient because we compute the equilibrium only once per iteration.
>
> - The computational cost of finding an approximate Nash equilibrium does not affect our theoretical guarantees (regret bounds and sample complexity bounds). This is because the equilibrium is computed using our transition kernel estimator $M_{f_{t-1}}$ without requiring additional environment interactions. Therefore, while equilibrium computation adds to wall-clock time, it doesn't increase the sample complexity.
>
> >**Q2: regarding linear parameterization on matrix games**
>
> We have several considerations to address your concerns:
>
> - First, one apparent benefit of introducing function approximation is it allows us to tackle problems with infinite action pairs (i.e., $m,n=+\infty$ or the action space is continuous), one application is the stochastic linear bandit (Example 4.2 in [Lattimore & Szepesvári, 2020]).
>
> - You are correct that the regime of interest is $d< mn$. When the system of linear equations do not have a solution, one could relax the realizability assumption (Assumption 2.2) to allow approximation errors, which can be incorporated straightforwardly into our analysis in a similar manner as in [Yuan et al., 2023; Xie et al., 2021]. Even when exact representation is impossible, approximate solutions often perform remarkably well in practice [Bertsekas & Tsitsiklis, 1996].
>
> -   In addition, function approximation offers substantial benefits including computational efficiency for large action spaces, generalization across similar state-action pairs, transfer learning capabilities and compatibility with gradient-based optimization.
>
> - We introduce linear approximation in matrix games also as a "warm-up" for our main contributions in Markov games. This helps establish key intuitions and technical foundations before extending to the more complex Markov game setting.
>
> ---
>
> Zhang et al., 2022. Policy optimization for markov games: Unified framework and faster convergence.
>
> Cen et al., 2022. Faster Last-iterate Convergence of Policy Optimization in Zero-Sum Markov Games.
>
> Chen et al., 2022. Almost optimal algorithms for two-player zero-sum linear mixture Markov games.
>
> T Lattimore, C Szepesvári, 2020. Bandit algorithms.
>
> Yuan et al., 2023. Linear Convergence of Natural Policy Gradient Methods with Log-Linear Policies.
>
> Xie et al., 2021. Bellman-consistent Pessimism for Offline Reinforcement Learning.
>
> Bertsekas, D. P., & Tsitsiklis, J. N. (1996). Neuro-Dynamic Programming.

---

> > ### Comment · Reviewer_NUsj · 2025-04-02
> >
> > Thank you for your response, and I am sorry for missing the reference in the review. In particular, Ref. [4] is "Provable self-play algorithms for competitive reinforcement learning", and it would be appreciated if you could comment on it.
> >
> > ## Reference
> > [1] Zhang, Tong. "Feel-good thompson sampling for contextual bandits and reinforcement learning." SIAM Journal on Mathematics of Data Science 4.2 (2022): 834-857.
> > [2] Liu, Zhihan, et al. "Maximize to explore: One objective function fusing estimation, planning, and exploration." Advances in Neural Information Processing Systems 36 (2023): 22151-22165.
> > [3] Chen, Zixiang, Dongruo Zhou, and Quanquan Gu. "Almost optimal algorithms for two-player zero-sum linear mixture markov games." International Conference on Algorithmic Learning Theory. PMLR, 2022.
> > [4] Bai, Yu, and Chi Jin. "Provable self-play algorithms for competitive reinforcement learning." International conference on machine learning. PMLR, 2020.
> > [5] Wu, Runzhe, and Wen Sun. "Making RL with Preference-based Feedback Efficient via Randomization." The Twelfth International Conference on Learning Representations.
> > [6] Xiong, Wei, et al. "A self-play posterior sampling algorithm for zero-sum markov games." International Conference on Machine Learning. PMLR, 2022.
> > [7] Zhang, Qiaosheng, et al. "Provably efficient information-directed sampling algorithms for multi-agent reinforcement learning." arXiv preprint arXiv:2404.19292 (2024).

---

> > > ### Author Response · Authors · 2025-04-02
> > >
> > > # Re: Reviewer NUsj (3)
> > >
> > > Thank you for your response and the additional information!
> > >
> > > - We misidentified which paper [4] was referring to, but our argument regarding the question about 'regret bound worse than [4] in the tabular setting' still holds.
> > >
> > > - We'll add the following discussion in our revised paper:
> > >
> > > > [6,7] propose sampling-based algorithms which maintain and sample from a posterior distribution in each iteration, offering a complementary perspective to our optimization-based VMG approach.
> > >
> > > Thank you for your suggestion.

---

### Official Review · Reviewer_4hiw · 2025-03-16

**Overall Recommendation:** 3

**Summary:**

This paper introduces VMG, a model-based MARL algorithm that balances exploration and exploitation without requiring explicit uncertainty quantification. By biasing model estimation toward higher collective best-response values, VMG enables simultaneous and uncoupled policy updates while achieving near-optimal regret for Nash equilibria (NE) in two-player zero-sum games and coarse correlated equilibria (CCE) in multi-player general-sum games.

**Claims And Evidence:**

The claims made in the submission are supported by clear and convincing evidence.

**Essential References Not Discussed:**

[1] also propose a quite similar algorithm with general function approximation to solve the multi-agent RL without bonus or confidence level set, which is highly relevant to the key contribution in this paper.

[1] Xiong, Nuoya, et al. "SAMPLE-EFFICIENT MULTI-AGENT RL: AN OPTIMIZATION PERSPECTIVE." 12th International Conference on Learning Representations, ICLR 2024. 2024.

**Experimental Designs Or Analyses:**

No experiments in the paper.

**Methods And Evaluation Criteria:**

Yes.

**Other Comments Or Suggestions:**

No

**Other Strengths And Weaknesses:**

Weakness 1: Though the paper provides the theoretical solution to the multi-agent RL problems, it remains unclear how to design the practical version of the algorithms and if the practical method works well.

Weakness 2:  In Line 412, the author claims that 'To the best of our knowledge, this is the first result that establishes a near-optimal sublinear regret for general-sum Markov games without explicit uncertainty quantification via constructing bonus functions or uncertainty sets', which is overclaimed. [1] also propose a quite similar algorithm with general function approximation to solve the multi-agent RL without bonus or confidence level set.

Weakness 3: The assumption of the linear mixture model is too restrictive. Could the author extend the current results to accommodate general function approximation as in [1] and [2]?




[1] Xiong, Nuoya, et al. "SAMPLE-EFFICIENT MULTI-AGENT RL: AN OPTIMIZATION PERSPECTIVE." 12th International Conference on Learning Representations, ICLR 2024. 2024.

[2] Wang, Yuanhao, et al. "Breaking the curse of multiagency: Provably efficient decentralized multi-agent rl with function approximation." The Thirty Sixth Annual Conference on Learning Theory. PMLR, 2023.

**Questions For Authors:**

Question 1: The authors use the V-learning-like method in multi-agent RL but the proved bound does not include the cardinality of the policy space. Could the authors give some explanations on this?

Question 2: In the linear mixture model setting, could one design a bonus to achieve a similar result?

**Relation To Broader Scientific Literature:**

This paper studies how to achieve sample efficiency for online multi-agent RL without a confidence level or hand-crafted bonus, which has been studied in the single-agent RL setting. Previous work on online RL that is proven to be sample efficient either requires a confidence level, hand-crafted bonus, or complicated sampling procedure, which are hard to apply in practice. Hence, it is meaningful to study how to design a sample efficient algorithm without a confidence level or hand-crafted bonus for online multi-agent RL.

**Theoretical Claims:**

I roughly checked the proofs in this paper.

---

> ### Author Rebuttal · Authors · 2025-03-30
>
> # Response to Reviewer 4hiw
>
> Thank you for your feedback. If our responses resolve your questions, we'd appreciate your consideration in raising the score.
>
> > **missing the reference [Xiong et al, 2024] and overclaim of contribution**
>
> Thank you for bringing up this highly relevant work! We will add a detailed discussion in our revised paper. We acknowledge that [Xiong et al, 2024] is the first paper that addressed the general-sum Markov game setting, and will adjust our claim accordingly. Nonetheless, we want to emphasize that our algorithmic design is much simpler and computationally tractable compared with that in [Xiong et al, 2024],  which requires solving a normal-form game whose size scales exponentially with the number of agents, as elaborated below.
>
> Specifically, their algorithm (Alg. 1) requires evaluating the value function for **all pure policies** (whose size scales with the size of the joint action space) in the policy space (line 3) and solving a bilevel optimization problem with an equilibrium computation at the upper level (line 4) for a normal-form game defined over the set of all pure policies. These steps are computationally prohibitive since the size of the joint action space scales exponentially in the number of agents, contrasting with the tractable design of our approach. Furthermore, when the size of the action space is infinite, their algorithm is computationally intractable.
>
> In addition, for linear mixture model, they give a regret bound $O(dH^5\sqrt{T})$ (see their discussion below Theorem 5.14), which is worse than our  $O(dH^{3/2}\sqrt{T})$ bound given in Theorem 3.3. Besides, our VMG (Algorithm 6) could be extended to the infinite-horizon setting, while [Xiong et al, 2024] only considers the finite-horizon setting.
>
> > **W1: practical version of VMG**
>
> We stress that each of our proposed VMG algorithms can be implemented by standard procedures. To demonstrate this, we provide a complete Python implementation and experimental results of our Algorithm 2 on randomly generated MDPs for two-player zero-sum Markov games under the linear mixture model setting in this anonymous repo https://anonymous.4open.science/r/VMG-6BC7, where the python code, important exepriment details and the curve of duality gap ($\max_{\pi_1} V^{\pi_1,\pi_{2,t}}(s_0) - \min_{\pi_2} V^{\pi_{1,t},\pi_2}(s_0)$ at iteration t) v.s. iteration numbers are provided.
>
>
> > **W3: going beyond the linear mixture assumption**
>
> While our paper primarily focuses on linear mixture Markov games, our algorithms can indeed be implemented with general function approximation, and our theoretical results readily extend to accommodate this broader framework.
>
> In [Xiong et al, 2024], the authors introduce the Multi-Agent Decoupling Coefficient (MADC) complexity measure and establish the Finite MADC assumption (Assumption 3.11). They demonstrate that this assumption naturally holds for linear mixture Markov games (as shown in Example 5.13 and Theorem 5.14). Notably, our framework provides similar bounds under this same Assumption 3.11, establishing a clear path for extending our results beyond linear mixture models.
>
> > **Q1: no cardinality of the policy space in the given bounds**
>
> We are not quite sure which V-learning method you refer to, and we'll use [Wang et al., 2023] you mentioned previously as our reference point for providing an explanation - please let us know if you are referring to other papers. The difference in sample complexity bounds between our VMG approach and [Wang et al., 2023] is due to fundamentally different exploration mechanisms:
>
> - Our algorithms achieve exploration implicitly by biasing model estimates toward parameters that maximize best-response values.
>
> - [Wang et al., 2023] is based on policy replay and explicitly maintains a set of policies for exploration (represented by $\Gamma_{\text{explore}}(\pi,\pi')$), and thus their bound (given in Corollary 3) depends on the cardinality $|\Gamma_{\text{explore}}(\pi,\pi')|$.
>
> >**Q2: linear mixture model assumption**
>
> [Chen et al., 2022] constructs bonus in the linear mixture model setting. However, their Nash-UCRL algorithm requires multiple complex and computationally intensive subroutines. Specifically, their approach necessitates matrix inversions to be computed in each iteration of Algorithms 1-3. These matrix operations scale poorly with state and action space dimensions, becoming a bottleneck in large-scale applications.
>
>
> ---
>
> Xiong, Nuoya, et al. "SAMPLE-EFFICIENT MULTI-AGENT RL: AN OPTIMIZATION PERSPECTIVE." ICLR 2024.
>
> Wang, Yuanhao, et al. "Breaking the curse of multiagency: Provably efficient decentralized multi-agent rl with function approximation." PMLR, 2023.
>
> Chen et al., 2022. Almost Optimal Algorithms for Two-player Zero-Sum Linear Mixture Markov Games
> Ni et al., 2022. Representation Learning for General-sum Low-rank Markov Games.
>
> Duan et al., 2016. RL^2: Fast Reinforcement Learning via Slow Reinforcement Learning.

---

> > ### Comment · Reviewer_4hiw · 2025-04-01
> >
> > Thank the authors for their responses. After carefully re-examing the methods proposed in this paper and in [Xiong et al, 2024] and [Wang et al., 2023], I find that the key difference is that this paper studies the **model-based** setting in the general markov game, which means that we need to estimate the transition kernels. Instead, [Xiong et al, 2024] and [Wang et al., 2023] study the **model-free** setting, where they estimate the Q-function (payoff function) and need to apply concentration on the Bellman equation with a policy evaluation bellman operator (instead of the max operator in the single-agent MDP). In that case, their final bound will include a cardinality of the joint policy class. I would recommend the authors discuss these settings carefully in the revision. From my understanding, there is no absolute advantage of model-based or model-free methods and the authors should discuss when to apply the model-based method. Given the current reply, I would increase the score to 3.

---

### Decision · Program_Chairs · 2025-05-01

**Decision:**

Accept (poster)

**Comment:**

This paper introduces VMG, a novel model-based algorithm for multi-agent reinforcement learning (MARL) in Markov games. It promotes exploration without requiring bonus-based uncertainty quantification, instead biasing model estimates toward higher collective best-response values. The method allows for uncoupled, simultaneous policy updates and achieves near-optimal regret bounds in both two-player zero-sum and multi-player general-sum settings with linear function approximation. While reviewers commend the theoretical rigor and conceptual elegance, they highlight concerns about overclaimed novelty, computational efficiency, and the practicality of implementation. Comparisons with existing works like MEX, VPO, and Foster et al.'s frameworks sparked discussion, with the authors defending their approach’s simplicity, scalability, and generalizability. Despite some disagreement over its significance, the paper received generally positive feedback and weak acceptance.